# FedAvg with Fine Tuning: Local Updates Lead to Representation Learning

**Liam Collins**
ECE Department
The University of Texas at Austin
`liamc@utexas.edu`

**Hamed Hassani**
ESE Department
University of Pennsylvania
`hassani@seas.upenn.edu`

**Aryan Mokhtari**
ECE Department
The University of Texas at Austin
`mokhtari@austin.utexas.edu`

**Sanjay Shakkottai**
ECE Department
The University of Texas at Austin
`sanjay.shakkottai@utexas.edu`

## Abstract

The Federated Averaging (FedAvg) algorithm, which consists of alternating between a few local stochastic gradient updates at client nodes, followed by a model averaging update at the server, is perhaps the most commonly used method in Federated Learning. Notwithstanding its simplicity, several empirical studies have illustrated that the model output by FedAvg leads to a model that generalizes well to new unseen tasks after a few fine-tuning steps. This surprising performance of such a simple method, however, is not fully understood from a theoretical point of view. In this paper, we formally investigate this phenomenon in the multi-task linear regression setting. We show that the reason behind the generalizability of the FedAvg output is FedAvg's power in learning the common data representation among the clients' tasks, by leveraging the diversity among client data distributions via multiple local updates between communication rounds. We formally establish the iteration complexity required by the clients for proving such result in the setting where the underlying shared representation is a linear map. To the best of our knowledge, this is the first result showing that FedAvg learns an expressive representation in any setting. Moreover, we show that multiple local updates between communication rounds are necessary for representation learning, as distributed gradient methods that make only one local update between rounds provably cannot recover the ground-truth representation in the linear setting, and empirically yield neural network representations that generalize drastically worse to new clients than those learned by FedAvg trained on heterogeneous image classification datasets.

## 1 Introduction

Federated Learning (FL) [1] provides a communication-efficient and privacy preserving means to learn from data distributed across clients such as cell phones, autonomous vehicles, and hospitals. FL aims for each client to benefit from collaborating in the learning process without sacrificing data privacy or paying a substantial communication cost. Federated Averaging (FedAvg) [1] is the predominant FL algorithm. In FedAvg, also known as Local SGD [2–4], the clients achieve communication efficiency by making multiple local updates of a shared global model before sending the result to the server, which averages the locally updated models to compute the next global model.

FedAvg is motivated by settings with *homogeneous* data across clients, since multiple local updates should improve model performance on all other clients' data when their data is similar. In contrast,

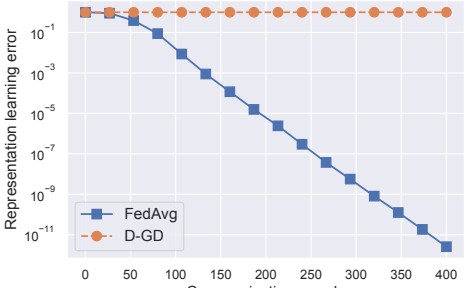
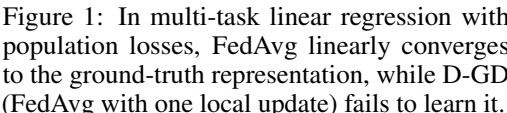

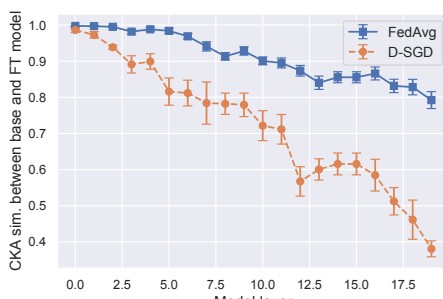

Figure 1: In multi-task linear regression with population losses, FedAvg linearly converges to the ground-truth representation, while D-GD (FedAvg with one local update) fails to learn it.

Figure 2: The NN representation learned by FedAvg on CIFAR-100 with 5 classes/client does not change significantly when fine-tuned on a new dataset (CIFAR-10), unlike D-SGD.

FedAvg faces two major challenges in more realistic *heterogeneous* data settings: learning a single global model may not necessarily yield good performance for each individual client, and, multiple local updates may cause the FedAvg updates to drift away from solutions of the global objective [5–9]. Despite these challenges, several empirical studies [10–12] have observed that this shared global model trained by FedAvg *with several local updates per round* when further fine-tuned for individual clients is surprisingly effective in heterogeneous FL settings. These studies motivate us to explore the impact of local updates on post-fine-tuning performance.

Meanwhile, a large number of recent works have shown that representation learning is a powerful paradigm for attaining high performance in multi-task settings, including FL. This is because the tasks' data often share a small set of features which are useful for downstream tasks, even if the datasets as a whole are heterogeneous. Consider, for example, heterogeneous federated image classification in which each client (task) may have images of different types of animals. It is safe to assume the images share a small number of features, such as body shape and color, which admit a simple and accurate mapping from feature space to label space. Since the number of important features is much smaller than the dimension of the data, knowing these features greatly simplifies each client's task.

To explore the connection between local updates and representation learning, we first study multi-task linear regression sharing a common ground-truth representation (Figure 1). We observe that FedAvg converges (exponentially fast) to the ground-truth representation in principal angle distance, while Distributed-GD (D-GD), which is effectively FedAvg with one local gradient update, fails to learn the shared representation. A similar concept can be shown in the nonlinear setting. We study a multi-layer CNN on a heterogeneous partition of CIFAR-100 (Figure 2). Since there is not necessarily a ground-truth model here, we evaluate representation learning as follows. We first train the models with FedAvg and Distributed-SGD (D-SGD) then fine-tune the pre-trained models on clients from a new dataset, CIFAR-10. Finally we evaluate the quality of the learned representation by measuring the amount that each model layer changes during fine-tuning using CKA similarity [13]. Observe that the early layers of FedAvg's pre-trained model (corresponding to the representation) change much less than those of D-SGD. More details for both experiments are in Section 5 and Appendix C. These observations suggest that FedAvg learns a shared representation that generalizes to new clients, even when trained in a heterogeneous setting. Hence, a natural question that arises is:

*Does FedAvg provably learn effective representations of heterogeneous data?*

We answer "yes" to this question by proving that FedAvg recovers the ground-truth representation in the case of multi-task linear regression. Critically, we show that FedAvg's local updates leverage the diversity among client data distributions to learn their common representation. This is surprising because FedAvg is a general-purpose algorithm not designed for representation learning. Our analysis thus yields new insights on how FedAvg finds generalizable models. Our contributions are:

- **Representation learning guarantees.** We study the behavior of FedAvg in multi-task linear regression with common representation. Here, each client aims to solve a $d$-dimensional regression with ground-truth solution that belongs to a shared $k$-dimensional subspace of $\mathbb{R}^d$, where $k \ll d$. Our results show that FedAvg with $\tau \geq 2$ local updates learns the representation

at a linear rate when each client accesses population gradients. *To the best of our knowledge, this is the first result showing that FedAvg learns an effective representation in any setting.*

- **Insights on the importance of local updates.** Our analysis reveals that executing more than one local update between communication rounds *exploits the diversity* of the clients' ground truth regressors to improve the learned representation in all $k$ directions in the linear setting. In contrast, we prove that D-GD fails to learn the representation.

- **Empirical evidence of representation learning.** We provide experimental results showing Fedavg learns a generalizable representation when we use deep neural networks on image classification datasets. In contrast, the representations learned by D-SGD generalize drastically worse to data from new clients. This suggests that the main message of our theoretical results that local updates facilitate representation learning can generalize to more complex scenarios beyond the bilinear setting.

**Related work.** Recently there has been a surge of interest motivated by FL in analyzing FedAvg/Local SGD in heterogeneous settings. Multiple works have shown that FedAvg converges to a global optimum (resp. stationary point) of the global objective in convex (resp. nonconvex) settings but with decaying learning rate [5, 14–17], leading to sublinear rates and communication complexity sometimes dominated by Distributed-SGD [18]. These results are tight in the sense that FedAvg with fixed learning rate may *not* converge to a stationary point of the global objective in the presence of data heterogeneity, as its multiple local updates cause it to optimize a distinct, unknown objective [6–9, 16, 19, 20]. Several methods have tried to correct this objective inconsistency via gradient tracking [5, 19, 21–25], local regularization [20, 26–28], operator splitting [7], and strategic client sampling [29–31]. In contrast, we show that local updates with *constant* learning rate benefit *learning* in heterogeneous settings by resulting in linear convergence to generalizable models.

Several papers have also studied FedAvg from a generalization perspective. It was shown in [32] that in a setting with strongly convex losses, either local training or FedAvg with fine-tuning (but not both) achieves minimax risk, depending on the level of data heterogeneity. Similarly, [33] argued that FedAvg with fine-tuning generalizes as well as more sophisticated methods, including Model-Agnostic Meta-Learning (MAML) [34, 35], in a strongly convex regularized linear regression setting. Additional work has studied the generalization of FedAvg in kernel regression, but for convex objectives that do not allow for representation learning [36], and the generalization of a variant of FedAvg, known as Reptile [37], on wide two-layer ReLU networks with homogeneous data [38]. We focus on the multi-task linear representation learning setting [39], which has become popular in recent years as it is an expressive but tractable nonconvex setting for studying the sample-complexity benefits of learning representations and the representation learning abilities of popular algorithms in data heterogeneous settings [11, 40–46]. Remarkably, our study of FedAvg reveals that it can learn an effective representation even though it was not designed for this goal, unlike a variety of personalized FL methods specifically tailored for representation learning [11, 47–51].

**Notations.** We use $\mathcal{N}(\mathbf{u}, \boldsymbol{\Sigma})$ to signify the multivariate Gaussian distribution with mean $\mathbf{u}$ and covariance $\boldsymbol{\Sigma}$. $\mathcal{O}^{d \times k}$ denotes the set of matrices in $\mathbb{R}^{d \times k}$ with orthonormal columns. The notation $\mathrm{col}(\mathbf{B})$ represents the column space of the matrix $\mathbf{B}$, and $\mathrm{col}(\mathbf{B})^{\perp}$ is the orthogonal complement to this space. The norm $\| \cdot \|$ is the spectral norm and $\mathbf{I}_d$ is the identity matrix in $\mathbb{R}^{d \times d}$. We use $[m]$ to indicate the set of natural numbers up to and including $m$.

## 2 Problem Formulation

Consider a federated setting with a central server and $M$ clients. Each client $i \in [M]$ has a training dataset $\hat{\mathcal{D}}_i$ of $n_i$ labeled samples drawn from a distribution $\mathcal{D}_i$ over $\mathcal{X} \times \mathcal{Y}$, where $\mathcal{X}$ is the input space and $\mathcal{Y}$ is the label space. The learning model is given by $h_{\boldsymbol{\theta}} : \mathcal{X} \to \mathcal{Y}$ for model parameters $\boldsymbol{\theta} \in \mathbb{R}^D$. The loss of the model on a sample $(\mathbf{x}, \mathbf{y}) \in \mathcal{X} \times \mathcal{Y}$ is given by $\ell(h_{\boldsymbol{\theta}}(\mathbf{x}), \mathbf{y})$, which may be, for example, the squared or cross entropy loss. The loss of model parameters $\boldsymbol{\theta}$ on the $i$-th client is the average loss of the model $h_{\boldsymbol{\theta}}$ on the samples in $\hat{\mathcal{D}}_i$, namely $f_i(\boldsymbol{\theta}) := \frac{1}{n_i} \sum_{j=1}^{n_i} \ell(h_{\boldsymbol{\theta}}(\mathbf{x}_{i,j}), \mathbf{y}_{i,j})$, where $(\mathbf{x}_{i,j}, \mathbf{y}_{i,j})$ is the $j$-th sample in $\hat{\mathcal{D}}_i$. The server aims to leverage all of the data across clients to find models that achieve small loss $f_i(\boldsymbol{\theta})$ for each client. To do so, the standard approach is to find

a single model $\boldsymbol{\theta}$ that minimizes the average of the client losses weighted by number of samples:

$$\min_{\boldsymbol{\theta}} \frac{1}{N} \sum_{i=1}^{M} n_i f_i(\boldsymbol{\theta}) \; = \; \frac{1}{N} \sum_{i=1}^{M} \sum_{j \in \hat{\mathcal{D}}_i} \ell(h_{\boldsymbol{\theta}}(\mathbf{x}_{i,j}), \mathbf{y}_{i,j})), \tag{1}$$

where $N = \sum_{i=1}^{M} n_i$. Due to communication and privacy constraints, the clients cannot share their local data $\hat{\mathcal{D}}_i$, so (1) must be solved in a federated manner.

**FedAvg.** The most common FL method is FedAvg. On each round $t$ of FedAvg, the server uniformly samples a set $\mathcal{I}_t$ of $m \leq M$ clients. Each selected client receives the current global parameters $\boldsymbol{\theta}_t$, executes multiple SGD steps on its local data starting from $\boldsymbol{\theta}_t$, then sends the result back to the server. The server then computes $\boldsymbol{\theta}_{t+1}$ as the weighted average of the updates. Specifically, upon receiving the global model $\boldsymbol{\theta}_t$, client $i$ computes

$$\boldsymbol{\theta}_{t,i,s+1} = \boldsymbol{\theta}_{t,i,s} - \alpha \mathbf{g}_{t,i,s}(\boldsymbol{\theta}_{t,i,s}), \tag{2}$$

for $s = 0, \ldots, \tau - 1$, where $\tau$ is the number of local steps, $\boldsymbol{\theta}_{t,i,0} = \boldsymbol{\theta}_t$ and $\mathbf{g}_{t,i,s}(\boldsymbol{\theta}_{t,i,s})$ is a stochastic gradient of $f_i$ evaluated at $\boldsymbol{\theta}_{t,i,s}$ using $b$ samples from $\hat{\mathcal{D}}_i$. The client then sends $\boldsymbol{\theta}_{t,i,\tau}$ back to the server, which computes the next global iterate as:

$$\boldsymbol{\theta}_{t+1} = \frac{1}{N_t} \sum_{i \in \mathcal{I}_t} n_i \boldsymbol{\theta}_{t,i,\tau}, \tag{3}$$

where $N_t := \sum_{i \in \mathcal{I}_t} n_i$. Note that $\tau = 1$ corresponds to D-SGD, also known as mini-batch SGD whose convergence properties are well-understood [18, 52–54]. FedAvg improves the communication efficiency of D-SGD by making $\tau \geq 2$ local updates between communication rounds.

**Fine-tuning.** After training for $T$ communication rounds, the global parameters $\boldsymbol{\theta}_T$ learned by FedAvg are typically fine-tuned on each client before testing. In particular, starting from $\boldsymbol{\theta}_T$, client $i$ executes $\tau'$ steps of SGD on its local data as follows:

$$\boldsymbol{\theta}_{T,i,s+1} = \boldsymbol{\theta}_{T,i,s} - \alpha \mathbf{g}_{T,i,s}(\boldsymbol{\theta}_{t,i,s}) \tag{4}$$

for $s = 0, \ldots, \tau - 1$. The fine-tuned model ultimately used for testing is $\boldsymbol{\theta}_{T,i,\tau'}$. Note that a new client, indexed by $M + 1$, entering the system after FedAvg training has completed can also fine-tune $\boldsymbol{\theta}_T$ using the same procedure to obtain a personalized solution $\boldsymbol{\theta}_{T,M+1,\tau'}$.

**Representation learning.** We aim to answer why the fine-tuned models $\{\boldsymbol{\theta}_{T,i,\tau'}\}_{i=1}^{M+1}$ perform well in practice by taking a representation learning perspective. We show that the output of FedAvg, i.e., $\theta_T$, has learned the common data representation among clients assuming that such a representation exists. To formalize this result, we consider a class of models that can be written as the composition of a representation $h^{\text{rep}}$ and a prediction module, i.e. head, denoted as $h^{\text{head}}$. Let the model parameters be split as $\boldsymbol{\theta} := [\boldsymbol{\phi}, \boldsymbol{\psi}]$, where $\boldsymbol{\phi}$ contains the representation parameters and $\boldsymbol{\psi}$ contains the head parameters. Then, for any $\mathbf{x} \in \mathcal{X}$, the prediction of the learning model is $h_{\boldsymbol{\theta}}(\mathbf{x}) = (h_{\boldsymbol{\psi}}^{\text{head}} \circ h_{\boldsymbol{\phi}}^{\text{rep}})(\mathbf{x}) = h_{\boldsymbol{\psi}}^{\text{head}}(h_{\boldsymbol{\phi}}^{\text{rep}}(\mathbf{x}))$. For instance, if $h_{\boldsymbol{\theta}}$ is a neural network with weights $\boldsymbol{\theta}$, then $h_{\boldsymbol{\phi}}^{\text{rep}}$ is the first many layers of the network with weights $\boldsymbol{\phi}$, and $h_{\boldsymbol{\psi}}^{\text{head}}$ is the network last few layers with weights $\boldsymbol{\psi}$. A standard assumption in multi-task settings, including the settings we consider, is the existence of a common representation $h_{\boldsymbol{\phi}_*}^{\text{rep}}$ that admits an easily learnable head $h_{\boldsymbol{\psi}_{*,i}}^{\text{rep}}$ such that $h_{\boldsymbol{\psi}_{*,i}}^{\text{head}} \circ h_{\boldsymbol{\phi}_*}^{\text{rep}}$ performs well for task $i$. As a result, in these settings it is of interest to all the clients to learn $h_{\boldsymbol{\phi}_*}^{\text{rep}}$.

## 3   Main Results

To rigorously study the representation learning abilities of FedAvg, we employ the standard setting used for algorithmic representation learning analysis: multi-task linear regression [11, 40, 41, 46, 55, 56]. In this setting, samples $(\mathbf{x}_{i,j}, y_{i,j})$ for each client $i$ are drawn independently from a distribution $\mathcal{D}_i$ on $\mathbb{R}^d \times \mathbb{R}$ such that

$$\mathbf{x}_{i,j} \overset{\text{i.i.d.}}{\sim} p_{\mathbf{x}}, \;\; y_{i,j} = \langle \boldsymbol{\beta}_{*,i}, \mathbf{x}_{i,j} \rangle + \zeta_{i,j} \;\; \text{where} \;\; \zeta_{i,j} \overset{\text{i.i.d.}}{\sim} p_{\zeta}$$

for an unobserved ground-truth regressor $\boldsymbol{\beta}_{*,i} \in \mathbb{R}^d$ and label noise $\zeta_{i,j}$. We assume the distributions $p_{\mathbf{x}}$ and $p_{\zeta}$ are such that $\mathbb{E}[\mathbf{x}_{i,j}] = \mathbf{0}, \mathbb{E}[\mathbf{x}_{i,j} \mathbf{x}_{i,j}^{\top}] = \mathbf{I}_d, \mathbb{E}[\zeta_{i,j}] = 0$.

To incentivize representation learning, each $\boldsymbol{\beta}_{*,i}$ belongs to the same $k$-dimensional subspace of $\mathbb{R}^d$, where $k \ll d$. Let $\mathbf{B}_* \in \mathcal{O}^{d \times k}$ have columns that form an orthogonal basis for the shared subspace, so that $\boldsymbol{\beta}_{*,i} = \mathbf{B}_* \mathbf{w}_{*,i}$ for some $\mathbf{w}_{*,i} \in \mathbb{R}^k$ for each $i$. In other words, there exists a low-dimensional set of parameters known as the "head" that can specify the ground-truth model for client $i$ once the shared representation, i.e., $\mathrm{col}(\mathbf{B}_*)$, is known. It is advantageous to learn $\mathrm{col}(\mathbf{B}_*)$ because once it is known, all clients (including potentially new clients entering the system) have sample complexity $O(k) \ll d$ as they only need to learn the parameters of their head [40, 41].

Each client $i$ ultimately aims to learn a model $\hat{\boldsymbol{\beta}}_i$ that approximates $\boldsymbol{\beta}_{*,i}$ in order to achieve good generalization on its local distribution. To eventually achieve this for each client, FedAvg with fine-tuning first aims to learn a global model consisting of a representation $\mathbf{B} \in \mathbb{R}^{d \times k}$ and a head $\mathbf{w} \in \mathbb{R}^k$ that minimizes the average loss across clients. The loss for client $i$ is $f_i(\mathbf{B}, \mathbf{w}) := \frac{1}{2n_i} \sum_{j=1}^{n_i} (y_{i,j} - \langle \mathbf{B}\mathbf{w}, \mathbf{x}_{i,j} \rangle)^2$, i.e. the average squared loss on the local data, so FedAvg tries to learn a global model that solves the nonconvex problem:

$$\min_{\mathbf{B} \in \mathbb{R}^{d \times k}, \mathbf{w} \in \mathbb{R}^k} \frac{1}{N} \sum_{i=1}^{M} n_i \left\{ f_i(\mathbf{B}, \mathbf{w}) := \frac{1}{2n_i} \sum_{j=1}^{n_i} (y_{i,j} - \langle \mathbf{B}\mathbf{w}, \mathbf{x}_{i,j} \rangle)^2 \right\}. \tag{5}$$

where $N = \sum_{i=1}^{M} n_i$. To solve (5) in a distributed manner, FedAvg dictates that each client makes a series of local updates of the current global model before returning the models to the server for averaging (see Section 2). Our aim is to show that FedAvg training learns the column space of $\mathbf{B}_*$. First, we make standard diversity and normalization assumptions on the ground-truth heads.

**Assumption 1** (Client normalization). *There exists $L < \infty$ s.t. $\forall i \in [M]$, $\|\mathbf{w}_{*,i}\|_2 \leq L\sqrt{k}$.*

**Assumption 2** (Client diversity). *There exists $\mu > 0$ s.t. $\sigma_{\min}\left(\frac{1}{M} \sum_{i=1}^{M} (\mathbf{w}_{*,i} - \bar{\mathbf{w}}_*)(\mathbf{w}_{*,i} - \bar{\mathbf{w}}_*)^\top\right) \geq \mu^2$, where $\bar{\mathbf{w}}_* := \frac{1}{M} \sum_{i=1}^{M} \mathbf{w}_{*,i}$. Define $\kappa := {}^L/_\mu$.*

Assumption 2 is very similar to typical task diversity assumptions except that it quantifies the diversity of the centered rather than un-centered tasks [40, 41]. Intuitively, task diversity is required so that all of the directions in $\mathrm{col}(\mathbf{B}_*)$ are observed. The smaller $\kappa$, the more evenly spread the ground-truth heads are, and the larger the task (i.e. client) diversity. Next, to obtain convergence results we must define the variance of the ground-truth heads and the principal angle distance between representations.

**Definition 1** (Client variance). *For $\gamma > 0$, define: $\gamma^2 := \frac{1}{kM} \sum_{i=1}^{M} \|\mathbf{w}_{*,i} - \bar{\mathbf{w}}_*\|^2$, where $\bar{\mathbf{w}}_*$ is defined in Assumption 2. For $H > 0$, define $H^4 := \frac{1}{k^2 M} \sum_{i=1}^{M} \|\mathbf{w}_{*,i}\mathbf{w}_{*,i}^\top - \frac{1}{M} \sum_{i'=1}^{M} \mathbf{w}_{*,i'}\mathbf{w}_{*,i'}^\top\|^2$.*

**Definition 2** (Principal angle distance). *For two matrices $\mathbf{B}_1, \mathbf{B}_2 \in \mathbb{R}^{d \times k}$, the principal angle distance between $\mathbf{B}_1$ and $\mathbf{B}_2$ is defined as $\mathrm{dist}(\mathbf{B}_1, \mathbf{B}_2) := \|\bar{\mathbf{B}}_{1,\perp}^\top \bar{\mathbf{B}}_2\|_2$, where the columns of $\bar{\mathbf{B}}_{1,\perp} \in \mathcal{O}^{d \times (d-k)}$ and $\bar{\mathbf{B}}_2 \in \mathcal{O}^{d \times k}$ form orthonormal bases for $\mathrm{col}(\mathbf{B}_1)^\perp$ and $\mathrm{col}(\mathbf{B}_2)$, respectively.*

Intuitively, the principal angle distance between $\mathbf{B}_1$ and $\mathbf{B}_2$ is the sine of the largest angle between the subspaces spanned by their columns. Now we are ready to state our main result. We suppose each client has $n_i = \infty$ samples, i.e. it accesses the gradients of the population loss on its local distribution.

**Theorem 1.** *Consider the case that each client takes gradient steps with respect to their population loss $f_i(\mathbf{B}, \mathbf{w}) := \frac{1}{2} \|\mathbf{B}\mathbf{w} - \mathbf{B}_* \mathbf{w}_{*,i}\|^2$ and all losses are weighed equally in the global objective. Suppose Assumptions 1-2 hold, the number of clients participating each round satisfies $m \geq \min(M, 20((\gamma/L)^2 + (H/L)^4)(\alpha L\sqrt{k})^{-4} \log(kT))$, and the initial parameters satisfy (i) $\delta_0 := \mathrm{dist}(\mathbf{B}_0, \mathbf{B}_*) \leq \sqrt{1-E_0}$ for any $E_0 \in (0,1]$, (ii) $\|\mathbf{I} - \alpha \mathbf{B}_0^\top \mathbf{B}_0\|_2 = O(\alpha^2 \tau L^2 \kappa^2 k^2)$ and (iii) $\|\mathbf{w}_0\|_2 = O(\alpha^{2.5} \tau L^3 k^{1.5})$. Choose step size $\alpha = O(\frac{1-\delta_0}{\sqrt{\tau}L\kappa^2 k^{1.5}})$. Then for any $\epsilon \in (0,1)$, the distance of the representation learned by FedAvg with $\tau \geq 2$ local updates satisfies $\mathrm{dist}(\mathbf{B}_T, \mathbf{B}_*) < \epsilon$ after at most $T = O\left(\frac{\log(1/\epsilon)}{\alpha^2 \tau \mu^2 E_0}\right)$ communication rounds with probability at least $1 - 4(kT)^{-99}$.*

Theorem 1 shows that FedAvg converges exponentially fast to the ground-truth representation when executed on the clients' population losses. We provide intuition for the proof in Section 4 and the full proof in Appendix B. First, some comments are in order.

**Mild initial conditions.** Theorem 1 holds under benign initial conditions. In particular, condition (i) requires that the initial distance is only a constant smaller than 1. Condition (ii) ensures that the

initial representation is well-conditioned with appropriate scaling, and (*iii*) guarantees the initial head is not too large. The last two conditions can be easily achieved by normalizing the inputs.

**Generalization without convergence in terms of the global loss.** When each client accesses its population loss as in Theorem 1, the global objective (5) becomes:

$$\min_{\mathbf{B} \in \mathbb{R}^{d \times k}, \mathbf{w} \in \mathbb{R}^k} \frac{1}{M} \sum_{i=1}^{M} \|\mathbf{B}\mathbf{w} - \mathbf{B}_* \mathbf{w}_{*,i}\|^2 \tag{6}$$

However, Theorem 1 does not imply that FedAvg solves (6). In fact, our simulations in Section 5 show that it does not even reach a stationary point of (6). This is consistent with prior works that have noticed the "objective inconsistency" phenomenon of FedAvg: it solves an unknown objective distinct from the global objective due to the fact that after multiple local updates, local gradients are no longer unbiased estimates of gradients of (6) [9]. Nevertheless, our results show that FedAvg is able to learn a generalizable model *even when it does not optimize the global loss in data heterogeneous settings*.

**Multiple local updates critically harness client diversity, whereas Distributed GD (D-GD) does not learn the representation.** Key to the proof of Theorem 1 is that the locally-updated heads become *diverse*, meaning that they cover all directions in $\mathbb{R}^k$, with greater diversity corresponding to more even covering in all directions. We will show in Section 4 that the locally-updated heads become roughly as diverse as the ground-truth heads, and this causes the representation to move towards the ground-truth at rate depending on the diversity level. Theorem 1 reflects this: the convergence rate improves with the diversity metric $\mu/L$. In this way FedAvg *exploits* data heterogeneity to learn the representation, as more diverse $\{\mathbf{w}_{*,i}\}_{i \in [M]}$ implies more heterogeneous data. Moreover, since $\tau$ also appears in the denominator of the communication round complexity, additional local updates improve the convergence rate up to $\tau = O(\alpha^{-2})$, which is the limit imposed due to the upper bound on $\alpha$.

Importantly, head diversity only benefits the global representation update if $\tau \geq 2$. We formally prove that D-GD (equivalent to FedAvg with $\tau = 1$ and $m = M$) cannot recover $\mathrm{col}(\mathbf{B}_*)$ in the following result.

**Proposition 1** (Distributed GD lower bound)**.** *Suppose we are in the setting described in Section 3 and $d > k > 1$. Then for any set of ground-truth heads $\{\mathbf{w}_{*,i}\}_{i=1}^M$, full-rank initialization $\mathbf{B}_0 \in \mathbb{R}^{d \times k}$, initial distance $\delta_0 \in (0, 1/2]$, step size $\alpha > 0$, and number of rounds $T$, there exists $\mathbf{B}_* \in \mathcal{O}^{d \times k}$ satisfying $\mathrm{dist}(\mathbf{B}_0, \mathbf{B}_*) = \delta_0$ and $\mathrm{dist}(\mathbf{B}_T^{D\text{-}GD}, \mathbf{B}_*) \geq 0.7\delta_0$, where $\mathbf{B}_T^{D\text{-}GD} \equiv \mathbf{B}_T^{D\text{-}GD}(\mathbf{B}_0, \mathbf{B}_*, \{\mathbf{w}_{*,i}\}_{i=1}^M, \alpha)$ is the result of D-GD with step size $\alpha$ and initialization $\mathbf{B}_0$ in the setting with ground-truth representation $\mathbf{B}_*$ and ground-truth heads $\{\mathbf{w}_{*,i}\}_{i=1}^M$.*

Proposition 1 shows that for any choice of $\delta_0 \in (0, 1/2]$, non-degenerate initialization $\mathbf{B}_0$, and ground-truth heads, there exists a $\mathbf{B}_*$ whose column space is $\delta_0$-close to $\mathrm{col}(\mathbf{B}_0)$, yet is at least $0.7\delta_0$-far from the representation learned by D-GD in the setting with $\mathbf{B}_*$ as ground-truth. Therefore, even allowing for a strong initialization, D-GD cannot guarantee to recover the ground-truth representation. This negative result combined with our previous results suggest that even if we had an infinite communication budget, it would still be advantageous to execute multiple local updates between communication rounds in order to achieve better generalization through representation learning.

## 4 Intuitions and Proof Sketch

Next we highlight the key ideas behind the importance of local updates and why FedAvg learns $\mathrm{col}(\mathbf{B}_*)$, while D-GD fails to achieve this goal.

**Global update $\mathbf{B}_{t+1}$.** To build intuition for why FedAvg can learn $\mathrm{col}(\mathbf{B}_*)$, we examine the global update of the representation in the full participation case ($m = M$):

$$\mathbf{B}_{t+1} = \mathbf{B}_t \underbrace{\left[ \frac{1}{M} \sum_{i=1}^{M} \prod_{s=0}^{\tau-1} (\mathbf{I}_k - \alpha \mathbf{w}_{t,i,s} \mathbf{w}_{t,i,s}^\top) \right]}_{\text{prior weight}} + \mathbf{B}_* \underbrace{\left[ \frac{\alpha}{M} \sum_{i=1}^{M} \mathbf{w}_{*,i} \sum_{s=0}^{\tau-1} \mathbf{w}_{t,i,s}^\top \prod_{r=s+1}^{\tau-1} (\mathbf{I}_k - \alpha \mathbf{w}_{t,i,r} \mathbf{w}_{t,i,r}^\top) \right]}_{\text{signal weight}}$$

Notice that $\mathbf{B}_{t+1}$ is a mixture of $\mathbf{B}_t$ and $\mathbf{B}_*$ with weight matrices in $\mathbb{R}^{k \times k}$. We aim to show that

(I) the 'prior weight' on $\mathbf{B}_t$ has spectral norm strictly less than 1, and

(II) the 'signal weight' on $\mathbf{B}_*$ adds energy from $\mathrm{col}(\mathbf{B}_*)$ to $\mathbf{B}_{t+1}$ so that $\sigma_{\min}(\mathbf{B}_{t+1}) \approx \sigma_{\min}(\mathbf{B}_t)$.

These two conditions imply that the contribution from $\mathrm{col}(\mathbf{B}_t)$ in $\mathrm{col}(\mathbf{B}_{t+1})$ contracts, while energy from $\mathrm{col}(\mathbf{B}_*)$ replaces the lost energy from $\mathrm{col}(\mathbf{B}_t)$. Hence, $\mathrm{col}(\mathbf{B}_{t+1})$ moves to $\mathrm{col}(\mathbf{B}_*)$ in all $k$ directions.

**The role of head diversity and multiple local updates.** To show (I) and (II), it is imperative to use the diversity of the locally-updated heads when $\tau \geq 2$. First consider (I). Notice that for each $i$, $\prod_{s=0}^{\tau-1}(\mathbf{I}_k - \alpha \mathbf{w}_{t,i,s}\mathbf{w}_{t,i,s}^\top)$ has singular values at most 1, and strictly less than 1 corresponding to directions spanned by $\{\mathbf{w}_{t,i,s}\}_{s\in[\tau-1]}$. Thus, the maximum singular value of the average of these matrices should be strictly less than 1 as long as $\{\mathbf{w}_{t,i,s}\}_{s\in[\tau-1],i\in[M]}$ spans $\mathbb{R}^k$, i.e. the locally-updated heads are diverse. Similarly, the signal weight is rank-$k$ if the locally-updated heads span $\mathbb{R}^k$, which leads to (II) as discussed below. In contrast, if $\tau = 1$, then the global update of the representation does *not* leverage head diversity, as it is only a function of the global head and the average ground-truth head: $\mathbf{B}_{t+1} = \mathbf{B}_t(\mathbf{I}_k - \alpha \mathbf{w}_t \mathbf{w}_t^\top) + \alpha \mathbf{B}_* \bar{\mathbf{w}}_* \mathbf{w}_t^\top$ in this case. As a result, $\mathrm{col}(\mathbf{B}_{t+1})$ can only improve in one direction, so D-GD ultimately fails to learn $\mathrm{col}(\mathbf{B}_*)$ (see Proposition 1).

**Achieving head diversity: the necessity of controlling $\mathbf{I}_k - \alpha \mathbf{B}_t^\top \mathbf{B}_t$.** We have discussed the intuition for why head diversity implies (I) and (II) for FedAvg. Next, we investigate why the heads become diverse. Let us examine client $i$'s first local update for the head at round $t$:

$$\mathbf{w}_{t,i,1} = (\mathbf{I}_k - \alpha \mathbf{B}_t^\top \mathbf{B}_t)\mathbf{w}_t + \alpha \mathbf{B}_t^\top \mathbf{B}_* \mathbf{w}_{*,i}$$

From this equation we see that if $\boldsymbol{\Delta}_t := \mathbf{I}_k - \alpha \mathbf{B}_t^\top \mathbf{B}_t \approx \mathbf{0}$ and $\|\mathbf{w}_t\|$ is bounded, then $\mathbf{w}_{t,i,1} \approx \alpha \mathbf{B}_t^\top \mathbf{B}_* \mathbf{w}_{*,i}$. If this approximation holds, then $\{\mathbf{w}_{t,i,1}\}_{i\in[M]}$ inherits the diversity of $\{\mathbf{w}_{*,i}\}_{i\in[M]}$, which is indeed diverse due to Assumption 2, meaning that the local heads are diverse after just one local update. Moreover, it can be shown that if $\boldsymbol{\Delta}_t \approx \mathbf{0}$ and the heads become diverse after one local update, then they remain diverse for all local updates due to the observation that each $\mathbf{B}_{t,i,s}$ changes slowly over $s$. Note that in addition to implying local head diversity, $\boldsymbol{\Delta}_t \approx \mathbf{0}$ for all $t$ implies $\sigma_{\min}(\mathbf{B}_t) \approx \sigma_{\min}(\mathbf{B}_{t+1}) \approx \frac{1}{\sqrt{\alpha}}$, which directly ensures (II). Thus we aim to show $\boldsymbol{\Delta}_t \approx \mathbf{0}$ for all communication rounds, i.e. $\mathbf{B}_t$ remains close to a scaled orthonormal matrix.

However, it is surprising why $\|\boldsymbol{\Delta}_t\|$ remains small: $\mathbf{B}_{t+1}$ is the average of nonlinearly locally-updated representations, and the local updates could 'overfit' by adding more energy to some columns than others, and/or lead to cancellation when summed, so it is not intuitive why $\sqrt{\alpha}\mathbf{B}_t$ remains almost orthonormal. Nor does the expression above for $\mathbf{B}_{t+1}$ provide any clarity on this. Nevertheless, through a careful induction we show that $\boldsymbol{\Delta}_t$ indeed stays close to zero since the local heads converge quickly and the projection of the local representation gradient onto $\mathrm{col}(\mathbf{B}_t)$ is exponentially decaying.

**Inductive argument.** While the above intuitions seem to simplify the behavior of FedAvg, showing that they all hold simultaneously is not at all obvious. To study this, we are inspired by recent work [44] that developed an inductive argument for representation learning in the context of gradient based-meta-learning. To formalize our intuition discussed previously, in our proof we need to show that (i) the learned representation does not overfit to each client's loss despite *many local updates* and simultaneously the heads quickly become diverse, and (ii) the update at the global server preserves the learned representation despite *averaging* many nonlinearly perturbed representations gathered from clients after local updates. To address these challenges, we construct a pair of intertwined inductive hypotheses over time, one for tracking the effect of local updates, and another for tracking the global averaging. Each inductive hypothesis (local and global) itself consists of several hypotheses (in effect, a nested induction) that evolve within communication rounds.

*Local induction.* The proof leverages the following local inductive hypotheses for every $t, i$:

1. $A_{1,t,i}(s) := \{\|\mathbf{w}_{t,i,s'} - \alpha \mathbf{B}_{t,i,s'-1}^\top \mathbf{B}_* \mathbf{w}_{*,i}\|_2 = c_1 \alpha^{2.5} \tau L_{\max}^3 \kappa_{\max}^2 E_0^{-1} \ \forall s' \in \{1,\ldots,s\}\}$

2. $A_{2,t,i}(s) := \{\|\mathbf{w}_{t,i,s'}\|_2 \leq c_2 \sqrt{\alpha} L_{\max} \quad \forall s' \in \{1,\ldots,s\}\}$

3. $A_{3,t,i}(s) := \{\|\mathbf{I}_k - \alpha \mathbf{B}_{t,i,s'}^\top \mathbf{B}_{t,i,s'}\|_2 = c_3 \alpha^2 L_{\max}^2 \kappa_{\max}^2 E_0^{-1} \quad \forall s' \in \{1,\ldots,s\}\}$

4. $A_{4,t,i}(s) := \{\mathrm{dist}(\mathbf{B}_{t,i,s'}, \mathbf{B}_*) \leq c_4 \, \mathrm{dist}(\mathbf{B}_t, \mathbf{B}_*) \quad \forall s' \in \{1,\ldots,s\}\}$

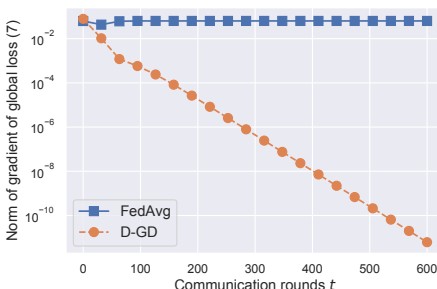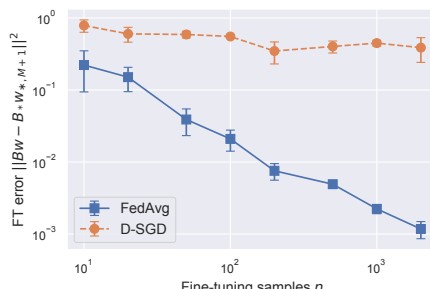

Figure 3: (Left) D-GD converges to a stationary point of the global objective (6), unlike FedAvg, yet (Right) FedAvg achieves smaller error after fine-tuning with various numbers of samples.

The local induction tracks the effect of updates at each client node: At the end of $\tau$ local updates, $A_{1,t,i}(\tau)$ captures the diversity of the local heads, $A_{2,t,i}(\tau)$ ensures that the heads remain uniformly bounded, $A_{3,t,i}(\tau)$ shows that the locally adapted representations stay close to a scaled orthonormal matrix, and $A_{4,t,i}(\tau)$ shows that the locally adapted representations do not diverge too quickly from the ground-truth. The second set of inductions below controls the global behavior.

*Global induction.* The global induction utilizes a similar set of inductive hypotheses.

1. $A_1(t) \coloneqq \{\|\mathbf{w}_{t'} - \alpha(\mathbf{I}_k + \boldsymbol{\Delta}_{t'})\mathbf{B}_{t'}^\top \mathbf{B}_* \bar{\mathbf{w}}_{*,t'}\|_2 = c_1'\alpha^{2.5}\tau L_{\max}^3 \quad \forall t' \in \{1,\ldots,t\}\}$

2. $A_2(t) \coloneqq \{\|\mathbf{w}_{t'}\|_2 \le c_2'\sqrt{\alpha}L_{\max} \quad \forall t' \in \{1,\ldots,t\}\}$

3. $A_3(t) \coloneqq \{\|\boldsymbol{\Delta}_{t'}\|_2 = c_3'\alpha^2\tau L_{\max}^2\kappa_{\max}^2 E_0^{-1} \quad \forall t' \in \{1,\ldots,t\}\}$

4. $A_4(t) \coloneqq \{\|\mathbf{B}_{*,\perp}^\top \mathbf{B}_{t'}\|_2 \le (1 - c_4'\alpha^2\tau\mu^2 E_0)\|\mathbf{B}_{*,\perp}^\top \mathbf{B}_{t'-1}\|_2 \quad \forall t' \in \{1,\ldots,t\}\}$

5. $A_5(t) \coloneqq \{\mathrm{dist}(\mathbf{B}_t, \mathbf{B}_*) \le (1 - c_5'\alpha^2\tau\mu^2 E_0)^{t-1} \quad \forall t' \in \{1,\ldots,t\}\}$

Hypotheses $A_1(t)$, $A_2(t)$ and $A_3(t)$ are analogous to $A_{1,t,i}(s)$, $A_{2,t,i}(s)$ and $A_{3,t,i}(s)$, respectively. $A_4(t)$ shows that the energy of $\mathrm{col}(\mathbf{B}_t)$ that is orthogonal to the ground-truth subspace is contracting, and $A_5(t)$ finally shows that the principal angle distance between the learned and ground-truth representations is exponentially decreasing. Our main claim follows from $A_5(T)$. However, proving this result requires showing that all the above local and global hypotheses hold for all times $t \ge 1$, as these hypotheses are heavily coupled. As mentioned previously, the most difficult challenge is controlling $\|\boldsymbol{\Delta}_t\|$ ($A_3(t)$) despite many local updates, and doing so requires leveraging both local and global properties. The details of this local-global induction argument are in Appendix B.

## 5 Experiments

In this section, we conduct experiments to (I) verify our theoretical results in the linear setting and (II) determine whether our established insights generalize to deep neural networks. Notably, demonstrating the competitive performance of FedAvg plus fine-tuning for personalized FL is *not* a goal of this section, as this is evident from prior experiments [10–12, 33]. Rather, to achieve (II) we test whether FedAvg learns effective representations when trained with neural networks in heterogeneous data settings via three popular benchmarks for evaluating the quality of learned representations. Since our main claim is that local updates are key to representation learning, we use D-(S)GD as our baseline in all experiments.

### 5.1 Multi-task linear regression

We first experiment with the regression setting from our theory. We randomly generate $\mathbf{B}_* \in \mathbb{R}^{d \times k}$ and $\{\mathbf{w}_{*,i}\}_{i \in [M]}$ by sampling each element i.i.d. from the normal distribution, where $d = 100$, $k = 5$ and $M = 40$, and then orthogonalizing $\mathbf{B}_*$. Then we run FedAvg with $\tau = 2$ local updates and D-GD, both sampling $m = M$ clients per round. We have seen in Figure 1 that the principal angle distance between the representation learned by FedAvg and the ground-truth representation linearly converges to zero, whereas D-GD does not learn the ground-truth representation. Conversely, Figure 3 (left) tracks the gradient of the global loss (6) and shows that D-GD linearly converges to stationary point of (6), while FedAvg does not converge to one at all. Although D-GD optimizes

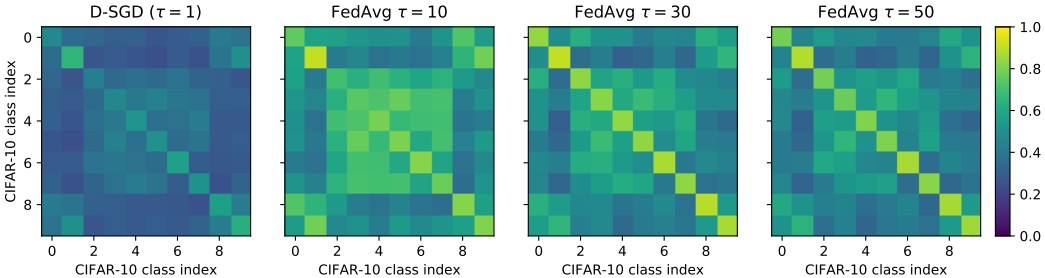

Figure 4: Average cosine similarity for features learned by D-SGD and FedAvg with varying numbers of local updates on a heterogeneous partition of CIFAR-10.

the global loss, it does not generalize as well as FedAvg to new clients as demonstrated by Figure 3 (right). Here, we fine-tune the models learned by FedAvg and D-GD on a new client with $n$ samples generated by $\mathbf{x}_{M+1} \sim \mathcal{N}(\mathbf{0}, \mathbf{I}_d)$, $\zeta_{M+1} \sim \mathcal{N}(0, 0.01)$, and $y_{M+1} = \langle \mathbf{B}_* \mathbf{w}_{*, M+1}, \mathbf{x}_{M+1} \rangle + \zeta_{M+1}$. We fine-tune using GD for $\tau' = 200$ iterations with batch size $b = n$, and plot the final error $\| \mathbf{B}_{T, M+1, \tau'} \mathbf{w}_{T, M+1, \tau'} - \mathbf{B}_* \mathbf{w}_{*, M+1} \|^2$. Both plots are generated by averaging 10 runs.

## 5.2 Image classification with neural networks

Next we evaluate FedAvg's representation learning ability on nonlinear neural networks. For fair comparison, in every experiment all methods make the same total amount of local updates during the course of training (e.g. D-SGD is trained for $50\times$ more rounds than FedAvg with $\tau = 50$).

**Datasets and models.** We use the image classification datasets CIFAR-10 and CIFAR-100 [57], which consist of 10 and 100 classes of RGB images, respectively. We use a convolutional neural network (CNN) with three convolutional blocks followed by a three-layer multi-layer perceptron, with each convolutional block consisting of two convolutional layers and a max pooling layer.

**Cosine similarity of features.** A desirable property of representations for downstream classification tasks is that features of examples from the same class are similar to each other, while features of examples from different classes are dissimilar [58]. In Figure 4 we examine whether the representations learned by FedAvg satisfy this property. Here we have trained FedAvg with varying $\tau$ and D-SGD (FedAvg with $\tau = 1$) on CIFAR-10. Image classes are heterogeneously allocated to $M = 100$ clients according to the Dirichlet distribution with parameter 0.6 as in [59]. Each subplot is a 10x10 matrix whose $(i, j)$-th element gives the average cosine similarity between features of images from the $i$-th and $j$-th classes learned by the corresponding model. Ideally, diagonal elements are close to 1 (high similarity) and off-diagonal elements are close to 0 (low similarity). Figure 4 shows that FedAvg indeed learns features with high intra-class similarity and low inter-class similarity, with representation quality improving with more local updates between communications. Meanwhile, D-SGD does not learn such features. The leftmost subplot shows that all of the features learned by D-SGD are dissimilar, regardless of whether two images belong to the same class.

**Fine-tuning performance.** We evaluate the generalization ability of the representations learned by FedAvg to new classes and also new datasets. An effective representation identifies universally important features, so it should generalize to new data, with perhaps a small amount of fine-tuning needed to learn a new mapping from feature space to label space. The transfer learning performance of fine-tuned models is a popular metric for evaluating the quality of learned representations [60, 61]. We first study how models trained by FedAvg and D-SGD generalize to unseen classes from the same dataset. To do so, we train models on heterogeneous partitions of CIFAR-100 using both FedAvg with $\tau = 50$ as well as D-SGD. In the left plot of Figure 5, we illustrate the case that models are trained on 80 clients each with 500 total images from $C$ classes sub-selected from 80 classes of CIFAR-100, and tested on new clients with images from the remaining 20 classes of CIFAR-100. We fine-tune the trained models on the new clients with 10 epochs of SGD, with varying numbers of samples per epoch as listed on the x-axis, before testing. Next, we investigate how well models trained by FedAvg and D-SGD generalize to an unseen dataset. In the right plot of Figure 5, we train models with $C$ classes/client from CIFAR-100, then test on new clients with samples drawn from CIFAR-10 (a different dataset, but with presumably similar "basic" features). Specifically, for these new clients, we fine-tune for 10 epochs as previously, then test the post-fine-tuned models on the test

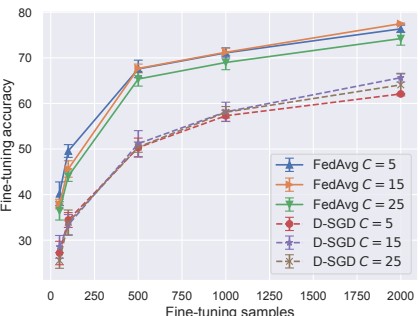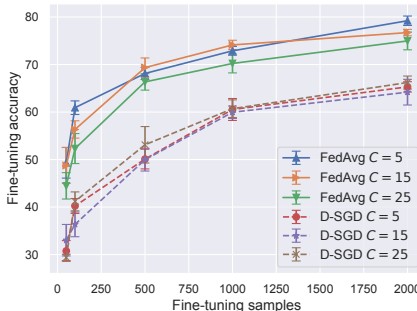

Figure 5: Average fine-tuning accuracies on new clients for models trained by FedAvg and D-SGD. (Left) Models trained on 80 classes from CIFAR-100 (with $C$ classes/client) and fine-tuned on new clients from 20 new classes from CIFAR-100. (Right) Models trained on CIFAR-100 with $C$ classes/client and fine-tuned on new clients from CIFAR-10 (10 classes/client). For FedAvg, $\tau = 50$ in all cases, and error bars give standard deviations over five trials with five new clients tested per trial.

data for each client. In both left and right plots, we observe that FedAvg significantly outperforms D-SGD, indicating that FedAvg has learned a representation that generalizes better to new classes.

## 6 Conclusion

We showed that FedAvg learns the ground-truth representation in the multi-task linear regression setting. To our knowledge, this is the first theoretical study showing FedAvg learns an effective representation in any setting. Our analysis reveals that multiple local updates are critical to FedAvg's representation learning ability, which is supported empirically on both linear and nonlinear models. These experimental results suggest future work can extend our findings to more complex settings.

## Acknowledgements

This research is supported in part by NSF Grants 2127697, 2019844, 2107037, and 2112471, ARO Grant W911NF2110226, ONR Grant N00014-19-1-2566, the Machine Learning Lab (MLL) at UT Austin, and the Wireless Networking and Communications Group (WNCG) Industrial Affiliates Program.

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
