# A   Additional Related Work

In this section we provide further discussion of the related works.

**Convergence of FedAvg.** The convergence of FedAvg, also known as Local SGD, has been the subject of intense study in recent years due to the algorithm's effectiveness combined with the difficulties of analyzing it. In homogeneous data settings, local updates are easier to reconcile with solving the global objective, allowing much progress to be made in understanding convergence rates in this case [2–4, 62–66]. In the heterogeneous case multiple works have shown that FedAvg with fixed learning rate may *not* solve the global objective because the local updates induce a non-vanishing bias by drifting towards local solutions, even with full gradient steps and and strongly convex objectives [5–9, 16, 20, 67, 68]. As a remedy, several papers have analyzed FedAvg with learning rate that decays over communication rounds, and have shown that this approach indeed reaches a stationary point of the global objective, but at sublinear rates [5, 14–17] that can be strictly slower than the convergence rates of D-SGD [5, 18]. Sublinear convergence rates to stationary points of the global objective have also been shown for gossip algorithms that generalize FedAvg when operating on time-varying communication graphs [69–71], but these rates have exponential dependence on $M$ and $\tau$.

Another line of work has shown that in overparameterized settings with strongly convex losses, FedAvg achieves linear convergence to the global optimum [15, 17]. We consider more challenging nonconvex losses, and our setting is not overparameterized in the same sense (in these works overparameterized implies that the model class contains a single model that achieves zero loss for all clients). Lastly, like our work, [72] empirically observed that FedAvg learns strong representations, but relative to local-only training (i.e., without any communication), thus they did not study the role of local updates between communication rounds in learning representations, nor did they provide theoretical analysis.

**Multi-task representation learning.** Multiple works have studied the multi-task linear representation learning setting [39] in recent years. [41] and [42] give statistical rates for a method-of-moments estimator for learning the representation and [45] analyze a projection and eigen-weighting based algorithm designed for the case in which ground-truth representation is unknown. Other works have studied alternating minimization procedures for learning $\mathrm{col}(\mathbf{B}_*)$ in the context of meta-learning [43], federated learning [11], and differentially private optimization [46]. However, these methods require a unique head for each client, which greatly simplifies the analysis since head diversity is guaranteed prior to local updates and is not applicable to some cross-device FL settings which cannot tolerate stateful clients. Outside of the multi-task linear regression setting, [73] and [74] have demonstrated the necessity of task diversity to learning generalizable representations from a learning theoretic perspective, and [40] considered the statistical rates of representation learning by solving an ERM with unique heads per task.

## B Proof of Theorem Main Results

### B.1 Proof of Theorem 1

In this section we provide the proof of Theorem 1. We make use of the notations in Table 1.

Table 1: Notations.

| Notation | Definition |
|---|---|
| $\mu$ | $\lambda_{\min}^{0.5}\left(\frac{1}{M}\sum_{i=1}^{M}(\mathbf{w}_{*,i}-\bar{\mathbf{w}}_*)(\mathbf{w}_{*,i}-\bar{\mathbf{w}}_*)^\top\right)$ |
| $L_{\max}$ | $\max_{i\in[M]}\|\mathbf{w}_{*,i}\|_2$, note that $L_{\max}:=L\sqrt{k}$ where $L$ is defined in Assumption 1 |
| $\kappa_{\max}$ | $L_{\max}/\mu$, note that $\kappa_{\max}:=\kappa\sqrt{k}$ where $\kappa$ is defined in Assumption 2 |
| $\bar{\mathbf{w}}_*$ | $\frac{1}{M}\sum_{i=1}^{M}\mathbf{w}_{*,i}$ |
| $\bar{\mathbf{w}}_{*,t}$ | $\frac{1}{m}\sum_{i\in\mathcal{I}_t}\mathbf{w}_{*,i}$ |
| $\mathbf{B}_{t,i,s},\mathbf{w}_{t,i,s}$ | the results of $s$ local updates of the global model at round $t$ by the $i$-th client |
| $\mathbf{B}_{t,i,0},\mathbf{w}_{t,i,0}$ | $\mathbf{B}_t,\mathbf{w}_t$, respectively |
| $\mathbf{e}_{t,i,s}$ | $\mathbf{B}_{t,i,s}\mathbf{w}_{t,i,s}-\mathbf{B}_*\mathbf{w}_{*,i}$, i.e. product error for $s$-th local update for task $i$, round $t$ |
| $\mathbf{G}_{t,i,s}$ | $(\mathbf{B}_{t,i,s+1}-\mathbf{B}_{t,i,s})/\alpha$, such that $\mathbf{B}_{t,i,s+1}=\mathbf{B}_{t,i,s}-\alpha\mathbf{G}_{t,i,s}$ |
| $\mathbf{G}_t$ | $(\mathbf{B}_{t+1}-\mathbf{B}_t)/\alpha$, such that $\mathbf{B}_{t+1}=\mathbf{B}_t-\alpha\mathbf{G}_t$ |
| $\mathbf{\Delta}_t$ | $\mathbf{I}_k-\alpha\mathbf{B}_t^\top\mathbf{B}_t$ |
| $\bar{\mathbf{\Delta}}_t$ | $\mathbf{I}_d-\alpha\mathbf{B}_t\mathbf{B}_t^\top$ |
| $\text{dist}_t$ | $\text{dist}(\mathbf{B}_t,\mathbf{B}_*)$ |
| $\delta_0$ | $\text{dist}_0$ |
| $E_0$ | $1-\text{dist}_0^2$ |
| $\text{col}(\mathbf{B}),\text{col}(\mathbf{B})^\perp$ | column space of $\mathbf{B}$, orthogonal complement to column space of $\mathbf{B}$, respectively |

Here the local updates are given by

$$\mathbf{B}_{t,i,s+1}=\mathbf{B}_{t,i,s}-\alpha(\mathbf{B}_{t,i,s}\mathbf{w}_{t,i,s}-\mathbf{B}_*\mathbf{w}_{*,i})\mathbf{w}_{t,i,s}^\top$$

$$\mathbf{w}_{t,i,s+1}=\mathbf{w}_{t,i,s}-\alpha\mathbf{B}_{t,i,s}^\top(\mathbf{B}_{t,i,s}\mathbf{w}_{t,i,s}-\mathbf{B}_*\mathbf{w}_{*,i})$$

$$=\mathbf{\Delta}_{t,i,s}\mathbf{w}_{t,i,s}+\alpha\mathbf{B}_{t,i,s}^\top\mathbf{B}_*\mathbf{w}_{*,i}$$

and the global updates are given by

$$\mathbf{B}_{t+1}=\frac{1}{m}\sum_{i\in\mathcal{I}_t}\mathbf{B}_{t,i,\tau}$$

$$\mathbf{w}_{t+1}=\frac{1}{m}\sum_{i\in\mathcal{I}_t}\mathbf{w}_{t,i,\tau}.$$

First we control the ground-truth heads sampled on each round.

**Lemma 1.** *Suppose* $m\geq\min(M,20((\gamma/L)^2+(H/L)^4)(\alpha L\sqrt{k})^{-4}\log(kT))$. *Then the event*

$$A_0:=\left\{\left\|\frac{1}{m}\sum_{i\in\mathcal{I}_t}\mathbf{w}_{*,i}-\bar{\mathbf{w}}_*\right\|\leq 4\alpha^2 L_{\max}^3,\right.$$

$$\left.\left\|\frac{1}{m}\sum_{i\in\mathcal{I}_t}\mathbf{w}_{*,i}\mathbf{w}_{*,i}^\top-\frac{1}{M}\sum_{i'=1}^{M}\mathbf{w}_{*,i'}\mathbf{w}_{*,i'}^\top\right\|\leq 4\alpha^2 L_{\max}^4\quad\forall t\in[T]\right\}$$

*occurs with probability at least* $1-4(kT)^{-99}$, *where* $L_{\max}:=L\sqrt{k}$.

*Proof.* If $m=M$ then $A_0$ holds almost surely. Otherwise, first let $\mathbf{W}_{*,i}:=\text{diag}(\mathbf{w}_{*,i})\in\mathbf{R}^{k\times k}$, and let $\bar{\mathbf{W}}_*:=\frac{1}{M}\sum_{i=1}^{M}\text{diag}(\mathbf{w}_{*,i})$. For any $t\in[T]$, $\{\mathbf{W}_{*,i}\}_{i\in\mathcal{I}_t}$ is a set of Hermitian matrices sampled uniformly without replacement from $\{\mathbf{W}_{*,i}\}_{i\in[M]}$, $\|\mathbf{W}_{*,i}\|\leq L\sqrt{k}$ almost surely, and

$\left\| \frac{1}{M} \sum_{i=1}^{M} (\mathbf{W}_{*,i} - \bar{\mathbf{W}}_*)^2 \right\| \leq \frac{1}{M} \sum_{i=1}^{M} \| \mathbf{w}_{*,i} - \bar{\mathbf{w}}_* \|^2 = \gamma^2 k$ by the triangle and Cauchy-Schwarz inequalities and Definition 1. Thus, we can apply Theorem 1 in [75] to obtain

$$\mathbb{P}\left( \left\| \sum_{i \in \mathcal{I}_t} \mathbf{W}_{*,i} - \bar{\mathbf{W}}_* \right\| > t \right) \leq 2k \exp(\tfrac{-t^2}{4m\gamma^2 k}) \tag{7}$$

as long as $t \leq 2m\gamma^2 \sqrt{k}/L$. Choose $t = 4m\alpha^2 L^3 k^{1.5}$. Note that indeed $t \leq 2m\gamma^2 \sqrt{k}$ since $\gamma^2 = \frac{1}{k} \sum_{l=1}^{k} \frac{1}{M} \sum_{i=1}^{M} \mathbf{u}_l^\top (\mathbf{w}_{*,i} - \bar{\mathbf{w}}_*)(\mathbf{w}_{*,i} - \bar{\mathbf{w}}_*)^\top \mathbf{u}_l \geq \mu^2$, where $\mathbf{u}_l$ is the $l$-th standard basis vector, and $\alpha \leq \mu^2/(2L^4 k^2)$. Thus we obtain

$$\mathbb{P}\left( \left\| \sum_{i \in \mathcal{I}_t} \mathbf{W}_{*,i} - \bar{\mathbf{W}}_* \right\| > 4m\alpha^2 L^3 k^{1.5} \right) \leq 2k \exp\left( \tfrac{-4m\alpha^4 L^6 k^2}{\gamma^2} \right)$$

$$\implies \mathbb{P}\left( \left\| \frac{1}{m} \sum_{i \in \mathcal{I}_t} \mathbf{W}_{*,i} - \bar{\mathbf{W}}_* \right\| > 4\alpha^2 L^3 k^{1.5} \right) \leq 2k \exp\left( -100 \log(kT) \right)$$

since $m \geq 20(\frac{\gamma^2}{L^2})(\alpha^4 L^4 k^2)^{-1} \log(kT)$. An analogous argument, without needing to lift the matrices to higher dimensions, yields

$$\mathbb{P}\left( \left\| \frac{1}{m} \sum_{i \in \mathcal{I}_t} \left( \mathbf{w}_{*,i} \mathbf{w}_{*,i}^\top - \frac{1}{M} \sum_{i'=1}^{M} \mathbf{w}_{*,i'} \mathbf{w}_{*,i'}^\top \right) \right\| > 4\alpha^2 L^4 k^2 \right) \leq 2k \exp\left( -100 \log(kT) \right)$$

Union bounding, we obtain that $\left\| \frac{1}{m} \sum_{i \in \mathcal{I}_t} \mathbf{W}_{*,i} - \bar{\mathbf{W}}_* \right\| \leq 4\alpha^2 L^3 k^{1.5}$ and $\left\| \frac{1}{m} \sum_{i \in \mathcal{I}_t} \left( \mathbf{w}_{*,i} \mathbf{w}_{*,i}^\top - \frac{1}{M} \sum_{i'=1}^{M} \mathbf{w}_{*,i'} \mathbf{w}_{*,i'}^\top \right) \right\| \leq 4\alpha^2 L^4 k^2$ with probability at least $1 - 4k \exp(-100 \log(kT)) = 1 - 4k^{-99} T^{-100}$. Union bounding over all $t \in [T]$ completes the proof. $\qquad\square$

Next we state and prove the version of Theorem 1 with explicit constants. Note that the constants are not optimized.

**Theorem 2** (FedAvg Representation Learning). *Consider the case that each client takes gradient steps with respect to their population loss $f_i(\mathbf{B}, \mathbf{w}) := \frac{1}{2} \| \mathbf{B}\mathbf{w} - \mathbf{B}_* \mathbf{w}_{*,i} \|^2$ and all losses are weighted equally in the global objective. Suppose Assumptions 1 and 2 hold, the number of clients participating each round satisfies $m \geq \min(M, 20((\gamma/L)^2 + (H/L)^4)(\alpha L \sqrt{k})^{-4} \log(kT))$, and the initial parameters satisfy (i) $\delta_0 := \mathrm{dist}(\mathbf{B}_0, \mathbf{B}_*) \leq \sqrt{1 - E_0}$ for any $E_0 \in (0, 1]$, (ii) $\| \mathbf{I} - \alpha \mathbf{B}_0^\top \mathbf{B}_0 \|_2 \leq \alpha^2 \tau L^2 \kappa^2 k^2$ and (iii) $\| \mathbf{w}_0 \|_2 \leq \alpha^{2.5} \tau L^3 k^{1.5}$. Choose step size $\alpha \leq \frac{1 - \delta_0}{4800 \sqrt{\tau} L \kappa^2 k^{1.5}}$. Then for any $\epsilon \in (0, 1)$, the distance of the representation learned by FedAvg with $\tau \geq 2$ local updates satisfies $\mathrm{dist}(\mathbf{B}_T, \mathbf{B}_*) < \epsilon$ after at most*

$$T \leq \frac{25}{\alpha^2 \tau \mu^2 E_0} \log(1/\epsilon)$$

*communication rounds with probability at least $1 - 4(kT)^{-99}$.*

*Proof.* In this proof we use the notation $L_{\max} := L\sqrt{k}$ and $\kappa_{\max} := \kappa\sqrt{k}$. First we condition on the event $A_0$, which occurs with probability at least $1 - 4(kT)^{-99}$ by Lemma 1. Conditioned on this event, we will show that that the following two sets of inductive hypotheses hold for all $s \in [\tau]$, $i \in \mathcal{I}_t$, and $t \in [T]$. The first set of inductive hypotheses controls local behavior. We apply the below local induction in parallel for each client $i \in [M]$ at every communication round $t \geq 0$, starting from the base case $s = 1$.

1. $A_{1,t,i}(s) := \{ \| \mathbf{w}_{t,i,s'} - \alpha \mathbf{B}_{t,i,s'-1}^\top \mathbf{B}_* \mathbf{w}_{*,i} \|_2 \leq 4c_3 \alpha^{2.5} \tau L_{\max}^3 \kappa_{\max}^2 E_0^{-1} \quad \forall s' \in \{1, \ldots, s\} \}$

2. $A_{2,t,i}(s) := \{ \| \mathbf{w}_{t,i,s'} \|_2 \leq 2\alpha^{0.5} L_{\max} \quad \forall s' \in \{1, \ldots, s\} \}$

3. $A_{3,t,i}(s) := \{ \| \boldsymbol{\Delta}_{t,i,s'} \|_2 \leq 2c_3 \alpha^2 \tau L_{\max}^2 \kappa_{\max}^2 E_0^{-1} \quad \forall s' \in \{1, \ldots, s\} \}$

4. $A_{4,t,i}(s) := \{\mathrm{dist}(\mathbf{B}_{t,i,s'}, \mathbf{B}_*) \le 1.1\,\mathrm{dist}(\mathbf{B}_t, \mathbf{B}_*) \quad \forall s' \in \{1, \ldots, s\}\}$

The second set of inductions controls the global behavior, starting from $t = 1$ as the base case:

1. $A_1(t) := \{\|\mathbf{w}_{t'} - \alpha(\mathbf{I}_k + \boldsymbol{\Delta}_{t'})\mathbf{B}_{t'}^\top \mathbf{B}_* \bar{\mathbf{w}}_{*,t'}\|_2 \le 91\alpha^{2.5}\tau L_{\max}^3 \quad \forall t' \in \{1, \ldots, t\}\}$

2. $A_2(t) := \{\|\mathbf{w}_{t'}\|_2 \le 2\alpha^{0.5}L_{\max} \quad \forall t' \in \{1, \ldots, t\}\}$

3. $A_3(t) := \{\|\boldsymbol{\Delta}_{t'}\|_2 \le c_3\alpha^2 \tau L_{\max}^2 \kappa_{\max}^2 E_0^{-1} \quad \forall t' \in \{1, \ldots, t\}\}$

4. $A_4(t) := \{\|\mathbf{B}_{*,\perp}^\top \mathbf{B}_{t'}\|_2 \le (1 - 0.04\alpha^2 \tau \mu^2 E_0)\|\mathbf{B}_{*,\perp}^\top \mathbf{B}_{t'-1}\|_2 \quad \forall t' \in \{1, \ldots, t\}\}$

5. $A_5(t) := \{\mathrm{dist}_t \le (1 - 0.04\alpha^2 \tau \mu^2 E_0)^{t-1} \quad \forall t' \in \{1, \ldots, t\}\}$

where $c_3 = 4800$. Without loss of generality let $\alpha \le \frac{1-\delta_0}{c_3\sqrt{\tau}L_{\max}\kappa_{\max}^2}$. For ease of presentation we refer to $c_3$ symbolically rather than by its value throughout the proof.

The above inductions are applied in the following manner. First, the global initialization at $t = 0$ implies that the local inductive hypotheses hold after one local update (the base case). Then, by the local inductive argument, these conditions continue to hold for all subsequent local updates. This in turn implies that the global inductive hypotheses hold after the first global averaging step, i.e. $A_1(1)$, $A_2(1)$ and $A_3(1)$ hold. Next, the global hypotheses holding at $t = 1$ implies that the local inductions hold in their base case at $t = 1$ (after one local update, i.e. $s = 1$), which implies they continue to hold for all subsequent local updates. Again, this implies the global hypotheses hold at $t = 2$, which implies the base case for the local inductions at $t = 2$, and so on. In summary, the ordering of the inductions is:

$$\text{Initialization at } t{=}0 \implies \text{Local inductions at } t{=}0 \implies \text{Global inductions at } t{=}1$$
$$\implies \text{Local inductions at } t{=}1 \implies \ldots$$

We start by showing that the base case $s = 1$ holds for the local inductions. The proof is identical for all $i \in [M]$ and $t \ge 0$.

- **If $t = 0$: initial conditions $\implies A_{1,t,i}(1)$, else $A_2(t) \cap A_3(t) \implies A_{1,t,i}(1)$.**

  Note that at initialization, $\|\boldsymbol{\Delta}_0 \mathbf{w}_0\| \le \|\boldsymbol{\Delta}_0\|\|\mathbf{w}_0\| \le \alpha^{2.5}\tau L_{\max}^3 \kappa_{\max}^2 \le 4c_3\alpha^{2.5}\tau L_{\max}^3 \kappa_{\max}^2 E_0^{-1}$. Likewise, at arbitrary $t$, $\|\boldsymbol{\Delta}_t \mathbf{w}_t\| \le 4c_3\alpha^{2.5}\tau L_{\max}^3 \kappa_{\max}^2 E_0^{-1}$ due to $A_2(t)$ and $A_3(t)$. Thus, since $\mathbf{w}_{t,i,1} = \boldsymbol{\Delta}_t \mathbf{w}_t + \alpha \mathbf{B}_t^\top \mathbf{B}_* \mathbf{w}_{*,i}$, we have

  $$\|\mathbf{w}_{t,i,1} - \alpha\mathbf{B}_t^\top \mathbf{B}_* \mathbf{w}_{*,i}\| = \|\boldsymbol{\Delta}_t \mathbf{w}_t\| \le \|\boldsymbol{\Delta}_t\|\|\mathbf{w}_t\| \le 4c_3\alpha^{2.5}\tau L_{\max}^3 \kappa_{\max}^2 E_0^{-1}$$

  as desired (recall that $\mathbf{B}_{t,i,0} \equiv \mathbf{B}_t$).

- **If $t = 0$: initial conditions $\cap$ $A_{1,t,i}(1) \implies A_{2,t,i}(1)$, else $A_3(t) \cap A_{1,t,i}(1) \implies A_{2,t,i}(1)$.**

  For any $t \ge 0$, we have $\|\boldsymbol{\Delta}_t\| \le c_3\alpha^2 \tau L_{\max}^2 \kappa_{\max}^2 E_0^{-1}$ due to either the initialization ($t = 0$) or $A_3(t)$ ($t > 0$). This implies that $\|\mathbf{B}_t\| \le \sqrt{\frac{1+c_3\alpha^2 \tau L_{\max}^2 \kappa_{\max}^2 E_0^{-1}}{\alpha}} \le \frac{1.1}{\sqrt{\alpha}}$ since $\alpha$ is sufficiently small (noting that $\frac{(1-\delta)^2}{E_0} = \frac{(1-\delta)^2}{1-\delta^2} \le 1$. Now we use $A_{1,t,i}(1)$ and the triangle inequality to obtain:

  $$\begin{aligned}\|\mathbf{w}_{t,i,1}\| &\le \|\mathbf{w}_{t,i,1} - \alpha\mathbf{B}_t^\top \mathbf{B}_* \mathbf{w}_{*,i}\| + \|\alpha\mathbf{B}_t^\top \mathbf{B}_* \mathbf{w}_{*,i}\| \\ &\le 4c_3\alpha^{2.5}\tau L_{\max}^3 \kappa_{\max}^2 E_0^{-1} + \alpha\|\mathbf{B}_t\|\|\mathbf{w}_{*,i}\| \\ &\le 2\sqrt{\alpha}L_{\max}.\end{aligned}$$

  as desired.

- **If $t = 0$: initial conditions $\implies A_{3,t,i}(1)$, else $A_2(t) \cap A_3(t) \implies A_{3,t,i}(1)$.**

We have

$$\boldsymbol{\Delta}_{t,i,1} = \mathbf{I}_k - \alpha \mathbf{B}_{t,i,1}^\top \mathbf{B}_{t,i,1}$$
$$= \boldsymbol{\Delta}_t + \alpha^2 \mathbf{B}_t^\top (\mathbf{B}_t \mathbf{w}_t - \mathbf{B}_* \mathbf{w}_{*,i}) \mathbf{w}_t^\top$$
$$+ \alpha^2 \mathbf{B}_t^\top (\mathbf{B}_t \mathbf{w}_t - \mathbf{B}_* \mathbf{w}_{*,i}) \mathbf{w}_t^\top - \alpha^3 \mathbf{w}_t \mathbf{w}_t^\top \|\mathbf{B}_t \mathbf{w}_t - \mathbf{B}_* \mathbf{w}_{*,i}\|_2^2 \quad (8)$$

By the initial conditions and by inductive hypotheses $A_1(t)$ and $A_2(t)$, for any $t \geq 0$ we have $\|\mathbf{w}_t\|_2 \leq 2\sqrt{\alpha} L_{\max}$, $\|\boldsymbol{\Delta}_t\| \leq c_3 \alpha^2 \tau L_{\max}^2 \kappa_{\max}^2 E_0^{-1}$, and $\|\mathbf{B}_t\|_2 \leq \frac{1.1}{\sqrt{\alpha}}$. This implies $\|\mathbf{B}_t \mathbf{w}_t - \mathbf{B}_* \mathbf{w}_{*,i}\|_2 \leq \|\mathbf{B}_t\|\|\mathbf{w}_t\| + \|\mathbf{B}_* \mathbf{w}_{*,i}\| \leq 3.2 L_{\max}$. Therefore using (8), we obtain

$$\|\boldsymbol{\Delta}_{t,i,1}\|_2 \leq \|\boldsymbol{\Delta}_t\|_2 + 2\alpha^2 \|\mathbf{B}_t^\top (\mathbf{B}_t \mathbf{w}_t - \mathbf{B}_* \mathbf{w}_{*,i}) \mathbf{w}_t^\top\|_2 + \alpha^3 \|\mathbf{w}_t\|_2^2 \|\mathbf{B}_t \mathbf{w}_t - \mathbf{B}_* \mathbf{w}_{*,i}\|_2^2$$
$$\leq \|\boldsymbol{\Delta}_t\|_2 + 15\alpha^2 L_{\max}^2 + 41\alpha^4 L_{\max}^4$$
$$\leq 2 c_3 \alpha^2 \tau L_{\max}^2 \kappa_{\max}^2 E_0^{-1} \quad (9)$$

as desired.

- **If $t = 0$: initial conditions $\implies A_{4,t,i}(1)$, else $A_{3,t,i}(1) \cap A_2(t) \cap A_3(t) \implies A_{4,t,i}(1)$.**

  Note that

  $$\|\mathbf{B}_{*,\perp}^\top \mathbf{B}_{t,i,1}\|_2 = \|\mathbf{B}_{*,\perp}^\top \mathbf{B}_t (\mathbf{I}_k - \alpha \mathbf{w}_t \mathbf{w}_t^\top)\|_2 \leq \|\mathbf{B}_{*,\perp}^\top \mathbf{B}_t\|\|\mathbf{I}_k - \alpha \mathbf{w}_t \mathbf{w}_t^\top\|_2 \leq \|\mathbf{B}_{*,\perp}^\top \mathbf{B}_t\|_2$$

  as $\alpha \|\mathbf{w}_t\|^2 \leq 1$ by either the initialization (if $t = 0$) or $A_2(t)$ (if $t \geq 1$) and the choice of $\alpha$ sufficiently small. Thus, letting $\hat{\mathbf{B}}_{t,i,1} \mathbf{R}_{t,i,1} = \mathbf{B}_{t,i,1}$ denote the QR-decomposition of $\mathbf{B}_{t,i,1}$, we have

  $$\mathrm{dist}(\mathbf{B}_{t,i,1}, \mathbf{B}_*) = \|\mathbf{B}_{*,\perp}^\top \hat{\mathbf{B}}_{t,i,1}\|_2$$
  $$\leq \tfrac{1}{\sigma_{\min}(\mathbf{B}_{t,i,1})} \|\mathbf{B}_{*,\perp}^\top \mathbf{B}_{t,i,1}\|_2$$
  $$\leq \tfrac{1}{\sigma_{\min}(\mathbf{B}_{t,i,1})} \|\mathbf{B}_{*,\perp}^\top \mathbf{B}_t\|_2$$
  $$\leq \tfrac{\|\mathbf{B}_t\|}{\sigma_{\min}(\mathbf{B}_{t,i,1})} \mathrm{dist}(\mathbf{B}_t, \mathbf{B}_*)$$
  $$\leq \sqrt{\tfrac{1 + c_3 \alpha^2 \tau L_{\max}^2 \kappa_{\max}^2 E_0^{-1}}{1 - 2 c_3 \alpha^2 \tau L_{\max}^2 \kappa_{\max}^2 E_0^{-1}}} \, \mathrm{dist}_t \quad (10)$$
  $$\leq 1.1 \, \mathrm{dist}(\mathbf{B}_t, \mathbf{B}_*) \quad (11)$$

  where the (10) follows by $A_{3,t,i}(1)$ and either the initial condition on $\|\boldsymbol{\Delta}_t\|$ (if $t = 0$) or $A_3(t)$ (if $t > 0$), and (11) follows as $\alpha$ is sufficiently small.

Now we show that the global inductions hold at global round $t = 1$ following the local updates at round $t = 0$.

- **Initialization** $\cap \big( \cap_{i \in \mathcal{I}_0} A_{1,0,i}(\tau) \cap A_{2,0,i}(\tau) \cap A_{3,0,i}(\tau) \big) \implies A_1(1) \cap A_2(1) \cap A_4(1) \cap A_5(1)$.

  To show each of these hypotheses hold we can apply the proofs of Lemmas 6, 7 9 and 11 respectively, since they only rely on inductive hypotheses $A_{1,0,i}(\tau)$, $A_{2,0,i}(\tau)$ and $A_{3,0,i}(\tau)$ and appropriate scaling of $\|\mathbf{B}_0\|$ and $\|\mathbf{w}_0\|$, which is guaranteed by the initialization. In particular, the proof of these inductive hypotheses is identical for all $t \geq 1$.

- **Initialization** $\cap \big( \cap_{i \in \mathcal{I}_0} A_{2,0,i}(\tau) \cap A_{3,0,i}(\tau) \big) \implies A_2(1)$.

  In the proof of $A_2(t)$ for $t \geq 2$ (Lemma 9) we leverage the fact that $\mathbf{w}_{t-1}$ is close to a matrix times the average of the $\bar{\mathbf{w}}_{*,t-1}$. Our initialization cannot guarantee that this holds for $\mathbf{w}_0$. Instead, we show that $\|\boldsymbol{\Delta}_1\|$ may increase from $\|\boldsymbol{\Delta}_0\|$ at a large rate that would cause $\|\boldsymbol{\Delta}_t\|$ to blow up if continued indefinitely, but since it only grows at this rate for the first round, this is ok. In particular, let $\mathbf{G}_0 = \frac{1}{\alpha}(\mathbf{B}_0 - \mathbf{B}_1)$ such that $\mathbf{B}_1 = \mathbf{B}_0 - \alpha \mathbf{G}_0$. Then

  $$\boldsymbol{\Delta}_1 = \boldsymbol{\Delta}_0 + \alpha^2 \mathbf{B}_0^\top \mathbf{G}_0 + \alpha^2 \mathbf{G}_0^\top \mathbf{B}_0 - \alpha^3 \mathbf{G}_0^\top \mathbf{G}_0$$

Moreover,

$$\|\mathbf{G}_0\| = \left\| \frac{1}{m} \sum_{i \in \mathcal{I}_0} \sum_{s=0}^{\tau-1} (\mathbf{B}_{0,i,s} \mathbf{w}_{0,i,s} - \mathbf{B}_* \mathbf{w}_{*,i}) \mathbf{w}_{0,i,s}^\top \right\| \leq 7\sqrt{\alpha}\tau L_{\max}^2$$

by the initialization and $A_{2,0,i}(\tau)$ and $A_{3,0,i}(\tau)$, thus

$$\|\mathbf{B}_0^\top \mathbf{G}_0\| \leq 8\tau L_{\max}^2, \qquad \|\mathbf{G}_0^\top \mathbf{G}_0\| \leq 49\alpha\tau^2 L_{\max}^4$$

which implies that $\|\boldsymbol{\Delta}_1\| \leq \|\boldsymbol{\Delta}_0\| + 10\alpha^2 \tau L_{\max}^2 \leq c_3 \alpha^2 \tau L_{\max}^2 \kappa_{\max}^2 E_0^{-1}$, as desired.

Assume that the inductive hypotheses hold up to time $t$ and local round $s \geq 1$. We first show that the local inductive hypotheses hold for local round $s + 1$. Then, we show that the global inductions hold at time $t + 1$. This is achieved by the following lemmas.

**Local inductions.**

- $A_{2,t,i}(s) \cap A_{3,t,i}(s) \implies A_{1,t,i}(s+1)$. This is Lemma 2.
- $A_{1,t,i}(s+1) \cap A_{2,t,i}(s) \cap A_{3,t,i}(s) \implies A_{2,t,i}(s+1)$. This is Lemma 3.
- $A_{2,t,i}(s) \cap A_{3,t,i}(s) \implies A_{3,t,i}(s+1)$. This is Lemma 4.
- $A_{2,t,i}(s) \cap A_{3,t,i}(s+1) \cap A_3(t) \implies A_{4,t,i}(s+1)$. This is Lemma 5.

**Global inductions.**

- $\cap_{i \in \mathcal{I}_t} \big( A_{1,t,i}(\tau - 1) \cap A_{3,t,i}(\tau - 1) \big) \cap A_1(t) \cap A_2(t) \cap A_3(t) \implies A_1(t+1)$. This is Lemma 6.
- $\cap_{i \in \mathcal{I}_t} A_{2,t,i}(\tau) \implies A_2(t+1)$. This is Lemma 7.
- $\cap_{i \in \mathcal{I}_t} \big( \cap_{h=1}^4 A_{h,t,i}(\tau) \big) \cap A_1(t) \cap A_2(t) \cap A_3(t) \cap A_5(t) \implies A_3(t+1)$. This is Lemma 9.
- $\cap_{i \in \mathcal{I}_t} \big( A_{1,t,i}(\tau) \cap A_{2,t,i}(\tau) \cap A_{3,t,i}(\tau) \big) \cap A_2(t) \cap A_3(t) \implies A_4(t+1)$. This is Lemma 10.
- $A_3(t+1) \cap A_4(t+1) \cap A_5(t) \implies A_5(t+1)$. This is Lemma 11.

These inductions complete the proof. $\qquad\square$

**Lemma 2.** $A_{2,t,i}(s) \cap A_{3,t,i}(s) \implies A_{1,t,i}(s+1)$.

*Proof.* Since $\mathbf{w}_{t,i,s+1} = \boldsymbol{\Delta}_{t,i,s} \mathbf{w}_{t,i,s} + \alpha \mathbf{B}_{t,i,s}^\top \mathbf{B}_* \mathbf{w}_{*,i}$, we have

$$\|\mathbf{w}_{t,i,s+1} - \alpha \mathbf{B}_{t,i,s}^\top \mathbf{B}_* \mathbf{w}_{*,i}\| = \|\boldsymbol{\Delta}_{t,i,s} \mathbf{w}_{t,i,s}\|_2$$
$$\leq \|\boldsymbol{\Delta}_{t,i,s}\| \|\mathbf{w}_{t,i,s}\|_2$$
$$\leq 4c_3 \alpha^{2.5} \tau L_{\max}^3 \kappa_{\max}^2 \qquad (12)$$

where the last inequality follows by $A_{2,t,i}(s)$ and $A_{3,t,i}(s)$. $\qquad\square$

**Lemma 3.** $A_{1,t,i}(s+1) \cap A_{3,t,i}(s) \implies A_{2,t,i}(s+1)$.

*Proof.* Note that by the triangle inequality,

$$\|\mathbf{w}_{t,i,s+1}\| \leq \|\mathbf{w}_{t,i,s+1} - \alpha \mathbf{B}_{t,i,s}^\top \mathbf{B}_* \mathbf{w}_{*,i}\| + \|\alpha \mathbf{B}_{t,i,s}^\top \mathbf{B}_* \mathbf{w}_{*,i}\|$$
$$\leq 4c_3 \alpha^{2.5} \tau L_{\max}^3 \kappa_{\max}^2 E_0^{-1} + \|\alpha \mathbf{B}_{t,i,s}^\top \mathbf{B}_* \mathbf{w}_{*,i}\| \qquad (13)$$
$$\leq 4c_3 \alpha^{2.5} \tau L_{\max}^3 \kappa_{\max}^2 E_0^{-1} + 1.1\sqrt{\alpha} L_{\max} \qquad (14)$$
$$\leq 2\sqrt{\alpha} L_{\max}$$

where (13) follows by $A_{1,t,i}(s)$ and (14) follows by the fact that $\|\mathbf{B}_{t,i,s}\| \leq \frac{1.1}{\sqrt{\alpha}}$ by $A_{3,t,i}(s)$, and choice of $\alpha \leq (1 - \delta_0)(c_3 \sqrt{\tau} L_{\max} \kappa_{\max}^2)^{-1}$. $\qquad\square$

**Lemma 4.** $A_{2,t,i}(s) \cap A_{3,t,i}(s) \implies A_{3,t,i}(s+1)$.

*Proof.* Let $\mathbf{e}_{t,i,s} := \mathbf{B}_{t,i,s}\mathbf{w}_{t,i,s} - \mathbf{B}_*\mathbf{w}_{*,i}$ and $\mathbf{G}_{t,i,s} := \mathbf{e}_{t,i,s}\mathbf{w}_{t,i,s}$. We have

$$\mathbf{\Delta}_{t,i,s+1} = \mathbf{\Delta}_{t,i,s} + \alpha^2 \mathbf{B}_{t,i,s}^\top \mathbf{G}_{t,i,s} + \alpha^2 \mathbf{G}_{t,i,s}^\top \mathbf{B}_{t,i,s} - \alpha^3 \mathbf{G}_{t,i,s}^\top \mathbf{G}_{t,i,s}$$

We use $A_{2,t,i}(s)$ and $A_{3,t,i}(s)$ throughout the proof. Recall that $A_{3,t,i}(s)$ directly implies $\|\mathbf{B}_{t,i,s}\| \leq \frac{1.1}{\sqrt{\alpha}}$. This bound as well as the bound on $\|\mathbf{w}_{t,i,s}\|$ from $A_{2,t,i}(s)$ and the Cauchy Schwarz inequality implies $\|\mathbf{e}_{t,i,s}\| \leq 3.2L_{\max}$ and $\|\mathbf{G}_{t,i,s}\|_2 \leq 7\sqrt{\alpha}L_{\max}^2$, thus $\|\alpha^3 \mathbf{G}_{t,i,s}^\top \mathbf{G}_{t,i,s}\| \leq 49\alpha^4 L_{\max}^4$. Next,

$$\begin{aligned}
\mathbf{B}_{t,i,s}^\top \mathbf{G}_{t,i,s} &= \mathbf{B}_{t,i,s}^\top \mathbf{e}_{t,i,s}\mathbf{w}_{t,i,s}^\top \\
&= \alpha \mathbf{B}_{t,i,s}^\top \mathbf{e}_{t,i,s}\mathbf{w}_{*,i}^\top \mathbf{B}_*^\top \mathbf{B}_{t,i,s-1} + \mathbf{B}_{t,i,s}^\top \mathbf{e}_{t,i,s}\mathbf{w}_{t,i,s-1}^\top \mathbf{\Delta}_{t,i,s-1} \\
&= \alpha \mathbf{B}_{t,i,s}^\top \mathbf{e}_{t,i,s}\mathbf{w}_{*,i}^\top \mathbf{B}_*^\top \mathbf{B}_{t,i,s} + \alpha \mathbf{B}_{t,i,s}^\top \mathbf{e}_{t,i,s}\mathbf{w}_{*,i}^\top \mathbf{B}_*^\top (\mathbf{B}_{t,i,s-1} - \mathbf{B}_{t,i,s}) \\
&\quad + \mathbf{B}_{t,i,s}^\top \mathbf{e}_{t,i,s}\mathbf{w}_{t,i,s-1}^\top \mathbf{\Delta}_{t,i,s-1}
\end{aligned}$$

where, by the Cauchy-Schwarz inequality and $A_{2,t,i}(s)$ and $A_{3,t,i}(s)$

$$\|\alpha \mathbf{B}_{t,i,s}^\top \mathbf{e}_{t,i,s}\mathbf{w}_{*,i}^\top \mathbf{B}_*^\top (\mathbf{B}_{t,i,s-1} - \mathbf{B}_{t,i,s})\| = \alpha^2 \|\mathbf{B}_{t,i,s}^\top \mathbf{e}_{t,i,s}\mathbf{w}_{*,i}^\top \mathbf{B}_*^\top \mathbf{e}_{t,i,s-1}\mathbf{w}_{t,i,s-1}^\top\| \leq 23\alpha^2 L_{\max}^4,$$
$$\|\mathbf{B}_{t,i,s}^\top \mathbf{e}_{t,i,s}\mathbf{w}_{t,i,s-1}^\top \mathbf{\Delta}_{t,i,s-1}\| \leq 15c_3\alpha^2 \tau L_{\max}^4 \kappa_{\max}^2 E_0^{-1}, \tag{15}$$

and

$$\begin{aligned}
&\|\alpha \mathbf{B}_{t,i,s}^\top \mathbf{e}_{t,i,s}\mathbf{w}_{*,i}^\top \mathbf{B}_*^\top \mathbf{B}_{t,i,s}\| \\
&= \|\alpha^2 \mathbf{B}_{t,i,s}^\top \mathbf{B}_{t,i,s}\mathbf{B}_{t,i,s-1}^\top \mathbf{B}_*\mathbf{w}_{*,i}\mathbf{w}_{*,i}^\top \mathbf{B}_*^\top \mathbf{B}_{t,i,s} - \alpha \mathbf{B}_{t,i,s}^\top \mathbf{B}_*\mathbf{w}_{*,i}\mathbf{w}_{*,i}^\top \mathbf{B}_*^\top \mathbf{B}_{t,i,s} \\
&\quad + \alpha^2 \mathbf{B}_{t,i,s}^\top \mathbf{B}_{t,i,s}\mathbf{\Delta}_{t,i,s-1}\mathbf{w}_{t,i,s-1}\mathbf{w}_{*,i}^\top \mathbf{B}_*^\top \mathbf{B}_{t,i,s}\| \\
&= \| -\alpha \mathbf{\Delta}_{t,i,s}\mathbf{B}_{t,i,s}^\top \mathbf{B}_*\mathbf{w}_{*,i}\mathbf{w}_{*,i}^\top \mathbf{B}_*^\top \mathbf{B}_{t,i,s} \\
&\quad + \alpha^2 \mathbf{B}_{t,i,s}^\top \mathbf{B}_{t,i,s}(\mathbf{B}_{t,i,s-1} - \mathbf{B}_{t,i,s})^\top \mathbf{B}_*\mathbf{w}_{*,i}\mathbf{w}_{*,i}^\top \mathbf{B}_*^\top \mathbf{B}_{t,i,s} \\
&\quad + \alpha^2 \mathbf{B}_{t,i,s}^\top \mathbf{B}_{t,i,s}\mathbf{\Delta}_{t,i,s-1}\mathbf{w}_{t,i,s-1}\mathbf{w}_{*,i}^\top \mathbf{B}_*^\top \mathbf{B}_{t,i,s}\| \\
&\leq \alpha\|\mathbf{\Delta}_{t,i,s}\mathbf{B}_{t,i,s}^\top \mathbf{B}_*\mathbf{w}_{*,i}\mathbf{w}_{*,i}^\top \mathbf{B}_*^\top \mathbf{B}_{t,i,s}\| \\
&\quad + \alpha^2\|\mathbf{B}_{t,i,s}^\top \mathbf{B}_{t,i,s}\mathbf{w}_{t,i,s-1}\mathbf{e}_{t,i,s-1}^\top \mathbf{B}_*\mathbf{w}_{*,i}\mathbf{w}_{*,i}^\top \mathbf{B}_*^\top \mathbf{B}_{t,i,s}\| \\
&\quad + \alpha^2\|\mathbf{B}_{t,i,s}^\top \mathbf{B}_{t,i,s}\mathbf{\Delta}_{t,i,s-1}\mathbf{w}_{t,i,s-1}\mathbf{w}_{*,i}^\top \mathbf{B}_*^\top \mathbf{B}_{t,i,s}\| \\
&= 7c_3\alpha^2 \tau L_{\max}^4 \kappa_{\max}^2 E_0^{-1} + 9\alpha^2 L_{\max}^4. \tag{16}
\end{aligned}$$

Thus,

$$\begin{aligned}
\|\mathbf{\Delta}_{t,i,s+1}\|_2 &\leq \|\mathbf{\Delta}_{t,i,s}\|_2 + 2\alpha^2\|\mathbf{B}_{t,i,s}^\top \mathbf{G}_{t,i,s}\| + \alpha^3\|\mathbf{G}_{t,i,s}^\top \mathbf{G}_{t,i,s}\| \\
&\leq \|\mathbf{\Delta}_{t,i,s}\|_2 + 46c_3\alpha^4 \tau L_{\max}^4 \kappa_{\max}^2 E_0^{-1} + 81\alpha^4 L_{\max}^4 \\
&\vdots \\
&\leq \|\mathbf{\Delta}_t\|_2 + 46c_3\alpha^4 \tau^2 L_{\max}^4 \kappa_{\max}^2 E_0^{-1} + 81\alpha^4 \tau L_{\max}^4 \\
&\leq c_3\alpha^2 \tau L_{\max}^2 \kappa_{\max}^2 E_0^{-1} + 46c_3\alpha^4 \tau^2 L_{\max}^4 \kappa_{\max}^2 E_0^{-1} + 81\alpha^4 \tau L_{\max}^4 \\
&\leq 2c_3\alpha^2 \tau L_{\max}^2 \kappa_{\max}^2 E_0^{-1} \tag{17}
\end{aligned}$$

by choice of $c_3$ and $\alpha$ sufficiently small. $\square$

**Lemma 5.** $A_{2,t,i}(s) \cap A_{3,t,i}(s+1) \cap A_3(t) \implies A_{4,t,i}(s+1)$.

*Proof.* Note that

$$\begin{aligned}
\|\mathbf{B}_{*,\perp}^\top \mathbf{B}_{t,i,s+1}\|_2 &= \|\mathbf{B}_{*,\perp}^\top \mathbf{B}_{t,i,s}(\mathbf{I}_k - \alpha\mathbf{w}_{t,i,s}\mathbf{w}_{t,i,s}^\top)\|_2 \\
&\leq \|\mathbf{B}_{*,\perp}^\top \mathbf{B}_{t,i,s}\|_2 \\
&\vdots \\
&\leq \|\mathbf{B}_{*,\perp}^\top \mathbf{B}_t\|_2 \tag{18}
\end{aligned}$$

where the first inequality follows since $\|\mathbf{w}_{t,i,s}\| \le 2\sqrt{\alpha}L_{\max}$ (by $A_{2,t,i}(s)$) and $\alpha$ is sufficiently small, and the last inequality follows by recursively applying the first inequality for all local iterations leading up to $s$. Thus

$$\mathrm{dist}_{t,i,s} \le \frac{\|\mathbf{B}_t\|_2}{\sigma_{\min}(\mathbf{B}_{t,i,s})}\,\mathrm{dist}_t \le \sqrt{\frac{1+c_3\alpha^2\tau L_{\max}^2\kappa_{\max}^2 E_0^{-1}}{1-2c_3\alpha^2\tau L_{\max}^2\kappa_{\max}^2 E_0^{-1}}}\,\mathrm{dist}_t \le 2\,\mathrm{dist}_t\,.$$

$\square$

**Lemma 6.** $\cap_{i\in\mathcal{I}_t}\big(A_{2,t,i}(\tau-1)\cap A_{3,t,i}(\tau-1)\big)\cap A_2(t)\cap A_3(t) \implies A_1(t+1).$

*Proof.* Expanding $\mathbf{w}_{t+1}$ yields

$$
\begin{aligned}
\mathbf{w}_{t+1} &= \tfrac{1}{m}\sum_{i\in\mathcal{I}_t}\mathbf{w}_{t,i,\tau}\\
&= \tfrac{1}{m}\sum_{i\in\mathcal{I}_t}\boldsymbol{\Delta}_{t,i,\tau-1}\mathbf{w}_{t,i,\tau-1} + \alpha\mathbf{B}_{t,i,\tau-1}^\top\mathbf{B}_*\mathbf{w}_{*,i}\\
&= \alpha\mathbf{B}_t^\top\mathbf{B}_*\bar{\mathbf{w}}_{*,t} + \tfrac{1}{m}\sum_{i\in\mathcal{I}_t}\boldsymbol{\Delta}_{t,i,\tau-1}\mathbf{w}_{t,i,\tau-1} + \alpha(\mathbf{B}_{t,i,\tau-1}-\mathbf{B}_t)^\top\mathbf{B}_*\mathbf{w}_{*,i}\\
&= \tfrac{1}{n}\sum_{i\in\mathcal{I}_t}\alpha(\mathbf{B}_{t,i,\tau-1}-\mathbf{B}_t)^\top\mathbf{B}_*\mathbf{w}_{*,i} + \alpha\boldsymbol{\Delta}_{t,i,\tau-1}\mathbf{B}_{t,i,\tau-2}^\top\mathbf{B}_*\mathbf{w}_{*,i}\\
&\qquad\qquad + \alpha\boldsymbol{\Delta}_{t,i,\tau-1}\boldsymbol{\Delta}_{t,i,\tau-2}\mathbf{w}_{t,i,\tau-2} + \alpha\mathbf{B}_t^\top\mathbf{B}_*\bar{\mathbf{w}}_{*,t}\\
&= \alpha\mathbf{B}_t^\top\mathbf{B}_*\bar{\mathbf{w}}_{*,t} + \alpha\boldsymbol{\Delta}_t\mathbf{B}_t^\top\mathbf{B}_*\bar{\mathbf{w}}_{*,t}\\
&\qquad + \tfrac{1}{m}\sum_{i\in\mathcal{I}_t}\alpha(\mathbf{B}_{t,i,\tau-1}-\mathbf{B}_t)^\top\mathbf{B}_*\mathbf{w}_{*,i} + \alpha(\boldsymbol{\Delta}_{t,i,\tau-1}\mathbf{B}_{t,i,\tau-2}^\top - \boldsymbol{\Delta}_t\mathbf{B}_t^\top)\mathbf{B}_*\mathbf{w}_{*,i}\\
&\qquad\qquad + \boldsymbol{\Delta}_{t,i,\tau-1}\boldsymbol{\Delta}_{t,i,\tau-2}\mathbf{w}_{t,i,\tau-2}\\
&= \alpha\mathbf{B}_{t+1}^\top\mathbf{B}_*\bar{\mathbf{w}}_{*,t+1} + \alpha\boldsymbol{\Delta}_{t+1}\mathbf{B}_{t+1}^\top\mathbf{B}_*\bar{\mathbf{w}}_{*,t+1}\\
&\qquad + \alpha\mathbf{B}_{t+1}^\top\mathbf{B}_*(\bar{\mathbf{w}}_{*,t}-\bar{\mathbf{w}}_{*,t+1}) + \alpha\boldsymbol{\Delta}_{t+1}\mathbf{B}_{t+1}^\top\mathbf{B}_*(\bar{\mathbf{w}}_{*,t}-\bar{\mathbf{w}}_{*,t+1})\\
&\qquad + \alpha(\mathbf{B}_t-\mathbf{B}_{t+1})^\top\mathbf{B}_*\bar{\mathbf{w}}_{*,t} + \alpha(\boldsymbol{\Delta}_t\mathbf{B}_t - \boldsymbol{\Delta}_{t+1}\mathbf{B}_{t+1})^\top\mathbf{B}_*\bar{\mathbf{w}}_{*,t}\\
&\qquad + \tfrac{1}{m}\sum_{i\in\mathcal{I}_t}\alpha(\mathbf{B}_{t,i,\tau-1}-\mathbf{B}_t)^\top\mathbf{B}_*\mathbf{w}_{*,i} + \alpha(\boldsymbol{\Delta}_{t,i,\tau-1}\mathbf{B}_{t,i,\tau-2}^\top - \boldsymbol{\Delta}_t\mathbf{B}_t^\top)\mathbf{B}_*\mathbf{w}_{*,i}\\
&\qquad\qquad + \boldsymbol{\Delta}_{t,i,\tau-1}\boldsymbol{\Delta}_{t,i,\tau-2}\mathbf{w}_{t,i,\tau-2} \tag{19}
\end{aligned}
$$

The remainder of the proof lies in bounding the error terms, which are all terms in the RHS of (19) besides the terms in the first line. First, by $A_0$ and the triangle inequality, we have

$$\|\bar{\mathbf{w}}_{*,t}-\bar{\mathbf{w}}_{*,t+1}\| \le \|\bar{\mathbf{w}}_{*,t}-\bar{\mathbf{w}}_*\| + \|\bar{\mathbf{w}}_{*,t+1}-\bar{\mathbf{w}}_*\| \le 8\alpha^2 L_{\max}^3$$

Thus, by $A_3(t+1)$, we have

$$\|\alpha\mathbf{B}_{t+1}^\top\mathbf{B}_*(\bar{\mathbf{w}}_{*,t}-\bar{\mathbf{w}}_{*,t+1})\| \le 1.1\sqrt{\alpha}\|\bar{\mathbf{w}}_{*,t}-\bar{\mathbf{w}}_{*,t+1}\| \le 9\alpha^{2.5}L_{\max}^3$$
$$\|\alpha\boldsymbol{\Delta}_{t+1}\mathbf{B}_{t+1}^\top\mathbf{B}_*(\bar{\mathbf{w}}_{*,t}-\bar{\mathbf{w}}_{*,t+1})\| \le 1.1\sqrt{\alpha}\|\boldsymbol{\Delta}_{t+1}\|\|\bar{\mathbf{w}}_{*,t}-\bar{\mathbf{w}}_{*,t+1}\| \le 18c_3\alpha^{4.5}\tau L_{\max}^5\kappa_{\max}^2 E_0^{-1}$$

Next, we can bound the difference between the locally-updated representation and the global representation as follows, for any $s\in\{1,\dots,\tau\}$

$$\|\mathbf{B}_{t,i,s}-\mathbf{B}_t\|_2 \le \sum_{r=1}^{s}\|\mathbf{B}_{t,i,r}-\mathbf{B}_{t,i,r-1}\|_2 \le \alpha\sum_{r=1}^{s}\|\mathbf{e}_{t,i,r-1}\mathbf{w}_{t,i,r-1}^\top\|_2 \le 7\alpha^{1.5}sL_{\max}^2 \tag{20}$$

using $A_2(t), A_3(t), A_{2,t,i}(\tau-1)$ and $A_{3,t,i}(\tau-1)$ to control the norms of $\mathbf{w}_{t,i,s-1}$ and $\mathbf{B}_{t,i,s-1}$. From (20) it follows that

$$\|\mathbf{B}_{t+1} - \mathbf{B}_t\|_2 \leq \tfrac{1}{m}\sum_{i\in\mathcal{I}_t}\|\mathbf{B}_{t,i,\tau} - \mathbf{B}_t\|_2 \leq 7\alpha^{1.5}\tau L_{\max}^2$$

$$\begin{aligned}
\|\mathbf{B}_{t,i,\tau-2}\boldsymbol{\Delta}_{t,i,\tau-1} - \mathbf{B}_t\boldsymbol{\Delta}_t\|_2 &\leq \|\mathbf{B}_{t,i,\tau-2} - \mathbf{B}_t\|_2 + \alpha\|\mathbf{B}_{t,i,\tau-2}\mathbf{B}_{t,i,\tau-1}^\top\mathbf{B}_{t,i,\tau-1} - \mathbf{B}_t\mathbf{B}_t^\top\mathbf{B}_t\|_2 \\
&\leq \|\mathbf{B}_{t,i,\tau-2} - \mathbf{B}_t\|_2 + \alpha\|(\mathbf{B}_{t,i,\tau-2} - \mathbf{B}_t)\mathbf{B}_{t,i,\tau-1}^\top\mathbf{B}_{t,i,\tau-1}\|_2 \\
&\quad + \alpha\|\mathbf{B}_t(\mathbf{B}_{t,i,\tau-1} - \mathbf{B}_t)^\top\mathbf{B}_{t,i,\tau-1}\|_2 \\
&\quad + \alpha\|\mathbf{B}_t\mathbf{B}_t^\top(\mathbf{B}_{t,i,\tau-1} - \mathbf{B}_t)\|_2 \\
&\leq \|\mathbf{B}_{t,i,\tau-2} - \mathbf{B}_t\|_2 + \alpha\|\mathbf{B}_{t,i,\tau-2} - \mathbf{B}_t\|\|\mathbf{B}_{t,i,\tau-1}^\top\mathbf{B}_{t,i,\tau-1}\|_2 \\
&\quad + \alpha\|\mathbf{B}_t\|\|\mathbf{B}_{t,i,\tau-1} - \mathbf{B}_t\|\|\mathbf{B}_{t,i,\tau-1}\|_2 \\
&\quad + \alpha\|\mathbf{B}_t\mathbf{B}_t^\top\|\|\mathbf{B}_{t,i,\tau-1} - \mathbf{B}_t\| \\
&\leq 31\alpha^{1.5}\tau L_{\max}^2
\end{aligned}$$

$$\|\mathbf{B}_t\boldsymbol{\Delta}_t - \mathbf{B}_{t+1}\boldsymbol{\Delta}_{t+1}\|_2 \leq 31\alpha^{1.5}\tau L_{\max}^2 \tag{21}$$

Also, we have by $A_{2,t,i}(\tau-1)$ and $A_{3,t,i}(\tau-1)$,

$$\|\boldsymbol{\Delta}_{t,i,\tau-1}\boldsymbol{\Delta}_{t,i,\tau-2}\mathbf{w}_{t,i,\tau-2}\|_2 \leq 8c_3^2\alpha^{4.5}\tau^2 L_{\max}^5\kappa_{\max}^4 E_0^{-2}. \tag{22}$$

Thus, using these bounds with (19), we obtain

$$\begin{aligned}
\|\mathbf{w}_{t+1} - \alpha(\mathbf{I}_k + \boldsymbol{\Delta}_{t+1})\mathbf{B}_{t+1}^\top\mathbf{B}_*\bar{\mathbf{w}}_{*,t+1}\|_2 &\leq 82\alpha^{2.5}\tau L_{\max}^3 + (8c_3^2 + 12c_3)\alpha^{4.5}\tau^2 L_{\max}^5\kappa_{\max}^4 E_0^{-2} \\
&\leq 91\alpha^{2.5}\tau L_{\max}^3
\end{aligned}$$

to complete the proof, where we have used that $\alpha$ is sufficiently small in the last inequality. $\square$

**Lemma 7.** $\cap_{i\in\mathcal{I}_t}A_{2,t,i}(\tau) \implies A_2(t+1)$

*Proof.* By the triangle inequality and $\cap_{i\in\mathcal{I}_t}A_{2,t,i}(\tau)$, we have

$$\|\mathbf{w}_{t+1}\| = \left\|\tfrac{1}{m}\sum_{i\in\mathcal{I}_t}\mathbf{w}_{t,i,\tau}\right\| \leq \tfrac{1}{m}\sum_{i\in\mathcal{I}_t}\|\mathbf{w}_{t,i,\tau}\| \leq 2\sqrt{\alpha}L_{\max}$$

as desired. $\square$

**Lemma 8.** $A_3(t) \cap A_4(t) \implies \sigma_{\min}^2(\mathbf{B}_t^\top\mathbf{B}_*) \geq \frac{0.1}{\alpha}E_0$.

*Proof.* First note that

$$\begin{aligned}
\sigma_{\min}^2(\mathbf{B}_t^\top\mathbf{B}_*) &\geq \sigma_{\min}^2(\mathbf{R}_t)\sigma_{\min}^2(\hat{\mathbf{B}}_t^\top\mathbf{B}_*) \\
&\geq \tfrac{0.9}{\alpha}\sigma_{\min}^2(\hat{\mathbf{B}}_t^\top\mathbf{B}_*) \tag{23} \\
&= \tfrac{0.9}{\alpha}(1 - \|\hat{\mathbf{B}}_t^\top\mathbf{B}_{*,\perp}\|_2^2) \\
&= \tfrac{0.9}{\alpha}(1 - \text{dist}_t^2) \tag{24}
\end{aligned}$$

where $\hat{\mathbf{B}}_t\mathbf{R}_t = \mathbf{B}_t$ is the QR factorization of $\mathbf{B}_t$. Next, we would like to show the RHS is at most $(2+\delta_0)/3$. Using $A_3(t)$ and $A_4(t)$, we obtain

$$\begin{aligned}
\text{dist}_t &= \|\mathbf{B}_{*,\perp}^\top\hat{\mathbf{B}}_t\|_2 \\
&\leq \tfrac{1}{\sigma_{\min}(\mathbf{B}_t)}\|\mathbf{B}_{*,\perp}^\top\mathbf{B}_t\|_2 \\
&\;\;\vdots \\
&\leq \tfrac{1}{\sigma_{\min}(\mathbf{B}_{t+1})}(1 - 0.04\alpha^2\tau E_0\mu^2)^t\|\mathbf{B}_{*,\perp}^\top\mathbf{B}_0\|_2 \\
&\leq \tfrac{\sigma_{\max}(\mathbf{B}_0)}{\sigma_{\min}(\mathbf{B}_t)}(1 - 0.04\alpha^2\tau E_0\mu^2)^t\delta_0 \\
&\leq \tfrac{\sqrt{1+\|\boldsymbol{\Delta}_0\|_2}/\sqrt{\alpha}}{\sqrt{1-\|\boldsymbol{\Delta}_t\|_2}/\sqrt{\alpha}}\delta_0
\end{aligned}$$

Next we use that $\|\mathbf{\Delta}_0\| \leq \alpha^2 \tau L_{\max}^2 \kappa_{\max}^2 \leq 0.1(1-\delta_0)^2$ by choice of initialization and choice of $\alpha$, and similarly $\|\mathbf{\Delta}_t\| \leq \alpha^2 \tau L_{\max}^2 \kappa_{\max}^2 E_0^{-1} \leq 0.1(1-\delta_0)^2/(1-\delta_0^2)$. Let $c := 0.1$. Then we have

$$\frac{\sqrt{1+\|\mathbf{\Delta}_0\|_2/\sqrt{\alpha}}}{\sqrt{1-\|\mathbf{\Delta}_t\|_2/\sqrt{\alpha}}}\delta_0 \leq \frac{\sqrt{1+c(1-\delta_0)^2}}{\sqrt{1-c(1-\delta_0)^2/(1-\delta_0^2)}}\delta_0$$

$$= \frac{\sqrt{1+c(1-\delta_0)^2}}{\sqrt{1-c(1-\delta_0)/(1+\delta_0)}}\delta_0$$

$$= \frac{\sqrt{1+\delta_0+c(1-\delta_0)^2(1+\delta_0)}}{\sqrt{1-c+(1+c)\delta_0}}\delta_0$$

Now, observe that

$$\frac{\sqrt{1+\delta_0+c(1-\delta_0)^2(1+\delta_0)}}{\sqrt{1-c+(1+c)\delta_0}}\delta_0 \leq \frac{2+\delta_0}{3}$$

$$\iff \frac{1+\delta_0+c(1-\delta_0)^2(1+\delta_0)}{1-c+(1+c)\delta_0}\delta_0^2 \leq \frac{4+4\delta_0+\delta_0^2}{9}$$

$$\iff (1+c)\delta_0^2 + \delta_0^3 - c\delta_0^4 + c\delta_0^5 \leq (4 - 4c + 8\delta_0 + 8\delta_0^2 + (1+c)\delta_0^3)/9$$

$$\iff c\delta_0^5 - c\delta_0^4 + \tfrac{8-c}{9}\delta_0^3 + \tfrac{1+9c}{9}\delta_0^2 - \tfrac{8}{9}\delta_0 - \tfrac{4-4c}{9} \leq 0 \tag{25}$$

where (25) holds for all $\delta_0 \in [0,1)$ and $c = 0.1$, therefore we have

$$\text{dist}_t \leq \frac{\sqrt{1+\|\mathbf{\Delta}_0\|_2/\sqrt{\alpha}}}{\sqrt{1-\|\mathbf{\Delta}_t\|_2/\sqrt{\alpha}}}\delta_0 \leq \frac{2+\delta_0}{3}. \tag{26}$$

Thus, using (24), we obtain

$$\sigma_{\min}^2(\mathbf{B}_t^\top \mathbf{B}_*) \geq \frac{0.9}{\alpha}\left(1 - \frac{4+4\delta_0+\delta_0^2}{9}\right)$$

$$\geq \frac{0.9}{\alpha}\left(1 - \frac{8+\delta_0^2}{9}\right)$$

$$= \frac{0.9}{9\alpha}E_0$$

$$= \frac{0.1}{\alpha}E_0$$

as desired. $\qquad\qquad\square$

**Lemma 9.** $\cap_{i \in \mathcal{I}_t}\left(\cap_{h=1}^4 A_{h,t,i}(\tau)\right) \cap A_1(t) \cap A_2(t) \cap A_3(t) \cap A_5(t) \implies A_3(t+1)$.

*Proof.* We aim to write $\mathbf{\Delta}_{t+1} = \frac{1}{2}(\mathbf{I}_k - \mathbf{P}_t)\mathbf{\Delta}_t + \frac{1}{2}\mathbf{\Delta}_t(\mathbf{I}_k - \mathbf{P}_t) + \mathbf{Z}_t$ for a positive definite matrix $\mathbf{P}_t$ and a perturbation matrix $\mathbf{Z}_t$. This will yield the inequality $\|\mathbf{\Delta}_{t+1}\|_2 \leq (1 - \lambda_{\min}(\mathbf{P}_t))\|\mathbf{\Delta}_t\|_2 + \|\mathbf{Z}_t\|_2$. Assuming $\lambda_{\min}(\mathbf{P}_t)$ and $\|\mathbf{Z}_t\|_2$ scale appropriately (defined later), this inequality combined with inductive hypothesis $A_5(t)$ will give the desired upper bound on $\|\mathbf{\Delta}_{t+1}\|_2$ (this is because the upper bound on $\|\mathbf{Z}_t\|_2$ scales with $\text{dist}_t$, so $A_5(t)$ contributes to controlling $\|\mathbf{Z}_t\|_2$). The proof therefore relies on showing the existence of appropriate $\mathbf{P}_t$ and $\mathbf{Z}_t$.

First recall $\mathbf{\Delta}_t := \mathbf{I} - \alpha\mathbf{B}_t^\top \mathbf{B}_t$ and $\bar{\mathbf{\Delta}}_t := \mathbf{I}_d - \alpha\mathbf{B}_t\mathbf{B}_t^\top$. Let $\mathbf{G}_t := \frac{1}{\alpha}(\mathbf{B}_t - \mathbf{B}_{t+1})$, i.e. $\mathbf{G}_t$ satisfies $\mathbf{B}_{t+1} = \mathbf{B}_t - \alpha\mathbf{G}_t$. Then

$$\mathbf{\Delta}_{t+1} = \mathbf{I}_k - \alpha\mathbf{B}_{t+1}^\top\mathbf{B}_{t+1} = \mathbf{\Delta}_t + \alpha^2\mathbf{B}_t^\top\mathbf{G}_t + \alpha^2\mathbf{G}_t^\top\mathbf{B}_t - \alpha^3\mathbf{G}_t^\top\mathbf{G}_t \tag{27}$$

The key is showing that $\alpha^2\mathbf{B}_t^\top\mathbf{G}_t = -\frac{1}{2}\mathbf{\Delta}_t\mathbf{P}_t + \mathbf{Z}_t'$ for appropriate $\mathbf{P}_t$ and $\mathbf{Z}_t'$. Then, by (27), we will have $\mathbf{\Delta}_{t+1} = \frac{1}{2}(\mathbf{I}_k - \mathbf{P}_t)\mathbf{\Delta}_t + \frac{1}{2}\mathbf{\Delta}_t(\mathbf{I}_k - \mathbf{P}_t) + \mathbf{Z}_t$ as desired, where $\mathbf{Z}_t = \mathbf{Z}_t' + (\mathbf{Z}_t')^\top - \alpha^3\mathbf{G}_t^\top\mathbf{G}_t$.

Notice that $\mathbf{G}_t$ is the average across clients of the sum of their local gradients on every local update. In particular, we have

$$\mathbf{G}_t = (\mathbf{B}_t\mathbf{w}_t - \mathbf{B}_*\bar{\mathbf{w}}_{*,t})\mathbf{w}_t^\top + \frac{1}{m}\sum_{i \in \mathcal{I}_t}\sum_{s=1}^{\tau-1}(\mathbf{B}_{t,i,s}\mathbf{w}_{t,i,s} - \mathbf{B}_*\mathbf{w}_{*,i})\mathbf{w}_{t,i,s}^\top \tag{28}$$

We will unroll the gradients for the first two local updates only, in order to obtain a negative term that will contribute to the contraction of $\|\mathbf{\Delta}_t\|$ (i.e. $\mathbf{P}_t$ will be extracted from the gradients for the first two local updates). The remaining terms will belong to $\mathbf{Z}_t$ and must be upper bounded (i.e. $\|\mathbf{\Delta}_{t+1}\|$

can grow due to local updates beyond the second local update, but we will show that it can't grow too much). In particular, we have

$$\mathbf{G}_t = (\mathbf{B}_t\mathbf{w}_t - \mathbf{B}_*\bar{\mathbf{w}}_{*,t})\mathbf{w}_t^\top + \frac{1}{m}\sum_{i\in\mathcal{I}_t}(\mathbf{B}_{t,i,1}\mathbf{w}_{t,i,1} - \mathbf{B}_*\mathbf{w}_{*,i})\mathbf{w}_{t,i,1}^\top$$

$$+ \frac{1}{m}\sum_{i\in\mathcal{I}_t}\sum_{s=2}^{\tau-1}(\mathbf{B}_{t,i,s}\mathbf{w}_{t,i,s} - \mathbf{B}_*\mathbf{w}_{*,i})\mathbf{w}_{t,i,s}^\top$$

$$= (\mathbf{B}_t\mathbf{w}_t - \mathbf{B}_*\bar{\mathbf{w}}_{*,t})\mathbf{w}_t^\top - \alpha\bar{\mathbf{\Delta}}_t\mathbf{B}_*\frac{1}{m}\sum_{i\in\mathcal{I}_t}\mathbf{w}_{*,i}\mathbf{w}_{*,i}^\top\mathbf{B}_*^\top\mathbf{B}_t - \bar{\mathbf{\Delta}}_t\mathbf{B}_*\bar{\mathbf{w}}_{*,t}\mathbf{w}_t^\top\mathbf{\Delta}_t$$

$$+ \frac{1}{m}\sum_{i\in\mathcal{I}_t}\alpha^2(\mathbf{B}_t\mathbf{w}_t - \mathbf{B}_*\mathbf{w}_{*,i})\mathbf{w}_t^\top\mathbf{B}_t^\top\mathbf{B}_*\mathbf{w}_{*,i}\mathbf{w}_{t,i,1}^\top$$

$$+ \frac{1}{m}\sum_{i\in\mathcal{I}_t}(\mathbf{B}_t - \alpha(\mathbf{B}_t\mathbf{w}_t - \mathbf{B}_*\mathbf{w}_{*,i})\mathbf{w}_t^\top)\mathbf{\Delta}_t\mathbf{w}_t\mathbf{w}_{t,i,1}^\top$$

$$+ \frac{1}{m}\sum_{i\in\mathcal{I}_t}\sum_{s=2}^{\tau-1}(\mathbf{B}_{t,i,s}\mathbf{w}_{t,i,s} - \mathbf{B}_*\mathbf{w}_{*,i})\mathbf{w}_{t,i,s}^\top$$

Multiplying both sides by $\mathbf{B}_t^\top$, and using the fact that $\mathbf{B}_t^\top\bar{\mathbf{\Delta}}_t = \mathbf{\Delta}_t\mathbf{B}_t^\top$, we obtain

$$\mathbf{B}_t^\top\mathbf{G}_t = \mathbf{B}_t^\top(\mathbf{B}_t\mathbf{w}_t - \mathbf{B}_*\bar{\mathbf{w}}_{*,t})\mathbf{w}_t^\top - \alpha\mathbf{\Delta}_t\mathbf{B}_t^\top\mathbf{B}_*\frac{1}{m}\sum_{i\in\mathcal{I}_t}\mathbf{w}_{*,i}\mathbf{w}_{*,i}^\top\mathbf{B}_*^\top\mathbf{B}_t - \mathbf{\Delta}_t\mathbf{B}_t^\top\mathbf{B}_*\bar{\mathbf{w}}_{*,t}\mathbf{w}_t^\top\mathbf{\Delta}_t$$

$$- \mathbf{B}_t^\top\frac{1}{m}\sum_{i\in\mathcal{I}_t}\alpha^2(\mathbf{B}_t\mathbf{w}_t - \mathbf{B}_*\mathbf{w}_{*,i})\mathbf{w}_t^\top\mathbf{B}_t^\top\mathbf{B}_*\mathbf{w}_{*,i}\mathbf{w}_{t,i,1}^\top$$

$$+ \mathbf{B}_t^\top\frac{1}{m}\sum_{i\in\mathcal{I}_t}(\mathbf{B}_t - \alpha(\mathbf{B}_t\mathbf{w}_t - \mathbf{B}_*\mathbf{w}_{*,i})\mathbf{w}_t^\top)\mathbf{\Delta}_t\mathbf{w}_t\mathbf{w}_{t,i,1}^\top$$

$$+ \mathbf{B}_t^\top\frac{1}{m}\sum_{i\in\mathcal{I}_t}\sum_{s=2}^{\tau-1}(\mathbf{B}_{t,i,s}\mathbf{w}_{t,i,s} - \mathbf{B}_*\mathbf{w}_{*,i})\mathbf{w}_{t,i,s}^\top$$

$$= -\alpha\mathbf{\Delta}_t\mathbf{B}_t^\top\mathbf{B}_*\frac{1}{m}\sum_{i\in\mathcal{I}_t}\mathbf{w}_{*,i}\mathbf{w}_{*,i}^\top\mathbf{B}_*^\top\mathbf{B}_t + \mathbf{N}_t \tag{29}$$

where the first term is a negative term that helps $\|\mathbf{\Delta}_{t+1}\|$ stay small, and the remaining terms are given by

$$
\begin{aligned}
\mathbf{N}_t := {}& \mathbf{B}_t^\top (\mathbf{B}_t\mathbf{w}_t - \mathbf{B}_*\bar{\mathbf{w}}_{*,t})\mathbf{w}_t^\top - \mathbf{\Delta}_t\mathbf{B}_t^\top\mathbf{B}_*\bar{\mathbf{w}}_{*,t}\mathbf{w}_t^\top\mathbf{\Delta}_t \\
& - \mathbf{B}_t^\top \tfrac{1}{m}\sum_{i\in\mathcal{I}_t}\alpha^2(\mathbf{B}_t\mathbf{w}_t - \mathbf{B}_*\mathbf{w}_{*,i})\mathbf{w}_t^\top\mathbf{B}_t^\top\mathbf{B}_*\mathbf{w}_{*,i}\mathbf{w}_{t,i,1}^\top \\
& + \mathbf{B}_t^\top \tfrac{1}{m}\sum_{i\in\mathcal{I}_t}(\mathbf{B}_t - \alpha(\mathbf{B}_t\mathbf{w}_t - \mathbf{B}_*\mathbf{w}_{*,i})\mathbf{w}_t^\top)\mathbf{\Delta}_t\mathbf{w}_t\mathbf{w}_{t,i,1}^\top \\
& + \mathbf{B}_t^\top \tfrac{1}{m}\sum_{i\in\mathcal{I}_t}\sum_{s=2}^{\tau-1}(\mathbf{B}_{t,i,s}\mathbf{w}_{t,i,s} - \mathbf{B}_*\mathbf{w}_{*,i})\mathbf{w}_{t,i,s}^\top \\
= {}& \mathbf{B}_t^\top (\mathbf{B}_t\mathbf{w}_t - \mathbf{B}_*\bar{\mathbf{w}}_{*,t})\mathbf{w}_t^\top + \mathbf{B}_t^\top\mathbf{B}_t\mathbf{\Delta}_t\mathbf{w}_t\tfrac{1}{m}\sum_{i\in\mathcal{I}_t}\mathbf{w}_{t,i,1}^\top \\
& - \mathbf{B}_t^\top \tfrac{1}{m}\sum_{i\in\mathcal{I}_t}\alpha^2(\mathbf{B}_t\mathbf{w}_t - \mathbf{B}_*\mathbf{w}_{*,i})\mathbf{w}_t^\top\mathbf{B}_t^\top\mathbf{B}_*\mathbf{w}_{*,i}\mathbf{w}_{t,i,1}^\top \\
& - \mathbf{B}_t^\top \tfrac{1}{m}\sum_{i\in\mathcal{I}_t}\alpha(\mathbf{B}_t\mathbf{w}_t - \mathbf{B}_*\mathbf{w}_{*,i})\mathbf{w}_t^\top\mathbf{\Delta}_t\mathbf{w}_t\mathbf{w}_{t,i,1}^\top \\
& - \mathbf{\Delta}_t\mathbf{B}_t^\top\mathbf{B}_*\bar{\mathbf{w}}_{*,t}\mathbf{w}_t^\top\mathbf{\Delta}_t + \mathbf{B}_t^\top \tfrac{1}{m}\sum_{i\in\mathcal{I}_t}\sum_{s=2}^{\tau-1}(\mathbf{B}_{t,i,s}\mathbf{w}_{t,i,s} - \mathbf{B}_*\mathbf{w}_{*,i})\mathbf{w}_{t,i,s}^\top \\
= {}& \underbrace{\mathbf{B}_t^\top (\mathbf{B}_t\mathbf{w}_t - \mathbf{B}_*\bar{\mathbf{w}}_{*,t})\mathbf{w}_t^\top + \alpha\mathbf{B}_t^\top\mathbf{B}_t\mathbf{\Delta}_t\mathbf{w}_t\bar{\mathbf{w}}_{*,t}^\top\mathbf{B}_*^\top\mathbf{B}_t}_{=:\mathbf{E}_1} \\
& - \underbrace{\tfrac{1}{m}\sum_{i\in\mathcal{I}_t}\alpha\mathbf{B}_t^\top (\mathbf{B}_t\mathbf{w}_t - \mathbf{B}_*\mathbf{w}_{*,i})\mathbf{w}_t^\top\mathbf{w}_{t,i,1}\mathbf{w}_{t,i,1}^\top}_{=:\mathbf{E}_2} \\
& + \underbrace{\mathbf{\Delta}_t\mathbf{B}_t^\top (\mathbf{B}_t\mathbf{w}_t - \mathbf{B}_*\bar{\mathbf{w}}_{*,t})\mathbf{w}_t^\top\mathbf{\Delta}_t}_{=:\mathbf{E}_3} + \underbrace{\mathbf{B}_t^\top \tfrac{1}{m}\sum_{i\in\mathcal{I}_t}\sum_{s=2}^{\tau-1}(\mathbf{B}_{t,i,s}\mathbf{w}_{t,i,s} - \mathbf{B}_*\mathbf{w}_{*,i})\mathbf{w}_{t,i,s}^\top}_{=:\mathbf{E}_4} \quad (30)
\end{aligned}
$$

To get from the first to the second equation we expanded the fourth term in the first equation. Now we need to upper bound the spectral norm of each of the terms $\mathbf{E}_i$. The matrices $\mathbf{E}_2$ and $\mathbf{E}_3$ are straightforward to control; we will take care of them shortly. For now we are concerned with $\mathbf{E}_1$. In order to control this matrix, we must use the fact that $\mathbf{w}_t$ is close to a matrix times $\bar{\mathbf{w}}_{*,t}$. This will allow us to subsume the dominant term from $\mathbf{E}_1$ into the negative term in (29). In particular, note that by $A_1(t)$, we have $\mathbf{w}_t = \alpha\mathbf{B}_t^\top\mathbf{B}_*\bar{\mathbf{w}}_{*,t} + \alpha\mathbf{\Delta}_t\mathbf{B}_t^\top\mathbf{B}_*\bar{\mathbf{w}}_{*,t} + \mathbf{h}_t$, where $\|\mathbf{h}_t\|_2 \le 91\alpha^{2.5}\tau L_{\max}^3$. This implies that

$$
\begin{aligned}
\mathbf{B}_t^\top (\mathbf{B}_t\mathbf{w}_t - \mathbf{B}_*\bar{\mathbf{w}}_{*,t}) &= \mathbf{B}_t^\top (\alpha\mathbf{B}_t\mathbf{B}_t^\top\mathbf{B}_*\bar{\mathbf{w}}_{*,t} - \mathbf{B}_*\bar{\mathbf{w}}_{*,t}) + \alpha\mathbf{\Delta}_t\mathbf{B}_t^\top\mathbf{B}_t\mathbf{B}_t^\top\mathbf{B}_*\bar{\mathbf{w}}_{*,t} + \mathbf{B}_t^\top\mathbf{B}_t\mathbf{h}_t \\
&= -\mathbf{\Delta}_t\mathbf{B}_t^\top\mathbf{B}_*\bar{\mathbf{w}}_{*,t} + \alpha\mathbf{\Delta}_t\mathbf{B}_t^\top\mathbf{B}_t\mathbf{B}_t^\top\mathbf{B}_*\bar{\mathbf{w}}_{*,t} + \mathbf{B}_t^\top\mathbf{B}_t\mathbf{h}_t \\
&= -\mathbf{\Delta}_t^2\mathbf{B}_t^\top\mathbf{B}_*\bar{\mathbf{w}}_{*,t} + \mathbf{B}_t^\top\mathbf{B}_t\mathbf{h}_t \qquad (31)
\end{aligned}
$$

Making this substitution in $\mathbf{E}_1$, we obtain,

$$
\begin{aligned}
\mathbf{E}_1 &= -\mathbf{\Delta}_t^2\mathbf{B}_t^\top\mathbf{B}_*\bar{\mathbf{w}}_{*,t}\mathbf{w}_t^\top + \mathbf{B}_t^\top\mathbf{B}_t\mathbf{h}_t\mathbf{w}_t^\top + \alpha\mathbf{B}_t^\top\mathbf{B}_t\mathbf{\Delta}_t\mathbf{w}_t\bar{\mathbf{w}}_{*,t}^\top\mathbf{B}_*^\top\mathbf{B}_t \\
&= -\mathbf{\Delta}_t^2\mathbf{B}_t^\top\mathbf{B}_*\bar{\mathbf{w}}_{*,t}\mathbf{w}_t^\top + \mathbf{B}_t^\top\mathbf{B}_t\mathbf{h}_t\mathbf{w}_t^\top - \mathbf{\Delta}_t^2\mathbf{w}_t\bar{\mathbf{w}}_{*,t}^\top\mathbf{B}_*^\top\mathbf{B}_t + \mathbf{\Delta}_t\mathbf{w}_t\bar{\mathbf{w}}_{*,t}^\top\mathbf{B}_*^\top\mathbf{B}_t \\
&= -\mathbf{\Delta}_t^2\mathbf{B}_t^\top\mathbf{B}_*\bar{\mathbf{w}}_{*,t}\mathbf{w}_t^\top + \mathbf{B}_t^\top\mathbf{B}_t\mathbf{h}_t\mathbf{w}_t^\top - \mathbf{\Delta}_t^2\mathbf{w}_t\bar{\mathbf{w}}_{*,t}^\top\mathbf{B}_*^\top\mathbf{B}_t \\
&\quad + \mathbf{\Delta}_t(\alpha\mathbf{\Delta}_t\mathbf{B}_t^\top\mathbf{B}_*\bar{\mathbf{w}}_{*,t} + \mathbf{h}_t)\bar{\mathbf{w}}_{*,t}^\top\mathbf{B}_*^\top\mathbf{B}_t + \alpha\mathbf{\Delta}_t\mathbf{B}_t^\top\mathbf{B}_*\bar{\mathbf{w}}_{*,t}\bar{\mathbf{w}}_{*,t}^\top\mathbf{B}_*^\top\mathbf{B}_t. \qquad (32)
\end{aligned}
$$

The dominant term in (32) is the last term. Specifically, we have,

$$\|\mathbf{E}_1 - \alpha \boldsymbol{\Delta}_t \mathbf{B}_t^\top \mathbf{B}_* \bar{\mathbf{w}}_{*,t} \bar{\mathbf{w}}_{*,t}^\top \mathbf{B}_*^\top \mathbf{B}_t\|$$

$$\leq \|\boldsymbol{\Delta}_t^2 \mathbf{B}_t^\top \mathbf{B}_* \bar{\mathbf{w}}_{*,t} \mathbf{w}_t^\top\| + \|\mathbf{B}_t^\top \mathbf{B}_t \mathbf{h}_t \mathbf{w}_t^\top\| + \|\boldsymbol{\Delta}_t^2 \mathbf{w}_t \bar{\mathbf{w}}_{*,t}^\top \mathbf{B}_*^\top \mathbf{B}_t\|$$

$$\qquad + \|\boldsymbol{\Delta}_t(\alpha \boldsymbol{\Delta}_t \mathbf{B}_t^\top \mathbf{B}_* \bar{\mathbf{w}}_{*,t} + \mathbf{h}_t) \bar{\mathbf{w}}_{*,t}^\top \mathbf{B}_*^\top \mathbf{B}_t\|$$

$$\leq \|\boldsymbol{\Delta}_t^2 \mathbf{B}_t^\top \mathbf{B}_* \bar{\mathbf{w}}_{*,t} \mathbf{w}_t^\top\| + \|\boldsymbol{\Delta}_t^2 \mathbf{w}_t \bar{\mathbf{w}}_{*,t}^\top \mathbf{B}_*^\top \mathbf{B}_t\| + \alpha \|\boldsymbol{\Delta}_t^2 \mathbf{B}_t^\top \mathbf{B}_* \bar{\mathbf{w}}_{*,t} \bar{\mathbf{w}}_{*,t}^\top \mathbf{B}_*^\top \mathbf{B}_t\|$$

$$\qquad + \|\mathbf{B}_t^\top \mathbf{B}_t \mathbf{h}_t \mathbf{w}_t^\top\| + \|\boldsymbol{\Delta}_t \mathbf{h}_t \bar{\mathbf{w}}_{*,t}^\top \mathbf{B}_*^\top \mathbf{B}_t\|$$

$$\leq 5.5 c_3^2 \alpha^4 \tau^2 L_{\max}^6 \kappa_{\max}^4 E_0^{-2} + 2.2 \times 91 \alpha^2 \tau L_{\max}^4 + 1.1 \times 91 c_3 \alpha^4 \tau^2 L_{\max}^6 \kappa_{\max}^2 E_0^{-1} \quad (33)$$

$$\leq (206 + 101/c_3) \alpha^2 \tau L_{\max}^4 \quad (34)$$

where (33) follows by applying the Cauchy-Schwarz inequality to each of the terms in the previous inequality, and using $A_2(t)$, $A_3(t)$, and our bound on $\mathbf{h}_t$ (from $A_1(t)$), and (34) follows as $\alpha$ is sufficiently small. The last first term term can be subsumed by completing a square as follows. Combining (29), (30) and (32) yields

$$\mathbf{B}_t^\top \mathbf{G}_t = -\alpha \boldsymbol{\Delta}_t \mathbf{B}_t^\top \mathbf{B}_* \frac{1}{m} \sum_{i \in \mathcal{I}_t} \mathbf{w}_{*,i} \mathbf{w}_{*,i}^\top \mathbf{B}_*^\top \mathbf{B}_t + \mathbf{E}_1 - \mathbf{E}_2 + \mathbf{E}_3 + \mathbf{E}_4$$

$$= -\alpha \boldsymbol{\Delta}_t \mathbf{B}_t^\top \mathbf{B}_* \frac{1}{m} \sum_{i \in \mathcal{I}_t} \mathbf{w}_{*,i} \mathbf{w}_{*,i}^\top \mathbf{B}_*^\top \mathbf{B}_t + \alpha \boldsymbol{\Delta}_t \mathbf{B}_t^\top \mathbf{B}_* \bar{\mathbf{w}}_{*,t} \bar{\mathbf{w}}_{*,t}^\top \mathbf{B}_*^\top \mathbf{B}_t$$

$$\qquad + (\mathbf{E}_1 - \alpha \boldsymbol{\Delta}_t \mathbf{B}_t^\top \mathbf{B}_* \bar{\mathbf{w}}_{*,t} \bar{\mathbf{w}}_{*,t}^\top \mathbf{B}_*^\top \mathbf{B}_t) - \mathbf{E}_2 + \mathbf{E}_3 + \mathbf{E}_4$$

$$= -\alpha \boldsymbol{\Delta}_t \mathbf{B}_t^\top \mathbf{B}_* \frac{1}{m} \sum_{i \in \mathcal{I}_t} (\mathbf{w}_{*,i} - \bar{\mathbf{w}}_{*,t})(\mathbf{w}_{*,i} - \bar{\mathbf{w}}_{*,t})^\top \mathbf{B}_*^\top \mathbf{B}_t$$

$$\qquad + (\mathbf{E}_1 - \alpha \boldsymbol{\Delta}_t \mathbf{B}_t^\top \mathbf{B}_* \bar{\mathbf{w}}_{*,t} \bar{\mathbf{w}}_{*,t}^\top \mathbf{B}_*^\top \mathbf{B}_t) - \mathbf{E}_2 + \mathbf{E}_3 + \mathbf{E}_4$$

$$= -\tfrac{1}{2\alpha^2} \boldsymbol{\Delta}_t \mathbf{P}_t + \tfrac{1}{\alpha^2} \mathbf{Z}_t' \quad (35)$$

where $\mathbf{P}_t = 2\alpha^3 \mathbf{B}_t^\top \mathbf{B}_* \frac{1}{m} \sum_{i \in \mathcal{I}_t} (\mathbf{w}_{*,i} - \bar{\mathbf{w}}_{*,t})(\mathbf{w}_{*,i} - \bar{\mathbf{w}}_{*,t})^\top \mathbf{B}_*^\top \mathbf{B}_t$ and

$$\mathbf{Z}_t' := \alpha^2 (\mathbf{E}_1 - \alpha \boldsymbol{\Delta}_t \mathbf{B}_t^\top \mathbf{B}_* \bar{\mathbf{w}}_{*,t} \bar{\mathbf{w}}_{*,t}^\top \mathbf{B}_*^\top \mathbf{B}_t) - \alpha^2 \mathbf{E}_2 + \alpha^2 \mathbf{E}_3 + \alpha^2 \mathbf{E}_4$$

we have performed the desired decomposition; it remains to show that $\lambda_{\min}(\mathbf{P}_t)$ and $\|\mathbf{Z}_t'\|_2$ scale appropriately. First we lower bound $\lambda_{\min}(\mathbf{P}_t)$.

$$\lambda_{\min}(\mathbf{P}_t) = \lambda_{\min}\left(2\alpha^3 \mathbf{B}_t^\top \mathbf{B}_* \frac{1}{m} \sum_{i \in \mathcal{I}_t} (\mathbf{w}_{*,i} - \bar{\mathbf{w}}_{*,t})(\mathbf{w}_{*,i} - \bar{\mathbf{w}}_{*,t})^\top \mathbf{B}_*^\top \mathbf{B}_t\right)$$

$$\geq 2\alpha^3 \sigma_{\min}(\mathbf{B}_t^\top \mathbf{B}_*)^2 \lambda_{\min}\left(\frac{1}{m} \sum_{i \in \mathcal{I}_t} (\mathbf{w}_{*,i} - \bar{\mathbf{w}}_{*,t})(\mathbf{w}_{*,i} - \bar{\mathbf{w}}_{*,t})^\top\right)$$

$$\geq 0.2\alpha^2 E_0 \lambda_{\min}\left(\frac{1}{m} \sum_{i \in \mathcal{I}_t} (\mathbf{w}_{*,i} - \bar{\mathbf{w}}_{*,t})(\mathbf{w}_{*,i} - \bar{\mathbf{w}}_{*,t})^\top\right) \quad (36)$$

$$\geq 0.2\alpha^2 E_0 \lambda_{\min}\left(\frac{1}{M} \sum_{i=1}^{M} (\mathbf{w}_{*,i} - \bar{\mathbf{w}}_*)(\mathbf{w}_{*,i} - \bar{\mathbf{w}}_*)^\top\right)$$

$$\qquad - 0.2\alpha^2 E_0 \left\|\frac{1}{m} \sum_{i \in \mathcal{I}_t} \mathbf{w}_{*,i} \mathbf{w}_{*,i}^\top - \frac{1}{M} \sum_{i'=1}^{M} \mathbf{w}_{*,i'} \mathbf{w}_{*,i'}^\top\right\|$$

$$\qquad - 0.2\alpha^2 E_0 \left\|\mathbf{w}_{*,t} \mathbf{w}_{*,t}^\top - \bar{\mathbf{w}}_* \bar{\mathbf{w}}_*^\top\right\|$$

$$\geq 0.15\alpha^2 E_0 \mu^2 - 6\alpha^4 E_0 L_{\max}^4 \quad (37)$$

$$\geq 0.15\alpha^2 E_0 \mu^2 \quad (38)$$

where (36) follows by Lemma 8, (37) follows by Assumption 2 and $A_0$, and (38) follows as $\alpha^2 \leq \frac{1}{120\kappa_*^2 L_{\max}^2}$. Now we upper bound $\|\mathbf{Z}_t'\|_2$. We have already upper bounded $\|\mathbf{E}_1 -$

$\alpha \boldsymbol{\Delta}_t \mathbf{B}_t^\top \mathbf{B}_* \bar{\mathbf{w}}_{*,t} \bar{\mathbf{w}}_{*,t}^\top \mathbf{B}_*^\top \mathbf{B}_t\|$ in (34). We next upper bound $\|\mathbf{E}_2\|_2$ and $\|\mathbf{E}_3\|_2$ by $A_2(t), A_3(t)$, and $A_{2,i,t}(1)$. We have

$$\|\alpha^2 \mathbf{E}_2\|_2 \le 32\alpha^4 L_{\max}^4$$
$$\|\alpha^2 \mathbf{E}_3\|_2 \le 28c_3^2 \alpha^6 \tau^2 L_{\max}^6 \kappa_{\max}^2 E_0^{-2}$$

using the triangle and Cauchy-Schwarz inequalities. Now we turn to $\|\alpha^2 \mathbf{E}_4\|_2$. Recall that $\mathbf{E}_4$ is the sum of local gradients across all clients and all local updates beyond the first local update. We show that these gradients are sufficiently small such that $\|\boldsymbol{\Delta}_{t+1}\|_2$ cannot grow beyond the desired threshold.

Recall that $\mathbf{E}_4 = \sum_{s=2}^{\tau-1} \mathbf{B}_t^\top \mathbf{e}_{t,i,s} \mathbf{w}_{t,i,s}^\top$. To bound this sum it is critical to control the evolution of $\mathbf{e}_{t,i,s}$. The idea is to split $\mathbf{e}_{t,i,s}$ into its projection onto $\mathrm{col}(\mathbf{B}_{t,i,s-1}) \approx \mathrm{col}(\mathbf{B}_t)$ and its projection onto $\mathrm{col}(\mathbf{B}_{t,i,s-1})^\perp \approx \mathrm{col}(\mathbf{B}_t)^\perp$. Then, we can show that the magnitude of the projection onto $\mathrm{col}(\mathbf{B}_{t,i,s-1})$ is going to zero very fast (the head is quickly learned, meaning it fits the product as much as it can with what it has to work with, i.e. $\mathrm{col}(\mathbf{B}_{t,i,s-1})$). On the other hand, the magnitude of the projection onto $\mathrm{col}(\mathbf{B}_{t,i,s-1})^\perp$ is slowly going to zero, since this reducing this error requires changing the representation and the representation changes slower than the head. The saving grace is that this error is proportional to $\mathrm{dist}(\mathbf{B}_{t,i,s-1}, \mathbf{B}_*)$, which for all $s$ is linearly converging to zero with $t$.

To show this, pick any $i \in \mathcal{I}_t$ and let $\hat{\mathbf{B}}_{t,i,s}, \mathbf{R}_{t,i,s}$ denote the QR-factorization of $\mathbf{B}_{t,i,s}$. Define $\tilde{\boldsymbol{\Delta}}_{t,i,s-1} := \hat{\mathbf{B}}_{t,i,s-1} \hat{\mathbf{B}}_{t,i,s-1}^\top - \alpha \mathbf{B}_{t,i,s-1} \mathbf{B}_{t,i,s-1}^\top$ and $\omega_{t,i,s-1} := \alpha \mathbf{w}_{t,i,s-1}^\top \boldsymbol{\Delta}_{t,i,s-1} \mathbf{w}_{t,i,s} + \alpha^2 \mathbf{w}_{t,i,s-1}^\top \mathbf{B}_{t,i,s-1}^\top \mathbf{B}_* \mathbf{w}_{*,i}$. By expanding $\mathbf{e}_{t,i,s}$, we find

$\mathbf{e}_{t,i,s}$
$$= (\mathbf{I}_d - \alpha \mathbf{B}_{t,i,s-1} \mathbf{B}_{t,i,s-1}^\top - \alpha \mathbf{w}_{t,i,s-1}^\top \boldsymbol{\Delta}_{t,i,s-1} \mathbf{w}_{t,i,s} \mathbf{I}_d - \alpha^2 \mathbf{w}_{t,i,s-1}^\top \mathbf{B}_{t,i,s-1}^\top \mathbf{B}_* \mathbf{w}_{*,i} \mathbf{I}_d) \mathbf{e}_{t,i,s-1}$$
$$= (\mathbf{I}_d - \hat{\mathbf{B}}_{t,i,s-1} \hat{\mathbf{B}}_{t,i,s-1}^\top) \mathbf{e}_{t,i,s-1} + (\tilde{\boldsymbol{\Delta}}_{t,i,s-1} - \omega_{t,i,s-1} \mathbf{I}_d) \mathbf{e}_{t,i,s-1}$$
$$= (\mathbf{I}_d - \hat{\mathbf{B}}_{t,i,s-1} \hat{\mathbf{B}}_{t,i,s-1}^\top) \mathbf{e}_{t,i,s-1} + (\tilde{\boldsymbol{\Delta}}_{t,i,s-1} - \omega_{t,i,s-1} \mathbf{I}_d)(\mathbf{I}_d - \hat{\mathbf{B}}_{t,i,s-2} \hat{\mathbf{B}}_{t,i,s-2}^\top) \mathbf{e}_{t,i,s-2}$$
$$\quad + (\tilde{\boldsymbol{\Delta}}_{t,i,s-1} - \omega_{t,i,s-1} \mathbf{I}_d)(\tilde{\boldsymbol{\Delta}}_{t,i,s-2} - \omega_{t,i,s-2} \mathbf{I}_d) \mathbf{e}_{t,i,s-2} \tag{39}$$

Therefore,
$$\mathbf{B}_t^\top \mathbf{e}_{t,i,s} = \mathbf{B}_t^\top (\mathbf{I}_d - \hat{\mathbf{B}}_{t,i,s-1} \hat{\mathbf{B}}_{t,i,s-1}^\top) \mathbf{e}_{t,i,s-1}$$
$$\quad + \mathbf{B}_t^\top (\tilde{\boldsymbol{\Delta}}_{t,i,s-1} - \omega_{t,i,s-1} \mathbf{I}_d)(\mathbf{I}_d - \hat{\mathbf{B}}_{t,i,s-2} \hat{\mathbf{B}}_{t,i,s-2}^\top) \mathbf{e}_{t,i,s-2}$$
$$\quad + \mathbf{B}_t^\top (\tilde{\boldsymbol{\Delta}}_{t,i,s-1} - \omega_{t,i,s-1} \mathbf{I}_d)(\tilde{\boldsymbol{\Delta}}_{t,i,s-2} - \omega_{t,i,s-2} \mathbf{I}_d) \mathbf{e}_{t,i,s-2} \tag{40}$$

For the first term, we have
$$\|\mathbf{B}_t^\top (\mathbf{I}_d - \hat{\mathbf{B}}_{t,i,s-1} \hat{\mathbf{B}}_{t,i,s-1}^\top) \mathbf{e}_{t,i,s-1}\|_2$$
$$\le \|\mathbf{B}_{t,i,s-1}^\top (\mathbf{I}_d - \hat{\mathbf{B}}_{t,i,s-1} \hat{\mathbf{B}}_{t,i,s-1}^\top) \mathbf{e}_{t,i,s-1}\|_2$$
$$\quad + \|(\mathbf{B}_t - \mathbf{B}_{t,i,s-1})^\top (\mathbf{I}_d - \hat{\mathbf{B}}_{t,i,s-1} \hat{\mathbf{B}}_{t,i,s-1}^\top) \mathbf{e}_{t,i,s-1}\|_2$$
$$= \|(\mathbf{B}_t - \mathbf{B}_{t,i,s-1})^\top (\mathbf{I}_d - \hat{\mathbf{B}}_{t,i,s-1} \hat{\mathbf{B}}_{t,i,s-1}^\top) \mathbf{e}_{t,i,s-1}\|_2 \tag{41}$$
$$\le \|(\mathbf{I}_d - \hat{\mathbf{B}}_{t,i,s-1} \hat{\mathbf{B}}_{t,i,s-1}^\top) \mathbf{e}_{t,i,s-1}\|_2 \sum_{r=1}^{s-1} \|\mathbf{B}_{t,i,r} - \mathbf{B}_{t,i,r-1}\|_2 \tag{42}$$
$$\le 7\|(\mathbf{I}_d - \hat{\mathbf{B}}_{t,i,s-1} \hat{\mathbf{B}}_{t,i,s-1}^\top) \mathbf{B}_* \mathbf{w}_{*,i}\|_2 \alpha^{1.5} \tau L_{\max}^2 \tag{43}$$
$$\le 8\alpha^{1.5} \tau L_{\max}^3 \mathrm{dist}_t \tag{44}$$

where (41) follows since $\mathbf{B}_{t,i,s-1}^\top (\mathbf{I}_d - \hat{\mathbf{B}}_{t,i,s-1} \hat{\mathbf{B}}_{t,i,s-1}^\top) = \mathbf{0}$, (42) follows using the Cauchy-Schwarz and triangle inequalities, (43) follows using that $(\mathbf{I}_d - \hat{\mathbf{B}}_{t,i,s-1} \hat{\mathbf{B}}_{t,i,s-1}^\top) \mathbf{B}_{t,i,s-1} = \mathbf{0}$ and applying $A_{2,t,i}(\tau)$ and $A_{3,t,i}(\tau)$, (44) follows by the fact that $\|(\mathbf{I}_d - \hat{\mathbf{B}}_{t,i,s-1} \hat{\mathbf{B}}_{t,i,s-1}^\top) \mathbf{B}_*\| = \mathrm{dist}(\mathbf{B}_{t,i,s}, \mathbf{B}_*) \le 1.1 \mathrm{dist}_t$ by $A_{4,t,i}(\tau)$. For the second term in (40), note that
$$|\omega_{t,i,s-1}| \le \alpha |\mathbf{w}_{t,i,s-1}^\top \boldsymbol{\Delta}_{t,i,s-1} \mathbf{w}_{t,i,s-1}| + \alpha^2 |\mathbf{w}_{t,i,s-1}^\top \mathbf{B}_{t,i,s-1}^\top \mathbf{B}_* \mathbf{w}_{*,i}|$$
$$\le 8c_3 \alpha^4 \tau L_{\max}^4 \kappa_{\max}^2 E_0^{-1} + 2.2\alpha^{1.5} L_{\max}^2$$
$$\le 3\alpha^2 L_{\max}^2 \tag{45}$$

As a result, we have

$$\|\mathbf{B}_t^\top(\tilde{\boldsymbol{\Delta}}_{t,i,s-1} - \omega_{t,i,s-1}\mathbf{I}_d)(\mathbf{I}_d - \hat{\mathbf{B}}_{t,i,s-2}\hat{\mathbf{B}}_{t,i,s-2}^\top)\mathbf{e}_{t,i,s-2}\|_2$$

$$\leq \|\mathbf{B}_t^\top\tilde{\boldsymbol{\Delta}}_{t,i,s-1}(\mathbf{I}_d - \hat{\mathbf{B}}_{t,i,s-2}\hat{\mathbf{B}}_{t,i,s-2}^\top)\mathbf{e}_{t,i,s-2}\|_2$$

$$+ |\omega_{t,i,s-1}|\|\mathbf{B}_t^\top(\mathbf{I}_d - \hat{\mathbf{B}}_{t,i,s-2}\hat{\mathbf{B}}_{t,i,s-2}^\top)\mathbf{e}_{t,i,s-2}\|_2$$

$$\leq \|\mathbf{B}_t^\top\tilde{\boldsymbol{\Delta}}_{t,i,s-1}(\mathbf{I}_d - \hat{\mathbf{B}}_{t,i,s-2}\hat{\mathbf{B}}_{t,i,s-2}^\top)\mathbf{e}_{t,i,s-2}\|_2$$

$$+ 3.3\alpha^{1.5}L_{\max}^2\|(\mathbf{I}_d - \hat{\mathbf{B}}_{t,i,s-2}\hat{\mathbf{B}}_{t,i,s-2}^\top)\mathbf{B}_*\mathbf{w}_{*,i}\|_2$$

$$\leq \|\mathbf{B}_t^\top\tilde{\boldsymbol{\Delta}}_{t,i,s-1}(\mathbf{I}_d - \hat{\mathbf{B}}_{t,i,s-2}\hat{\mathbf{B}}_{t,i,s-2}^\top)\mathbf{e}_{t,i,s-2}\|_2$$

$$+ 3.7\alpha^{1.5}L_{\max}^3\operatorname{dist}_t \tag{46}$$

where (46) follows by $A_{4,t,i}(\tau)$, and

$$\|\mathbf{B}_t^\top\tilde{\boldsymbol{\Delta}}_{t,i,s-1}(\mathbf{I}_d - \hat{\mathbf{B}}_{t,i,s-2}\hat{\mathbf{B}}_{t,i,s-2}^\top)\mathbf{e}_{t,i,s-2}\|_2$$

$$\leq \|\mathbf{B}_t^\top\hat{\mathbf{B}}_{t,i,s-1}\hat{\mathbf{B}}_{t,i,s-1}^\top(\mathbf{I}_d - \hat{\mathbf{B}}_{t,i,s-2}\hat{\mathbf{B}}_{t,i,s-2}^\top)\mathbf{e}_{t,i,s-2}\|_2$$

$$+ \alpha\|\mathbf{B}_t^\top\mathbf{B}_{t,i,s-1}\mathbf{B}_{t,i,s-1}^\top(\mathbf{I}_d - \hat{\mathbf{B}}_{t,i,s-2}\hat{\mathbf{B}}_{t,i,s-2}^\top)\mathbf{e}_{t,i,s-2}\|_2$$

$$= \|\mathbf{B}_t^\top\hat{\mathbf{B}}_{t,i,s-1}(\mathbf{R}_{t,i,s-1}^{-1})^\top\mathbf{B}_{t,i,s-1}^\top(\mathbf{I}_d - \hat{\mathbf{B}}_{t,i,s-2}\hat{\mathbf{B}}_{t,i,s-2}^\top)\mathbf{e}_{t,i,s-2}\|_2$$

$$+ \alpha\|\mathbf{B}_t^\top\mathbf{B}_{t,i,s-1}\mathbf{B}_{t,i,s-1}^\top(\mathbf{I}_d - \hat{\mathbf{B}}_{t,i,s-2}\hat{\mathbf{B}}_{t,i,s-2}^\top)\mathbf{e}_{t,i,s-2}\|_2$$

$$= \alpha\|\mathbf{B}_t^\top\hat{\mathbf{B}}_{t,i,s-1}(\mathbf{R}_{t,i,s-1}^{-1})^\top\mathbf{w}_{t,i,s-2}\mathbf{e}_{t,i,s-2}^\top(\mathbf{I}_d - \hat{\mathbf{B}}_{t,i,s-2}\hat{\mathbf{B}}_{t,i,s-2}^\top)\mathbf{e}_{t,i,s-2}\|_2$$

$$+ \alpha^2\|\mathbf{B}_t^\top\mathbf{B}_{t,i,s-1}\mathbf{w}_{t,i,s-2}\mathbf{e}_{t,i,s-2}^\top(\mathbf{I}_d - \hat{\mathbf{B}}_{t,i,s-2}\hat{\mathbf{B}}_{t,i,s-2}^\top)\mathbf{e}_{t,i,s-2}\|_2 \tag{47}$$

$$\leq 44\alpha^{1.5}L_{\max}^3\operatorname{dist}_t \tag{48}$$

where (47) follows since $\mathbf{B}_{t,i,s-2}(\mathbf{I}_d - \hat{\mathbf{B}}_{t,i,s-2}\hat{\mathbf{B}}_{t,i,s-2}^\top) = \mathbf{0}$ and (48) follows using the Cauchy-Schwarz inequality, $A_3(t)$, $A_{2,t,i}(\tau)$, $A_{3,t,i}(\tau)$, and $A_{4,t,i}(\tau)$. Next, recalling that $\hat{\mathbf{B}}_{t,i,s}\mathbf{R}_{t,i,s}$ is the QR-factorization of $\mathbf{B}_{t,i,s}$, we have, for any $s$,

$$\|\tilde{\boldsymbol{\Delta}}_{t,i,s-1}\| = \|\hat{\mathbf{B}}_{t,i,s-1}\hat{\mathbf{B}}_{t,i,s-1}^\top - \alpha\mathbf{B}_{t,i,s-1}\mathbf{B}_{t,i,s-1}^\top\|$$

$$\leq \|\hat{\mathbf{B}}_{t,i,s-1}(\mathbf{I}_k - \alpha\mathbf{R}_{t,i,s-1}\mathbf{R}_{t,i,s-1}^\top)\hat{\mathbf{B}}_{t,i,s-1}^\top\|$$

$$\leq \|\mathbf{I}_k - \alpha\mathbf{R}_{t,i,s-1}\mathbf{R}_{t,i,s-1}^\top\|$$

$$\leq \max(|1 - \alpha\sigma_{\min}^2(\mathbf{B}_{t,i,s-1})|, |1 - \alpha\sigma_{\max}^2(\mathbf{B}_{t,i,s-1})|) \tag{49}$$

$$\leq \max(|1 - \alpha\tfrac{1-\|\boldsymbol{\Delta}_{t,i,s-1}\|}{\alpha}|, |1 - \alpha\tfrac{1+\|\boldsymbol{\Delta}_{t,i,s-1}\|}{\alpha}|)|)$$

$$= \|\boldsymbol{\Delta}_{t,i,s-1}\|$$

$$\leq 2c_3\alpha^2\tau L_{\max}^2\kappa_{\max}^2 E_0^{-1} \tag{50}$$

where (49) follows by Weyl's inequality and (50) follows by $A_{3,t,i}(s-1)$. Furthermore, $\|\tilde{\boldsymbol{\Delta}}_{t,i,s-1} + \omega_{t,i,s-1}\mathbf{I}_d\| \leq 2c_3\alpha^2\tau L_{\max}^2\kappa_{\max}^2 E_0^{-1} + 3\alpha^2 L_{\max}^2 \leq 3c_3\alpha^2\tau L_{\max}^2\kappa_{\max}^2 E_0^{-1}$ for any $s$. Thus, for the

third term in (40), for any $s > 2$,

$$\|\mathbf{B}_t^\top (\tilde{\boldsymbol{\Delta}}_{t,i,s-1} - \omega_{t,i,s-1}\mathbf{I}_d)(\tilde{\boldsymbol{\Delta}}_{t,i,s-2} - \omega_{t,i,s-2}\mathbf{I}_d)\mathbf{e}_{t,i,s-2}\|_2$$

$$= \|\mathbf{B}_t^\top (\tilde{\boldsymbol{\Delta}}_{t,i,s-1} - \omega_{t,i,s-1}\mathbf{I}_d)(\tilde{\boldsymbol{\Delta}}_{t,i,s-2} - \omega_{t,i,s-2}\mathbf{I}_d)(\tilde{\boldsymbol{\Delta}}_{t,i,s-3} - \omega_{t,i,s-3}\mathbf{I}_d)\mathbf{e}_{t,i,s-3}\|_2$$

$$+ \|\mathbf{B}_t^\top (\tilde{\boldsymbol{\Delta}}_{t,i,s-1} - \omega_{t,i,s-1}\mathbf{I}_d)(\tilde{\boldsymbol{\Delta}}_{t,i,s-2} - \omega_{t,i,s-2}\mathbf{I}_d)(\mathbf{I}_d - \hat{\mathbf{B}}_{t,i,s-3}\hat{\mathbf{B}}_{t,i,s-3}^\top)\mathbf{e}_{t,i,s-3}\|_2$$

$$\leq \|\mathbf{B}_t^\top (\tilde{\boldsymbol{\Delta}}_{t,i,s-1} - \omega_{t,i,s-1}\mathbf{I}_d)(\tilde{\boldsymbol{\Delta}}_{t,i,s-2} - \omega_{t,i,s-2}\mathbf{I}_d)(\tilde{\boldsymbol{\Delta}}_{t,i,s-3} - \omega_{t,i,s-3}\mathbf{I}_d)\mathbf{e}_{t,i,s-3}\|_2$$

$$+ 10c_3^2\alpha^{3.5}\tau^2 L_{\max}^5 \kappa_{\max}^4 E_0^{-2} \, \mathrm{dist}_t$$

$$\vdots$$

$$\leq \left\| \mathbf{B}_t^\top \prod_{r=1}^{s} (\tilde{\boldsymbol{\Delta}}_{t,i,s-r} - \omega_{t,i,s-r}\mathbf{I}_d)\mathbf{e}_{t,i,s-r} \right\|_2$$

$$+ 10c_3^2\alpha^{3.5}\tau^2 L_{\max}^5 \kappa_{\max}^4 E_0^{-2} \, \mathrm{dist}_t \sum_{r=0}^{s} (3c_3)^r \alpha^{2r}\tau^r L_{\max}^{2r} \kappa_{\max}^{2r} E_0^{-r}$$

$$\leq \left\| \mathbf{B}_t^\top \prod_{r=1}^{s} (\tilde{\boldsymbol{\Delta}}_{t,i,s-r} - \omega_{t,i,s-r}\mathbf{I}_d)\mathbf{e}_{t,i,s-r} \right\|_2 + \frac{10c_3^2\alpha^{3.5}\tau^2 L_{\max}^5 \kappa_{\max}^4 E_0^{-2} \, \mathrm{dist}_t}{1 - 3c_3\alpha^2\tau L_{\max}^2 \kappa_{\max}^2 E_0^{-1}}$$

$$\leq 3.5 \times (3c_3)^s \alpha^{2s-0.5}\tau^s L_{\max}^{2s+1}\kappa_{\max}^{2s} E_0^{-s} + 15c_3^2\alpha^{3.5}\tau^2 L_{\max}^5 \kappa_{\max}^4 E_0^{-2} \, \mathrm{dist}_t \qquad (51)$$

and for $s = 2$ we have

$$\|\mathbf{B}_t^\top (\tilde{\boldsymbol{\Delta}}_{t,i,s-1} - \omega_{t,i,s-1}\mathbf{I}_d)(\tilde{\boldsymbol{\Delta}}_{t,i,s-2} - \omega_{t,i,s-2}\mathbf{I}_d)\mathbf{e}_{t,i,s-2}\|_2$$

$$\leq 3.5 \times (3c_3)^2 \alpha^{3.5}\tau^2 L_{\max}^5 \kappa_{\max}^4 E_0^{-2}. \qquad (52)$$

Thus, using $\|\mathbf{w}_{t,i,s}\|_2 \leq 2\sqrt{\alpha}L_{\max}$ with (40), (44), (46), (48), (51), and (52), we have

$$\|\alpha^2\mathbf{E}_4\|_2 \leq 2\alpha^{2.5}L_{\max} \sum_{s=2}^{\tau-1} \left( 3.5 \times (3c_3)^2 \alpha^{2s-0.5}\tau^s L_{\max}^{2s+1}\kappa_{\max}^{2s} E_0^{-2s} \right.$$

$$\left. + 15c_3^2\alpha^{3.5}\tau^2 L_{\max}^5 \kappa_{\max}^4 \, \mathrm{dist}_t + (8\tau + 48)\alpha^{1.5}L_{\max}^3 \, \mathrm{dist}_t \right)$$

$$\leq \frac{63c_3^2\alpha^6\tau^2 L_{\max}^6 \kappa_{\max}^4 E_0^{-2}}{1 - 3c_s\alpha^2\tau L_{\max}^2 \kappa_{\max}^2 E_0^{-1}} + (30c_3^2\alpha^6\tau^3 L_{\max}^6 \kappa_{\max}^4 E_0^{-2} + 66\alpha^4\tau^2 L_{\max}^4) \, \mathrm{dist}_t$$

$$\leq 90c_3^2\alpha^6\tau^2 L_{\max}^6 \kappa_{\max}^4 E_0^{-2} + 94\alpha^4\tau^2 L_{\max}^4 \, \mathrm{dist}_t$$

where the last inequality follows by choice of $\alpha$. Combining all terms, we obtain

$$\|\boldsymbol{\Delta}_{t+1}\|_2 \leq (1 - 0.15\alpha^2 E_0\mu^2)\|\boldsymbol{\Delta}_t\|_2 + (206 + 101/c_3)\alpha^4\tau L_{\max}^4 + 32\alpha^4 L_{\max}^4$$

$$+ 118c_3^2\alpha^6\tau^2 L_{\max}^6 \kappa_{\max}^4 E_0^{-2} + 94\alpha^4\tau^2 L_{\max}^4 \, \mathrm{dist}_t$$

$$\leq (1 - 0.15\alpha^2 E_0\mu^2)\|\boldsymbol{\Delta}_t\|_2 + (340 + 101/c_3)\alpha^4\tau L_{\max}^4 + 94\alpha^4\tau^2 L_{\max}^4 \, \mathrm{dist}_t \qquad (53)$$

$$\vdots$$

$$\leq (1 - 0.15\alpha^2 E_0\mu^2)^t \|\boldsymbol{\Delta}_0\|_2$$

$$+ \sum_{t'=1}^{t} (1 - 0.15\alpha^2 E_0\mu^2)^{t-t'} \left( (340 + 101/c_3)\alpha^4\tau L_{\max}^4 + 94\alpha^4\tau^2 L_{\max}^4 \, \mathrm{dist}_{t'} \right)$$

$$\leq \|\boldsymbol{\Delta}_0\|_2 + (340 + 101/c_3)\alpha^4\tau L_{\max}^4 \sum_{t'=1}^{t} (1 - 0.15\alpha^2 E_0\mu^2)^{t-t'}$$

$$+ 94\alpha^4\tau^2 L_{\max}^4 \sum_{t'=1}^{t} (1 - 0.04\alpha^2\tau E_0\mu^2)^{t'} \qquad (54)$$

$$\leq \|\boldsymbol{\Delta}_0\|_2 + (7(340 + 101/c_3) + 25 \times 94)\alpha^2\tau L_{\max}^2 \kappa_{\max}^2 E_0^{-1}$$

$$\leq \alpha^2\tau L_{\max}^2 \kappa_{\max}^2 + (4730 + 101/c_3)\alpha^2\tau L_{\max}^2 \kappa_{\max}^2 E_0^{-1}$$

$$\leq c_3\alpha^2\tau L_{\max}^2 \kappa_{\max}^2 \qquad (55)$$

where (53) follows by choice of $\alpha \leq \frac{1-\delta_0}{c_3\sqrt{\tau}L_{\max}^2\kappa_{\max}^2} \leq \frac{E_0}{c_3\sqrt{\tau}L_{\max}^2\kappa_{\max}^2}$, (54) follows from $A_5(t)$, and the last inequality is due to $c_3 = 4800$.

$\square$

**Lemma 10.** $\cap_{i\in\mathcal{I}_t}\big(A_{1,t,i}(\tau) \cap A_{2,t,i}(\tau) \cap A_{3,t,i}(\tau)\big) \cap A_2(t) \implies A_4(t+1)$.

*Proof.* We have

$$\mathbf{B}_{t+1} = \mathbf{B}_t\left(\frac{1}{m}\sum_{i\in\mathcal{I}_t}\prod_{s=0}^{\tau-1}(\mathbf{I}_k - \alpha\mathbf{w}_{t,i,s}\mathbf{w}_{t,i,s}^\top)\right)$$

$$+ \hat{\mathbf{B}}_*\left(\frac{\alpha}{m}\sum_{i\in\mathcal{I}_t}\mathbf{w}_{*,i}\sum_{s=0}^{\tau-1}\mathbf{w}_{t,i,s}^\top\prod_{r=s+1}^{\tau-1}(\mathbf{I}_k - \alpha\mathbf{w}_{t,i,r}\mathbf{w}_{t,i,r}^\top)\right)$$

which implies

$$\mathbf{B}_{*,\perp}^\top\mathbf{B}_{t+1} = \mathbf{B}_{*,\perp}^\top\mathbf{B}_t(\mathbf{I} - \alpha\mathbf{w}_t\mathbf{w}_t^\top)\left(\frac{1}{m}\sum_{i\in\mathcal{I}_t}\prod_{s=1}^{\tau-1}(\mathbf{I}_k - \alpha\mathbf{w}_{t,i,s}\mathbf{w}_{t,i,s}^\top)\right) \tag{56}$$

We can expand the right product of $\mathbf{B}_t(\mathbf{I} - \alpha\mathbf{w}_t\mathbf{w}_t^\top)$ using the binomial expansion as follows:

$$\frac{1}{m}\sum_{i\in\mathcal{I}_t}\prod_{s=1}^{\tau-1}(\mathbf{I}_k - \alpha\mathbf{w}_{t,i,s}\mathbf{w}_{t,i,s}^\top) = \mathbf{I}_k - \frac{\alpha}{m}\sum_{i\in\mathcal{I}_t}\sum_{s=1}^{\tau-1}\mathbf{w}_{t,i,s}\mathbf{w}_{t,i,s}^\top$$

$$+ \frac{\alpha^2}{m}\sum_{i\in\mathcal{I}_t}\sum_{s=1}^{\tau-1}\sum_{s^{(1)}=s+1}^{\tau-1}\mathbf{w}_{t,i,s}\mathbf{w}_{t,i,s}^\top\mathbf{w}_{t,i,s^{(1)}}\mathbf{w}_{t,i,s^{(1)}}^\top$$

$$- \cdots + \text{sign}(\tau)\frac{\alpha^\tau}{m}\sum_{i\in\mathcal{I}_t}\prod_{s=1}^{\tau-1}\mathbf{w}_{t,i,s}\mathbf{w}_{t,i,s}^\top$$

Recall that each $\|\mathbf{w}_{t,i,s}\|_2 \leq 2\sqrt{\alpha}L_{\max}$. Thus, after the identity, the spectral norm of the first set of summations has spectral norm at most $4\alpha^2\tau L_{\max}^2$, the second set has spectral norm at most $16\alpha^4\tau^2 L_{\max}^4$, and so on. We in fact use the first set of summations as a negative term, and bound all subsequent sets of summations as errors, exploiting the fact that their norms are geometrically decaying. In particular, we have:

$$\left\|\frac{1}{m}\sum_{i\in\mathcal{I}_t}\prod_{s=1}^{\tau-1}(\mathbf{I}_k - \alpha\mathbf{w}_{t,i,s}\mathbf{w}_{t,i,s}^\top)\right\|_2 \leq \left\|\mathbf{I}_k - \frac{\alpha}{m}\sum_{i\in\mathcal{I}_t}\sum_{s=1}^{\tau-1}\mathbf{w}_{t,i,s}\mathbf{w}_{t,i,s}^\top\right\|_2 + \sum_{z=2}^{\tau-1}(4\alpha^2\tau L_{\max}^2)^z$$

$$\leq \left\|\mathbf{I}_k - \frac{\alpha}{m}\sum_{i\in\mathcal{I}_t}\sum_{s=1}^{\tau-1}\mathbf{w}_{t,i,s}\mathbf{w}_{t,i,s}^\top\right\|_2 + \frac{4\alpha^4\tau^2 L_{\max}^4}{1 - 4\alpha^2\tau L_{\max}^2}$$

$$\leq \left\|\mathbf{I}_k - \frac{\alpha}{m}\sum_{i\in\mathcal{I}_t}\sum_{s=1}^{\tau-1}\mathbf{w}_{t,i,s}\mathbf{w}_{t,i,s}^\top\right\|_2 + 5\alpha^4\tau^2 L_{\max}^4 \tag{57}$$

Next, we use $\|\mathbf{w}_{t,i,s} - \alpha\mathbf{B}_{t,i,s-1}^\top\mathbf{B}_*\mathbf{w}_{*,i}\| \leq 4c_3\alpha^{2.5}\tau L_{\max}^3\kappa_{\max}^2 E_0^{-1}$ for all $s \geq 1$ to obtain

$$\left\|\mathbf{I}_k - \frac{\alpha}{m}\sum_{i\in\mathcal{I}_t}\sum_{s=1}^{\tau-1}\mathbf{w}_{t,i,s}\mathbf{w}_{t,i,s}^\top\right\|_2 \leq \left\|\mathbf{I}_k - \frac{\alpha^3}{m}\sum_{i\in\mathcal{I}_t}\sum_{s=1}^{\tau-1}\mathbf{B}_{t,i,s-1}^\top\mathbf{B}_*\mathbf{w}_{*,i}\mathbf{w}_{*,i}^\top\mathbf{B}_*^\top\mathbf{B}_{t,i,s-1}\right\|_2$$

$$+ 13c_3\alpha^4\tau^2 L_{\max}^4\kappa_{\max}^2 E_0^{-1} \tag{58}$$

Next, we have for any $s - 1 \in \{1, \dots, \tau - 1\}$,

$$
\begin{aligned}
\|\mathbf{B}_{t,i,s-1} - \mathbf{B}_t\|_2 = \|\mathbf{B}_{t,i,s-1} - \mathbf{B}_{t,i,0}\|_2 &\leq \sum_{r=1}^{s-1} \|\mathbf{B}_{t,i,r} - \mathbf{B}_{t,i,r-1}\|_2 \\
&\leq \alpha \sum_{r=1}^{s-1} \|\mathbf{e}_{t,i,r-1}\|_2 \|\mathbf{w}_{t,i,r-1}\|_2 \\
&\leq 7\alpha^{1.5}(s-1)L_{\max}^2
\end{aligned}
$$

Thus,

$$
\begin{aligned}
\Bigg\| \mathbf{I}_k - \frac{\alpha^3}{m} &\sum_{i \in \mathcal{I}_t} \sum_{s=1}^{\tau-1} \mathbf{B}_{t,i,s-1}^\top \mathbf{B}_* \mathbf{w}_{*,i} \mathbf{w}_{*,i}^\top \mathbf{B}_*^\top \mathbf{B}_{t,i,s-1} \Bigg\|_2 \\
&\leq \Bigg\| \mathbf{I}_k - \frac{\alpha^3}{m} \sum_{i \in \mathcal{I}_t} \sum_{s=1}^{\tau-1} \mathbf{B}_t^\top \mathbf{B}_* \mathbf{w}_{*,i} \mathbf{w}_{*,i}^\top \mathbf{B}_*^\top \mathbf{B}_t \Bigg\|_2 \\
&\quad + \Bigg\| \frac{\alpha^3}{m} \sum_{i \in \mathcal{I}_t} \sum_{s=1}^{\tau-1} (\mathbf{B}_t - \mathbf{B}_{t,i,s-1})^\top \mathbf{B}_* \mathbf{w}_{*,i} \mathbf{w}_{*,i}^\top \mathbf{B}_*^\top \mathbf{B}_{t,i,s-1} \Bigg\|_2 \\
&\quad + \Bigg\| \frac{\alpha^3}{m} \sum_{i \in \mathcal{I}_t} \sum_{s=1}^{\tau-1} \mathbf{B}_t^\top \mathbf{B}_* \mathbf{w}_{*,i} \mathbf{w}_{*,i}^\top \mathbf{B}_*^\top (\mathbf{B}_t - \mathbf{B}_{t,i,s-1}) \Bigg\|_2 \\
&\leq \Bigg\| \mathbf{I}_k - \frac{\alpha^3}{m} \sum_{i \in \mathcal{I}_t} \sum_{s=1}^{\tau-1} \mathbf{B}_t^\top \mathbf{B}_* \mathbf{w}_{*,i} \mathbf{w}_{*,i}^\top \mathbf{B}_*^\top \mathbf{B}_t \Bigg\|_2 + 16\alpha^4(\tau-1)^2 L_{\max}^4. \tag{59}
\end{aligned}
$$

Furtheromre,

$$
\begin{aligned}
\Bigg\| \mathbf{I}_k - \frac{\alpha^3}{m} &\sum_{i \in \mathcal{I}_t} \sum_{s=1}^{\tau-1} \mathbf{B}_t^\top \mathbf{B}_* \mathbf{w}_{*,i} \mathbf{w}_{*,i}^\top \mathbf{B}_*^\top \mathbf{B}_t \Bigg\|_2 \\
&= \Bigg\| \mathbf{I}_k - \frac{\alpha^3(\tau-1)}{m} \sum_{i \in \mathcal{I}_t} \mathbf{B}_t^\top \mathbf{B}_* \mathbf{w}_{*,i} \mathbf{w}_{*,i}^\top \mathbf{B}_*^\top \mathbf{B}_t \Bigg\|_2 \\
&\leq \Bigg\| \mathbf{I}_k - \frac{\alpha^3(\tau-1)}{M} \sum_{i=1}^{M} \mathbf{B}_t^\top \mathbf{B}_* \mathbf{w}_{*,i} \mathbf{w}_{*,i}^\top \mathbf{B}_*^\top \mathbf{B}_t \Bigg\|_2 \\
&\quad + \alpha^3(\tau-1)\|\mathbf{B}_t^\top \mathbf{B}_*\|_2^2 \Bigg\| \frac{1}{m} \sum_{i \in \mathcal{I}_t} \left( \mathbf{w}_{*,i} \mathbf{w}_{*,i}^\top - \frac{1}{M} \sum_{i=1}^{M} \mathbf{w}_{*,i} \mathbf{w}_{*,i}^\top \right) \Bigg\|_2 \\
&\leq \Bigg\| \mathbf{I}_k - \frac{\alpha^3(\tau-1)}{M} \sum_{i=1}^{M} \mathbf{B}_t^\top \mathbf{B}_* \mathbf{w}_{*,i} \mathbf{w}_{*,i}^\top \mathbf{B}_*^\top \mathbf{B}_t \Bigg\|_2 + 6\alpha^4(\tau-1)L_{\max}^4, \tag{60}
\end{aligned}
$$

noting that (60) follows since we are conditioning on the event $A_0$. Finally,

$$\left\| \mathbf{I}_k - \frac{\alpha^3(\tau-1)}{M} \sum_{i=1}^M \mathbf{B}_t^\top \mathbf{B}_* \mathbf{w}_{*,i} \mathbf{w}_{*,i}^\top \mathbf{B}_*^\top \mathbf{B}_t \right\|_2$$

$$\leq 1 - \alpha^3(\tau-1)\sigma_{\min}^2(\mathbf{B}_t^\top \mathbf{B}_*)\sigma_{\min}\left(\frac{1}{M}\sum_{i=1}^M \mathbf{w}_{*,i}\mathbf{w}_{*,i}^\top\right)$$

$$\leq 1 - \alpha^3(\tau-1)\sigma_{\min}^2(\mathbf{B}_t^\top \mathbf{B}_*)\sigma_{\min}\left(\frac{1}{M}\sum_{i=1}^M \mathbf{w}_{*,i}\mathbf{w}_{*,i}^\top - \bar{\mathbf{w}}_* \bar{\mathbf{w}}_*^\top\right)$$

$$= 1 - \alpha^3(\tau-1)\sigma_{\min}(\mathbf{B}_t^\top \mathbf{B}_*)^2 \sigma_{\min}\left(\frac{1}{M}\sum_{i=1}^M (\mathbf{w}_{*,i} - \bar{\mathbf{w}}_*)(\mathbf{w}_{*,i} - \bar{\mathbf{w}}_*)^\top\right)$$

$$\leq 1 - 0.1\alpha^2(\tau-1)E_0\mu \tag{61}$$

$$\leq 1 - 0.05\alpha^2\tau E_0\mu \tag{62}$$

where (61) follows by Lemma 8 and Assumption 2, and (62) follows since $\tau \geq 2$. Combining (62), (60), (59), (58), (57), and (56), we obtain

$$\|\mathbf{B}_{*,\perp}^\top \mathbf{B}_{t+1}\|_2 \leq \left(1 - 0.05\alpha^2\tau E_0\mu^2 + 6\alpha^4(\tau-1)L_{\max}^4 + 16\alpha^4(\tau-1)^2 L_{\max}^4\right.$$

$$\left. + 13c_3\alpha^4\tau^2 L_{\max}^4 \kappa_{\max}^2 E_0^{-1} + 5\alpha^4\tau^2 L_{\max}^4\right)\|\mathbf{B}_{*,\perp}^\top \mathbf{B}_t\|_2$$

$$\leq \left(1 - 0.05\alpha^2\tau E_0\mu^2 + (24/c_3^2)\alpha^2\tau E_0\mu^2 + (13/c_3)\alpha^2\tau E_0\mu^2\right)\|\mathbf{B}_{*,\perp}^\top \mathbf{B}_t\|_2$$

$$\leq \left(1 - 0.04\alpha^2\tau E_0\mu^2\right)\|\mathbf{B}_{*,\perp}^\top \mathbf{B}_t\|_2 \tag{63}$$

using $\alpha^2\tau \leq \frac{(1-\delta_0)^2}{c_3^2\tau\kappa_{\max}^4 L_{\max}^2} \leq \frac{E_0^2}{c_3^3\tau\kappa_{\max}^4 L_{\max}^2}$ and $c_3 > 1305$. $\qquad\square$

**Lemma 11.** $A_3(t+1) \cap A_4(t+1) \cap A_5(t) \implies A_5(t+1)$

*Proof.* We use the contraction of $\|\mathbf{B}_{*,\perp}^\top \mathbf{B}_s\|_2$ ($A_4(t)$) and the fact that $\|\mathbf{\Delta}_s\|_2$ is small for all $s \in [t]$ ($A_5(t)$), as in Lemma 8, to obtain

$$\text{dist}_{t+1} = \|\mathbf{B}_{*,\perp}^\top \hat{\mathbf{B}}_{t+1}\|_2$$

$$\leq \frac{1}{\sigma_{\min}(\mathbf{B}_{t+1})}\|\mathbf{B}_{*,\perp}^\top \mathbf{B}_{t+1}\|_2$$

$$\leq \frac{1}{\sigma_{\min}(\mathbf{B}_{t+1})}(1 - 0.04\alpha^2\tau E_0\mu^2)\|\mathbf{B}_{*,\perp}^\top \mathbf{B}_t\|_2$$

$$\vdots$$

$$\leq \frac{1}{\sigma_{\min}(\mathbf{B}_{t+1})}(1 - 0.04\alpha^2\tau E_0\mu^2)^t\|\mathbf{B}_{*,\perp}^\top \mathbf{B}_0\|_2$$

$$\leq \frac{\sigma_{\max}(\mathbf{B}_0)}{\sigma_{\min}(\mathbf{B}_{t+1})}(1 - 0.04\alpha^2\tau E_0\mu^2)^t\delta_0$$

$$\leq \frac{\sqrt{1+\|\mathbf{\Delta}_0\|_2/\sqrt{\alpha}}}{\sqrt{1-\|\mathbf{\Delta}_{t+1}\|_2/\sqrt{\alpha}}}(1 - 0.04\alpha^2\tau E_0\mu^2)^t\delta_0$$

Now, we argue as in (26) (with $\|\mathbf{\Delta}_t\|$ replaced by $\|\mathbf{\Delta}_{t+1}\|$ without anything changing in the analysis) to find $\frac{\sqrt{1+\|\mathbf{\Delta}_0\|_2/\sqrt{\alpha}}}{\sqrt{1-\|\mathbf{\Delta}_{t+1}\|_2/\sqrt{\alpha}}}\delta_0 \leq \frac{2+\delta_0}{3} \leq 1$. Thus $\text{dist}_{t+1} \leq (1 - 0.04\alpha^2\tau E_0\mu^2)^t$ as desired. $\qquad\square$

### B.2   Proof of Proposition 1

**Proposition 2** (Distributed GD lower bound). *Suppose we are in the setting described in Section 3 and $d > k > 1$. Then for any set of ground-truth heads $\{\mathbf{w}_{*,i}\}_{i=1}^M$, full-rank initialization $\mathbf{B}_0 \in \mathbb{R}^{d\times k}$, initial distance $\delta_0 \in (0, 1/2]$, step size $\alpha > 0$, and number of rounds $T$, there exists $\mathbf{B}_* \in \mathcal{O}^{d\times k}$ satisfying $\text{dist}(\mathbf{B}_0, \mathbf{B}_*) = \delta_0$ and $\text{dist}(\mathbf{B}_T^{D\text{-}GD}, \mathbf{B}_*) \geq 0.7\delta_0$, where*

$\mathbf{B}_T^{D\text{-}GD} \equiv \mathbf{B}_T^{D\text{-}GD}(\mathbf{B}_0, \mathbf{B}_*, \{\mathbf{w}_{*,i}\}_{i=1}^M, \alpha)$ *is the result of D-GD with step size* $\alpha$ *and initialization* $\mathbf{B}_0$ *in the setting with ground-truth representation* $\mathbf{B}_*$ *and ground-truth heads* $\{\mathbf{w}_{*,i}\}_{i=1}^M$.

*Proof.* Recall that $\mathbf{B}_T^{D\text{-}GD}(\mathbf{B}_0, \mathbf{B}_*, \{\mathbf{w}_{*,i}\}_{i=1}^M, \alpha)$ is the result of D-GD with step size $\alpha$ and initialization $\mathbf{B}_0$ on the system with ground-truth representation $\mathbf{B}_*$ and ground-truth heads $\{\mathbf{w}_{*,i}\}_{i=1}^M$.

There are two disjoint cases: (1) for all $\mathbf{B}_* \in \mathcal{B} \coloneqq \{\mathbf{B} \in \mathcal{O}^{d \times k} : \mathrm{dist}(\mathbf{B}_0, \mathbf{B}) = \delta_0, \mathbf{B}\bar{\mathbf{w}}_* \in \mathrm{col}(\mathbf{B}_0)\}$, $\mathrm{dist}(\mathbf{B}_T^{D\text{-}GD}(\mathbf{B}_0, \mathbf{B}_*, \{\mathbf{w}_{*,i}\}_{i=1}^M, \alpha), \mathbf{B}_*) \geq 0.7\delta_0$, or (2) there exists some $\mathbf{B}_* \in \mathcal{B}$ such that $\mathrm{dist}(\mathbf{B}_T^{D\text{-}GD}(\mathbf{B}_0, \mathbf{B}_*, \{\mathbf{w}_{*,i}\}_{i=1}^M, \alpha), \mathbf{B}_*) < 0.7\delta_0$. If case (1) holds then the proof is complete. Otherwise, let $\mathbf{B}_* \in \mathcal{B}$ such that $\mathrm{dist}(\mathbf{B}_T^{D\text{-}GD}(\mathbf{B}_0, \mathbf{B}_*, \{\mathbf{w}_{*,i}\}_{i=1}^M, \alpha), \mathbf{B}_*) < 0.7\delta_0$. We will show that there exists another $\mathbf{B}_{*'} \in \mathcal{B}$ such that $\mathrm{dist}(\mathbf{B}_T^{D\text{-}GD}(\mathbf{B}_0, \mathbf{B}_{*'}, \{\mathbf{w}_{*,i}\}_{i=1}^M, \alpha), \mathbf{B}_{*'}) \geq \delta_0$, so D-GD cannot guarantee to recover the ground-truth representation, completing the proof.

Consider case (2). Without loss of generality we can write $\mathbf{B}_0 = \frac{1}{\|\bar{\mathbf{w}}_*\|}\mathbf{B}_*\bar{\mathbf{w}}_*\mathbf{v}_0^\top + \tilde{\mathbf{B}}_0\tilde{\mathbf{V}}_0^\top$ for some $\mathbf{v}_0 \in \mathbb{R}^k : \|\mathbf{v}_0\| = 1, \tilde{\mathbf{B}}_0 \in \mathcal{O}^{d \times k-1} : \tilde{\mathbf{B}}_0^\top\mathbf{B}_*\bar{\mathbf{w}}_* = \mathbf{0}, \tilde{\mathbf{V}}_0 \in \mathcal{O}^{k \times k-1} : \tilde{\mathbf{V}}_0^\top\mathbf{v}_0 = \mathbf{0}$ using the SVD, since $\mathbf{B}_*\bar{\mathbf{w}}_* \in \mathrm{col}(\mathbf{B}_0)$. Likewise, we can write $\mathbf{B}_* = \frac{1}{\|\bar{\mathbf{w}}_*\|^2}\mathbf{B}_*\bar{\mathbf{w}}_*\bar{\mathbf{w}}_*^\top + \tilde{\mathbf{B}}_*\tilde{\mathbf{V}}_*^\top$ for some $\tilde{\mathbf{B}}_* \in \mathcal{O}^{d \times k-1} : \tilde{\mathbf{B}}_*^\top\mathbf{B}_*\bar{\mathbf{w}}_* = \mathbf{0}, \tilde{\mathbf{V}}_* \in \mathcal{O}^{k \times k-1} : \tilde{\mathbf{V}}_*^\top\bar{\mathbf{w}}_* = \mathbf{0}$. Using these decompositions, we can see that

$$
\begin{aligned}
\mathbf{B}_*\mathbf{B}_*^\top &= \left(\tfrac{1}{\|\bar{\mathbf{w}}_*\|^2}\mathbf{B}_*\bar{\mathbf{w}}_*\bar{\mathbf{w}}_*^\top + \tilde{\mathbf{B}}_*\tilde{\mathbf{V}}_*^\top\right)\left(\tfrac{1}{\|\bar{\mathbf{w}}_*\|^2}\mathbf{B}_*\bar{\mathbf{w}}_*\bar{\mathbf{w}}_*^\top + \tilde{\mathbf{B}}_*\tilde{\mathbf{V}}_*^\top\right)^\top \\
&= \tfrac{1}{\|\bar{\mathbf{w}}_*\|^4}\mathbf{B}_*\bar{\mathbf{w}}_*\bar{\mathbf{w}}_*^\top\bar{\mathbf{w}}_*\bar{\mathbf{w}}_*^\top\mathbf{B}_*^\top + \tfrac{1}{\|\bar{\mathbf{w}}_*\|^2}\mathbf{B}_*\bar{\mathbf{w}}_*\bar{\mathbf{w}}_*^\top\tilde{\mathbf{V}}_*\tilde{\mathbf{B}}_*^\top + \tfrac{1}{\|\bar{\mathbf{w}}_*\|^2}\tilde{\mathbf{B}}_*\tilde{\mathbf{V}}_*^\top\bar{\mathbf{w}}_*\bar{\mathbf{w}}_*^\top\mathbf{B}_*^\top \\
&\quad + \tilde{\mathbf{B}}_*\tilde{\mathbf{V}}_*^\top\tilde{\mathbf{V}}_*\tilde{\mathbf{B}}_*^\top \\
&= \tfrac{1}{\|\bar{\mathbf{w}}_*\|^2}\mathbf{B}_*\bar{\mathbf{w}}_*\bar{\mathbf{w}}_*^\top\mathbf{B}_*^\top + \tilde{\mathbf{B}}_*\tilde{\mathbf{B}}_*^\top
\end{aligned}
$$

and

$$
\begin{aligned}
\delta_0 &\coloneqq \mathrm{dist}(\mathbf{B}_0, \mathbf{B}_*) \\
&\coloneqq \|(\mathbf{I}_d - \mathbf{B}_*\mathbf{B}_*^\top)\mathbf{B}_0\| \\
&= \left\|\left(\mathbf{I}_d - \tfrac{1}{\|\bar{\mathbf{w}}_*\|^2}\mathbf{B}_*\bar{\mathbf{w}}_*\bar{\mathbf{w}}_*^\top\mathbf{B}_*^\top - \tilde{\mathbf{B}}_*\tilde{\mathbf{B}}_*^\top\right)\left(\tfrac{1}{\|\bar{\mathbf{w}}_*\|}\mathbf{B}_*\bar{\mathbf{w}}_*\mathbf{v}_0^\top + \tilde{\mathbf{B}}_0\tilde{\mathbf{V}}_0^\top\right)\right\| \\
&= \left\|\left(\mathbf{I}_d - \tfrac{1}{\|\bar{\mathbf{w}}_*\|^2}\mathbf{B}_*\bar{\mathbf{w}}_*\bar{\mathbf{w}}_*^\top\mathbf{B}_*^\top - \tilde{\mathbf{B}}_*\tilde{\mathbf{B}}_*^\top\right)\tfrac{1}{\|\bar{\mathbf{w}}_*\|}\mathbf{B}_*\bar{\mathbf{w}}_*\mathbf{v}_0^\top \right. \\
&\quad \left. + \left(\mathbf{I}_d - \tfrac{1}{\|\bar{\mathbf{w}}_*\|^2}\mathbf{B}_*\bar{\mathbf{w}}_*\bar{\mathbf{w}}_*^\top\mathbf{B}_*^\top - \tilde{\mathbf{B}}_*\tilde{\mathbf{B}}_*^\top\right)\tilde{\mathbf{B}}_0\tilde{\mathbf{V}}_0^\top\right\| \\
&= \left\|\left(\mathbf{I}_d - \tfrac{1}{\|\bar{\mathbf{w}}_*\|^2}\mathbf{B}_*\bar{\mathbf{w}}_*\bar{\mathbf{w}}_*^\top\mathbf{B}_*^\top - \tilde{\mathbf{B}}_*\tilde{\mathbf{B}}_*^\top\right)\tilde{\mathbf{B}}_0\tilde{\mathbf{V}}_0^\top\right\| \\
&= \left\|\left(\mathbf{I}_d - \tilde{\mathbf{B}}_*\tilde{\mathbf{B}}_*^\top\right)\tilde{\mathbf{B}}_0\tilde{\mathbf{V}}_0^\top\right\| \\
&= \left\|\left(\mathbf{I}_d - \tilde{\mathbf{B}}_*\tilde{\mathbf{B}}_*^\top\right)\tilde{\mathbf{B}}_0\right\| \\
&= \mathrm{dist}(\tilde{\mathbf{B}}_0, \tilde{\mathbf{B}}_*). \quad\quad\quad (64)
\end{aligned}
$$

Next, let $\mathbf{B}_{*'} = \frac{1}{\|\bar{\mathbf{w}}_*\|^2}\mathbf{B}_*\bar{\mathbf{w}}_*\bar{\mathbf{w}}_*^\top + (2\tilde{\mathbf{B}}_0\tilde{\mathbf{B}}_0^\top\tilde{\mathbf{B}}_* - \tilde{\mathbf{B}}_*)\tilde{\mathbf{V}}_*^\top$. We first check that $\mathbf{B}_{*'} \in \mathcal{O}^{d \times k}$:

$$
\begin{aligned}
\mathbf{B}_{*'}^\top\mathbf{B}_{*'} &= \left(\tfrac{1}{\|\bar{\mathbf{w}}_*\|^2}\mathbf{B}_*\bar{\mathbf{w}}_*\bar{\mathbf{w}}_*^\top + (2\tilde{\mathbf{B}}_0\tilde{\mathbf{B}}_0^\top\tilde{\mathbf{B}}_* - \tilde{\mathbf{B}}_*)\tilde{\mathbf{V}}_*^\top\right)^\top\left(\tfrac{1}{\|\bar{\mathbf{w}}_*\|^2}\mathbf{B}_*\bar{\mathbf{w}}_*\bar{\mathbf{w}}_*^\top + (2\tilde{\mathbf{B}}_0\tilde{\mathbf{B}}_0^\top\tilde{\mathbf{B}}_* - \tilde{\mathbf{B}}_*)\tilde{\mathbf{V}}_*^\top\right) \\
&= \tfrac{1}{\|\bar{\mathbf{w}}_*\|^2}\bar{\mathbf{w}}_*\bar{\mathbf{w}}_*^\top + \tilde{\mathbf{V}}_*(2\tilde{\mathbf{B}}_0\tilde{\mathbf{B}}_0^\top\tilde{\mathbf{B}}_* - \tilde{\mathbf{B}}_*)^\top(2\tilde{\mathbf{B}}_0\tilde{\mathbf{B}}_0^\top\tilde{\mathbf{B}}_* - \tilde{\mathbf{B}}_*)\tilde{\mathbf{V}}_*^\top \quad\quad (65) \\
&= \tfrac{1}{\|\bar{\mathbf{w}}_*\|^2}\bar{\mathbf{w}}_*\bar{\mathbf{w}}_*^\top + \tilde{\mathbf{V}}_*(4\tilde{\mathbf{B}}_*^\top\tilde{\mathbf{B}}_0\tilde{\mathbf{B}}_0^\top\tilde{\mathbf{B}}_0\tilde{\mathbf{B}}_0^\top\tilde{\mathbf{B}}_* - 4\tilde{\mathbf{B}}_*^\top\tilde{\mathbf{B}}_0\tilde{\mathbf{B}}_0^\top\tilde{\mathbf{B}}_* + \tilde{\mathbf{B}}_*^\top\tilde{\mathbf{B}}_*)\tilde{\mathbf{V}}_*^\top \\
&= \tfrac{1}{\|\bar{\mathbf{w}}_*\|^2}\bar{\mathbf{w}}_*\bar{\mathbf{w}}_*^\top + \tilde{\mathbf{V}}_*\tilde{\mathbf{V}}_*^\top \\
&= [\tilde{\mathbf{V}}_*, \tfrac{1}{\|\bar{\mathbf{w}}_*\|}\bar{\mathbf{w}}_*][\tilde{\mathbf{V}}_*, \tfrac{1}{\|\bar{\mathbf{w}}_*\|}\bar{\mathbf{w}}_*]^\top \\
&= \mathbf{I}_k \quad\quad\quad (66)
\end{aligned}
$$

as desired, where (65) follows since $\tilde{\mathbf{B}}_*^\top \mathbf{B}_* \bar{\mathbf{w}}_* = \tilde{\mathbf{B}}_0^\top \mathbf{B}_* \bar{\mathbf{w}}_* = \mathbf{0}$, and (66) follows since $[\tilde{\mathbf{V}}_*, \frac{1}{\|\bar{\mathbf{w}}_*\|}\bar{\mathbf{w}}_*] \in \mathcal{O}^{k \times k}$ by the definition of the SVD. Furthermore,

$$
\begin{aligned}
\mathrm{dist}(\mathbf{B}_0, \mathbf{B}_{*'}) &= \|(\mathbf{I}_d - \mathbf{B}_0\mathbf{B}_0^\top)\mathbf{B}_{*'}\| \\
&= \|(\mathbf{I}_d - \tfrac{1}{\|\bar{\mathbf{w}}_*\|^2}\mathbf{B}_*\bar{\mathbf{w}}_*\bar{\mathbf{w}}_*^\top\mathbf{B}_*^\top - \tilde{\mathbf{B}}_0\tilde{\mathbf{B}}_0^\top)\big(\tfrac{1}{\|\bar{\mathbf{w}}_*\|^2}\mathbf{B}_*\bar{\mathbf{w}}_*\bar{\mathbf{w}}_*^\top + (2\tilde{\mathbf{B}}_0\tilde{\mathbf{B}}_0^\top\tilde{\mathbf{B}}_* - \tilde{\mathbf{B}}_*)\tilde{\mathbf{V}}_*^\top)\| \\
&= \|(\mathbf{I}_d - \tfrac{1}{\|\bar{\mathbf{w}}_*\|^2}\mathbf{B}_*\bar{\mathbf{w}}_*\bar{\mathbf{w}}_*^\top\mathbf{B}_*^\top - \tilde{\mathbf{B}}_0\tilde{\mathbf{B}}_0^\top)\tfrac{1}{\|\bar{\mathbf{w}}_*\|^2}\mathbf{B}_*\bar{\mathbf{w}}_*\bar{\mathbf{w}}_*^\top \\
&\quad + (\mathbf{I}_d - \tfrac{1}{\|\bar{\mathbf{w}}_*\|^2}\mathbf{B}_*\bar{\mathbf{w}}_*\bar{\mathbf{w}}_*^\top\mathbf{B}_*^\top - \tilde{\mathbf{B}}_0\tilde{\mathbf{B}}_0^\top)(2\tilde{\mathbf{B}}_0\tilde{\mathbf{B}}_0^\top\tilde{\mathbf{B}}_* - \tilde{\mathbf{B}}_*)\tilde{\mathbf{V}}_*^\top\| \\
&= \|(\mathbf{I}_d - \tfrac{1}{\|\bar{\mathbf{w}}_*\|^2}\mathbf{B}_*\bar{\mathbf{w}}_*\bar{\mathbf{w}}_*^\top\mathbf{B}_*^\top - \tilde{\mathbf{B}}_0\tilde{\mathbf{B}}_0^\top)(2\tilde{\mathbf{B}}_0\tilde{\mathbf{B}}_0^\top\tilde{\mathbf{B}}_* - \tilde{\mathbf{B}}_*)\tilde{\mathbf{V}}_*^\top\| \\
&= \|(\mathbf{I}_d - \tilde{\mathbf{B}}_0\tilde{\mathbf{B}}_0^\top)(2\tilde{\mathbf{B}}_0\tilde{\mathbf{B}}_0^\top\tilde{\mathbf{B}}_* - \tilde{\mathbf{B}}_*)\tilde{\mathbf{V}}_*^\top\| \\
&= \|(\mathbf{I}_d - \tilde{\mathbf{B}}_0\tilde{\mathbf{B}}_0^\top)\tilde{\mathbf{B}}_*\tilde{\mathbf{V}}_*^\top\| \\
&= \|(\mathbf{I}_d - \tilde{\mathbf{B}}_0\tilde{\mathbf{B}}_0^\top)\tilde{\mathbf{B}}_*\| \\
&= \mathrm{dist}(\tilde{\mathbf{B}}_*, \tilde{\mathbf{B}}_0) \\
&= \delta_0
\end{aligned}
\tag{67}
$$

where (67) follows from (64). Moreover, $\mathbf{B}_{*'}\bar{\mathbf{w}}_* = \mathbf{B}_*\bar{\mathbf{w}}_* \in \mathrm{col}(\mathbf{B}_0)$, thus $\mathbf{B}_{*'} \in \mathcal{B}$. Next,

$$
\begin{aligned}
\mathrm{dist}(\mathbf{B}_*, \mathbf{B}_{*'}) &= \|(\mathbf{I}_d - \mathbf{B}_*\mathbf{B}_*^\top)\mathbf{B}_{*'}\| \\
&= \|(\mathbf{I}_d - \tfrac{1}{\|\bar{\mathbf{w}}_*\|^2}\mathbf{B}_*\bar{\mathbf{w}}_*\bar{\mathbf{w}}_*^\top\mathbf{B}_*^\top - \tilde{\mathbf{B}}_*\tilde{\mathbf{B}}_*^\top)\big(\tfrac{1}{\|\bar{\mathbf{w}}_*\|^2}\mathbf{B}_*\bar{\mathbf{w}}_*\bar{\mathbf{w}}_*^\top + (2\tilde{\mathbf{B}}_0\tilde{\mathbf{B}}_0^\top\tilde{\mathbf{B}}_* - \tilde{\mathbf{B}}_*)\tilde{\mathbf{V}}_*^\top)\| \\
&= \|(\mathbf{I}_d - \tilde{\mathbf{B}}_*\tilde{\mathbf{B}}_*^\top)(2\tilde{\mathbf{B}}_0\tilde{\mathbf{B}}_0^\top\tilde{\mathbf{B}}_* - \tilde{\mathbf{B}}_*)\| \\
&= 2\|(\mathbf{I}_d - \tilde{\mathbf{B}}_*\tilde{\mathbf{B}}_*^\top)\tilde{\mathbf{B}}_0\tilde{\mathbf{B}}_0^\top\tilde{\mathbf{B}}_*\| \\
&\geq 2\|(\mathbf{I}_d - \tilde{\mathbf{B}}_*\tilde{\mathbf{B}}_*^\top)\tilde{\mathbf{B}}_0\|\sigma_{\min}(\tilde{\mathbf{B}}_0^\top\tilde{\mathbf{B}}_*) \\
&= 2\,\mathrm{dist}(\tilde{\mathbf{B}}_0, \tilde{\mathbf{B}}_*)\sqrt{1 - \mathrm{dist}^2(\tilde{\mathbf{B}}_0, \tilde{\mathbf{B}}_*)} \\
&= 2\delta_0\sqrt{1 - \delta_0}
\end{aligned}
\tag{68}
$$

where (68) follows since $\sigma_{\min}^2(\tilde{\mathbf{B}}_1^\top\tilde{\mathbf{B}}_2) + \sigma_{\max}^2((\mathbf{I}_d - \tilde{\mathbf{B}}_1\tilde{\mathbf{B}}_1^\top)\tilde{\mathbf{B}}_2) = 1$ for any $\tilde{\mathbf{B}}_1, \tilde{\mathbf{B}}_2 \in \mathcal{O}^{d,k-1}$.

Note that for D-GD the global update for the representation is

$$
\mathbf{B}_{t+1} = \mathbf{B}_t - \frac{\alpha}{M}\sum_{i=1}^{M}\nabla_{\mathbf{B}}f_i(\mathbf{B}_t, \mathbf{w}_t) = \mathbf{B}_t - \alpha(\mathbf{B}_t\mathbf{w}_t - \mathbf{B}_*\bar{\mathbf{w}}_*)\mathbf{w}_t^\top,
\tag{69}
$$

and similarly, the update for the head is $\mathbf{w}_{t+1} = \mathbf{w}_t - \alpha\mathbf{B}_t^\top(\mathbf{B}_t\mathbf{w}_t - \mathbf{B}_*\bar{\mathbf{w}}_*)$. Thus, the behavior of D-GD is indistinguishable in the settings with ground-truth representations $\mathbf{B}_{*'}, \mathbf{B}_*$ since $\mathbf{B}_{*'}\bar{\mathbf{w}}_* = \mathbf{B}_*\bar{\mathbf{w}}_*$. In particular, $\mathbf{B}_T^{\text{D-GD}}(\mathbf{B}_0, \mathbf{B}_*, \{\mathbf{w}_{*,i}\}_{i=1}^{M}, \alpha) = \mathbf{B}_T^{\text{D-GD}}(\mathbf{B}_0, \mathbf{B}_{*'}, \{\mathbf{w}_{*,i}\}_{i=1}^{M}, \alpha)$ Using this equality along with the triangle inequality yields

$$
\begin{aligned}
\mathrm{dist}(\mathbf{B}_T^{\text{D-GD}}(\mathbf{B}_0, \mathbf{B}_{*'}, \{\mathbf{w}_{*,i}\}_{i=1}^{M}, \alpha), \mathbf{B}_{*'}) &= \mathrm{dist}(\mathbf{B}_T^{\text{D-GD}}(\mathbf{B}_0, \mathbf{B}_*, \{\mathbf{w}_{*,i}\}_{i=1}^{M}, \alpha), \mathbf{B}_{*'}) \\
&\geq \mathrm{dist}(\mathbf{B}_*, \mathbf{B}_{*'}) - \mathrm{dist}(\mathrm{dist}(\mathbf{B}_T^{\text{D-GD}}(\mathbf{B}_0, \mathbf{B}_*, \{\mathbf{w}_{*,i}\}_{i=1}^{M}, \alpha), \mathbf{B}_*) \\
&\geq 2\delta_0\sqrt{1 - \delta_0^2} - 0.7\delta_0 \\
&\geq (\sqrt{3} - 0.7)\delta_0 \\
&\geq \delta_0
\end{aligned}
$$

$$
\tag{70}
$$
$$
\tag{71}
$$

as desired, where (70) follows by the definition of case (2) and (68), and (71) follows by $\delta_0 \in (0, 1/2]$. $\qquad\square$

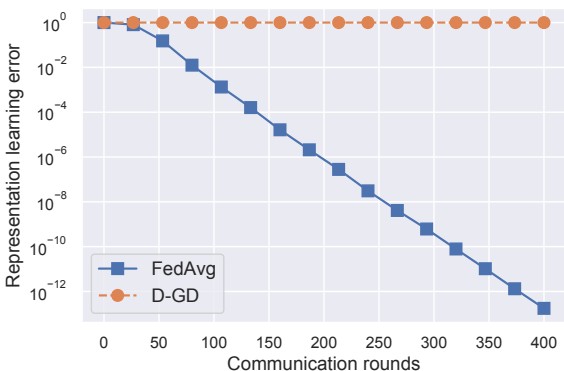

Figure 6: FedAvg with partial participation ($m = 10$, $M = 40$) learns the ground-truth representation at a linear rate.

## C   Experimental Details

### C.1   Multi-task linear regression

The multi-task linear regression experiments consist of two stages: training and fine-tuning. During training, we track $\mathrm{dist}(\mathbf{B}_t, \mathbf{B}_*)$ in Figure 1 and the Frobenius norm of the gradient of (6), i.e. $(\|\mathbf{B}_t^\top(\mathbf{B}_t\mathbf{w}_t - \mathbf{B}_*\bar{\mathbf{w}}_*)\|_2^2 + \|(\mathbf{B}_t\mathbf{w}_t - \mathbf{B}_*\bar{\mathbf{w}}_*)\mathbf{w}_t^\top\|_F^2)^{1/2}$, in Figure 3(left). For fine-tuning, we track the squared Euclidean distance of the post-fine-tuned model from the ground-truth in Figure 3(right) for various numbers of fine-tuning samples $n$.

Each training trial consists of first sampling $M$ ground truth heads $\mathbf{w}_{*,i} \sim \mathcal{N}(\mathbf{0}, \mathbf{I}_k)$ and a ground truth representation $\check{\mathbf{B}}_* \in \mathbb{R}^{d \times k}$ such that each element is i.i.d. sampled from a standard Gaussian distribution. Then, $\mathbf{B}_*$ is formed by computing the QR factorization of $\check{\mathbf{B}}_*$, i.e. $\mathbf{B}_*\mathbf{R}_* = \check{\mathbf{B}}$, where $\mathbf{R}_* \in \mathbb{R}^{k \times k}$ is upper triangular and $\mathbf{B}_* \in \mathcal{O}^{d \times k}$ has orthonormal columns. To initialize the model we set $\mathbf{w}_0 = \mathbf{0} \in \mathbb{R}^k$ and sample $\check{\mathbf{B}}_0 \in \mathbb{R}^{d \times k}$ such that each element is i.i.d. sampled from a standard Gaussian distribution, then compute $\mathbf{B}_0 = \frac{1}{\sqrt{\alpha}}\hat{\mathbf{B}}_0$ where $\hat{\mathbf{B}}_0 \in \mathcal{O}^{d \times k}$ is the matrix with orthonormal columns resulting from the QR factorization of $\check{\mathbf{B}}_0$. Then we run FedAvg with $\tau = 2$ and D-SGD on the population objective 6, with both sampling $m = M$ clients per round and using step size $\alpha = 0.4$. The training plots show quantities averaged over 10 independent trials.

For each fine-tuning trial, we similarly draw a new head $\mathbf{w}_{*,M+1} \sim \mathcal{N}(\mathbf{0}, \mathbf{I}_k)$, and data $\mathbf{x}_{M+1,j} \sim \mathcal{N}(\mathbf{0}, \mathbf{I}_k)$, $y_{M+1,j} = \langle \mathbf{B}_*\mathbf{w}_{*,i}, \mathbf{x}_{M+1,j}\rangle + \zeta_{M+1,j}$ where $\zeta_{M+1,j} \sim \mathcal{N}(0, 0.01)$. Then we run GD on the empirical loss $\frac{1}{2n}\sum_{j=1}^n(\langle \mathbf{B}\mathbf{w}, \mathbf{x}_{M+1,j}\rangle - \mathbf{y}_{M+1,j})^2$ for $\tau' = 200$ iterations with step size $\alpha = 0.01$. The fine-tuning plot shows average results over 10 independent, end-to-end trials (starting with training), and the error bars give standard deviations.

**Remark 1** (Full participation). *Although the aforementioned experiments used full participation ($m = M$), this condition is not required for FedAvg to learn the ground-truth representation. Figure 6 plots the representation learning error in the same setting as Figure 1 but with $m = 10$ (and $M = 40$) and one can see that FedAvg again learns the representation.*

**Remark 2** (Convergence to stationary point of the global objective). *The literature has thus far focused on analyzing the rates at which FedAvg converges to a stationary point of the global objective (1), which prior works have shown requires step size diminishing with $T$ to achieve at-best sublinear convergence in nonconvex data heterogeneous settings. In contrast, Theorem 1 and Figure 4 show that FedAvg with* fixed *step size converges* linearly *to the ground-truth representation despite* not *converging to a stationary point of the global objective.*

## C.2 Image classification

The CNN used in the image classification experiments has six convolutional layers with ReLU activations and max pooling after every other layer. On top of the six convolutional layers is a 3-layer MLP with ReLU activations.

All models are trained with a step size of $\alpha = 0.1$ after tuning in $\{0.5, 0.1, 0.05, 0.01, 0.005\}$ and selecting the best $\alpha$ that yields the smallest training loss. In all cases, $m = 0.1M$. We use the SGD optimizer with weight decay $10^{-4}$ and momentum 0.5. We train all models such that $T\tau = 125000$. Thus, D-SGD trains for $T = 125000$ rounds, since $\tau = 1$ in this case. Likewise, for FedAvg with $\tau = 50$, $T = 2500$ training rounds are executed. The batch size is 10 in all cases. We also experimented with larger batch sizes for D-SGD but they did not improve performance.

Each client has 500 training samples in all cases. Thus, FedAvg with $\tau = 50$ is equivalent to FedAvg with one local epoch. For the experiments with $C$ classes per client for all clients, each client has the same number of images from each class. For the experiment testing fine-tuning performance on new classes from the same dataset, the first 80 classes for CIFAR100 are used for training, while classes 80-99 are reserved for new clients. For fine-tuning, 10 epochs of SGD are executed on the training data for the new client. For fine-tuning on CIFAR10, each new client has images from all 10 classes, and equal numbers of samples per class for training. For fine-tuning on CIFAR100, each new client has images from all 20 classes, with equal numbers of samples from each class for training when possible, and either 2 or 3 samples from each class otherwise (when the number of fine-tuning samples equals 50). Accuracies are top 1 accuracies evaluated on 2000 test samples per client for CIFAR10 and 400 test samples per client for the last 20 classes of CIFAR100.

To compute the layer-wise similarities in Figure 2, we use the Centered Kernel Alignment (CKA) similarity metric, which is the most common metric used to measure the similarity between neural networks [13]. CKA similarity between model layers is evaluated by feeding the same input through both networks and computing the similarity between the outputs of the layers. The similarity metric is invariant to rotations and isotropic scaling of the layer outputs [13]. We use the code from [76] to compute CKA similarity.

For Figure 4, the cosine similarity is evaluated by first feeding $n$ images from class $c$ through the trained network and storing the output of the network layer before the final linear layer to obtain the features $\{\mathbf{f}_1^c, \ldots, \mathbf{f}_n^c\}$, where each $\mathbf{f}_i^c \in \mathbb{R}^{512}$. Then, to obtain the average cosine similarity between features from classes $c$ and $c'$, we compute $\frac{1}{n} \sum_{i=1}^{n} \frac{|\langle \mathbf{f}_i^c, \mathbf{f}_i^{c'} \rangle|}{\|\mathbf{f}_i^c\| \|\mathbf{f}_i^{c'}\|}$. We use $n = 25$.

All experiments were performed on two 8GB NVIDIA GeForce RTX 2070 GPUs.