# OpenReview forum: "FedAvg with Fine Tuning: Local Updates Lead to Representation Learning"
_NeurIPS.cc/2022/Conference — NeurIPS 2022 Accept_

### Official Review · Reviewer_1GSs · 2022-07-11

**Rating:** 7
**Confidence:** 3
**Soundness:** 3 good
**Presentation:** 3 good
**Contribution:** 3 good

**Summary:**

This paper analyzes both theoretically and empirically the performances of FedAvg with local fine-tuning as a post-processing step. More precisely, they consider the scenario where the model parameters are split into two distinct parts: a shared backbone model and a personalized head model. This scenario encompasses for instance convolutional neural networks where the backbone model extract high-level features and the head performs prediction. In order to derive theoretical results, the authors restrict their analysis to the linear case where the backbone parameter is a matrix $B \in \mathbb{R}^{d \times k}$ with $k < d$ and $\omega$ is the head.
Their main result is a Theorem showing the generalization power of FedAvg with at least 2 local updates via convergence towards the ground-truth backbone parameter $B_\star$. They validate their theory with experiments.

**Questions:**

See the previous remarks and concerns.

**Strengths And Weaknesses:**

Overall, the paper is well-written and clear. The related work section acknowledging previous works is sufficient.

My main concerns / remarks are:

* The work by Collins et al. (ICML 2021) on shared representations should be compared more deeply with the proposed analysis since the two setting bear some similarity. Some comparison has been done in the supplement but should appear in the main paper in my opinion.
* How difficult the analysis beyond the linear case is?

---

> ### Author Response · Authors · 2022-08-02
> **Response to Reviewer 1GSs**
>
> Thank you very much for taking the time to read our paper and give constructive feedback.
>
> **W1: The work by Collins et al. (ICML 2021) on shared representations should be compared more deeply with the proposed analysis since the two setting bear some similarity. Some comparison has been done in the supplement but should appear in the main paper in my opinion.** This is a good point as Collins et al. (2021) consider a similar setting as ours. However, there are several key differences. First, Collins et al. (2021) consider a specialized algorithm that learns  client-specific heads, whereas we consider the ubiquitous FedAvg which learns a single shared model, including a shared head. Our case is more difficult since we must show the locally-updated heads become sufficiently diverse due to local updates, whereas in (Collins et al. 2021), the client-specific heads are already diverse prior to local updates. Also, Collins et al. (2021) considers an alternating update scheme and only one local update for the representation, whereas we consider simultaneous local updates, including potentially many updates for the representation, consistent with FedAvg.
>
> **W2: How difficult the analysis beyond the linear case is?**  In nonlinear settings, we also seek a representation that generalizes to new data from the training clients as well as data from new clients. Again client diversity and local updates should be critical to obtaining information from the true representation. However, working out the details for such settings, e.g. the two-layer non-linear neural network case, is a substantial challenge worthy of a new paper, since the updates are more complicated. We believe that our work provides the foundation of understanding the role of representation learning in the success of FedAvg, and studying nonlinear settings can be considered as a future research direction and is beyond the scope of this paper.

---

### Official Review · Reviewer_rvkq · 2022-07-12

**Rating:** 6
**Confidence:** 2
**Soundness:** 2 fair
**Presentation:** 2 fair
**Contribution:** 2 fair

**Summary:**

This paper provides theoretical analysis of FedAvg with Fine Tuning. When the local model weights for clients lie on a linear subspace, theorem 1 suggests FedAvg gets linear convergence, while proposition 1 suggests Distributed GD may diverge for representation learning. Some empirical results on a toy problem are used to verify the theorem, and results on CIFAR-10 and CIFAR-100 shows FedAvg+FineTune outperforms D-SGD+FineTune.

**Questions:**

Like I mentioned,  I find it difficult to connect Theorem 1 to the FedAvg algorithm described in Section 2 and the objective eq. (5).  Specifically, what is the connection between (5) and (7), why do we care about  \|Bw − B∗w∗\|^2 in (7)?

Could the authors clarify what assumptions of FedAvg, and what assumptions of representation learning are made?

Related to previous questions, it is not quite clear to me what is the difference between a FL setting and a multi-task/transfer learning setting.

Could the authors discuss and compare to Adaptive Gradient-Based Meta-Learning Methods https://arxiv.org/pdf/1906.02717.pdf ? I am not an expert on meta-learning/representation learning theory, and this is only one of the papers that also discuss FedAvg.


**Strengths And Weaknesses:**

Strengths
+ FedAvg is one of the most popular algorithms in FL. Though the connection between FedAvg and MAML/Reptile is known, the theoretical justification of the advantage of FedAvg for representation is (to my knowledge) novel.

Weakness
- The paper can probably improve on clarity. I find it difficult to connect Theorem 1 to the FedAvg algorithm described in Section 2 and the objective eq. (5).
- As the authors acknowledged, previous work has drawn connections between FedAvg and MAML, and empirically shown FedAvg+FineTune is better than SGD+FineTune. The main contribution of this paper is theoretical, while the assumptions are very strong.

---

> ### Author Response · Authors · 2022-08-02
> **Response to Reviewer rvkq**
>
> Thank you very much for taking the time to read our paper and give constructive feedback.
>
> **W1Q1: Connection between Theorem 1 and FedAvg algorithm described in Section 2?** Thank you for raising this point. Theorem 1 studies the population version of (5), which is formally introduced in (7) and is derived by taking the number of samples per client to infinity. As mentioned within the statement of Theorem 1 our convergence guarantees are for the population case in (7). In the revised paper, we have also highlighted this point before introducing Theorem 1.
>
> **W2a: As the authors acknowledged, previous work has drawn connections between FedAvg and MAML, and empirically shown FedAvg+FineTune is better than SGD+FineTune. The main contribution of this paper is theoretical, while the assumptions are very strong.** We must emphasize the novelty of our contributions. First, our theoretical results provide a totally new perspective on why FedAvg  empirically yields models that generalize well, noting that the vast majority of prior analyses of FedAvg focus on its convergence (or lack thereof) to a stationary point of the global objective and ignore the generalizability of its solutions, let alone consider the representation learning ability of FedAvg. Second, we support our theoretical claims with empirical study of FedAvg vs FedSGD from a representation learning point of view that has *not* been considered in prior works. Forming a connection between MAML and FedAvg is not one of the contributions of this paper. We should also mention that, to the best of our knowledge, no prior work has proven that FedAvg is capable of representation learning (in any setting). Please see W2b and Q2 for clarification on our assumptions.
>
> **W2b: Contributions are mostly theoretical with strong assumptions.**  We believe our theoretical results are strong enough to stand alone since they provide a novel, rigorous perspective on why FedAvg is effective. Nevertheless, we have also provided a novel empirical study of FedAvg vs FedSGD/D-SGD from a representation learning perspective.  As noted in our response to Reviewer EBRg, we consider a simplified linear setting, but it is appropriate to consider such a setting as no prior works have shown whether FedAvg can learn an effective representation. We would also like to point out that the multi-task linear regression setting we consider is the typical setting for studying representation learning among recent works, and these works make similar assumptions as we do [42,44,45 in paper].
>
> **Q2: Could the authors clarify what assumptions of FedAvg, and what assumptions of representation learning are made?** Our assumptions are formally stated in the exposition; we will informally state them here. We require that the ground-truth heads are bounded (Assumption 1) and are diverse, i.e. well-spread, in $\mathbb{R}^k$ (Assumption 2). For initialization of $\mathbf{B}_0$, we require that its principal angle distance from $\mathbf{B}_\ast$ is bounded by only a constant away from 1 (basically, the column spaces cannot be perpendicular in any direction), and that it is close to a scaled orthonormal matrix ($\mathbf{I}_k- \alpha \mathbf{B}_0^\top \mathbf{B}_0$ is small). We also require that $\mathbf{w}_0$ is close to zero. Finally, we assume that each client has access to its population gradients (please see W1Q1), and sufficiently many clients participate per round (please see response to Reviewer Qkio, L1).
>
> **Q3: FL vs multi-task/transfer learning.** Federated learning is an instance of multi-task learning if we consider each client's objective as a task. However, FL has many more constraints than most multi-task learning settings. Most importantly, the data for each client/task cannot be collected centrally due to privacy and computation mandates, so any federated learning algorithm (including FedAvg) must make local updates, which are uncommon in multi-task learning to our knowledge. Another key point is that most multi-task learning approaches aim to learn task-specific parameters (e.g. a shared representation and task-specific heads) whereas FedAvg learns a single shared model among all tasks/clients, and relies on task/client-specific fine-tuning to generalize well on each task.
>
> **Q4: Comparison with Adaptive Gradient-Based Meta-Learning Methods.** This paper proposes a method for improving meta-learning algorithms by modifying the learning rate according to the learned similarities between tasks. Although they run an experiment in a federated setting, their contributions are quite different from ours due to: (1) they do not study representation learning, (2) they do not study FedAvg, and (3) they do not make observations about the benefit of local updates and task diversity to learning.

---

### Official Review · Reviewer_Qkio · 2022-07-14

**Rating:** 7
**Confidence:** 4
**Soundness:** 3 good
**Presentation:** 3 good
**Contribution:** 3 good

**Summary:**

This work shows that FedAvg learns better representations than Distributed SGD; with a largely theoretical contribution, and some well executed small-scale empirical results. This paper:
* considers multi-task linear regression in a non-convex setting (i.e., prediction is the composition of a learnable subspace matrix, and a linear head), where the shared component across tasks lives in a low-dimensional subspace. The works shows that FedAvg can regress to the correct concept (given a good initialization), because local client-updates in FedAvg encourage diverse high-rank representations, whereas Distributed SGD finds low-rank representations.
* provide empirical experiments (both linear regression and CNN-based image classification), showing that FedAvg discovers better representations, that generalize better than Distributed SGD.


**Questions:**

* In related work, please discuss the relation to work in the gossip literature. In general, gossip-algorithm can be seen as a a generalization of Local-GD; e.g., with diminishing step-sizes, Local-SGD with an arbitrary number of local steps converges to a stationary point of the global objective [1,2]. If clients perform a different, and potentially time-varying, number of local-steps, then from [3] we can also see that they will minimize a weighted average of the global objective, and that this can be corrected with knowledge of agents’ local update rates. Moreover, gradient tracking was actually first proposed in the gossip literature (e.g., see [4])

* Error in line 177; wbar_{*,i) with a summation over i^\prime... I believe here you should just have wbar_* without a summation

* On lines 205: if you analyze the sequence produced immediately after each synchronization step, and consider a diminishing step-size, this will converge (e.g., this is can be seen as a special case of Subgradient-Push, with a particular mixing matrix over a \tau-strongly connected graph). The intuition is that the mixing converges at an R-linear rate (much faster than the sublinear rate of optimization with a diminishing step-size).

* It is not clear to me how the method can discover the correct subspace, but not converge to a stationary point of the loss… e.g., after finding the correct subspace, fix B and optimize for w… this is a convex problem. I believe figure 3 supports this intuition, (i.e., the evolution of the loss with and without fine-tuning the head); please clarify this in the exposition.

* From the analysis, one implicit requirement for the diversity of the discovered representation in the proof is that the clients do not overfit to their local objectives, as this would remove energy from other elements in the column-space, not aligned with the local objective, what is the role of \tau in this case? is there an optimal range for the number of local steps?

[1] Nedic and Olshevsky, Distributed optimization over time-varying directed graphs, IEEE TAC 2014.
[2] Assran et al., Stochastic Gradient Push for Distributed Deep Learning, ICML 2019.
[3] Assran and Rabbat, Asynchronous Gradient Push, IEEE TAC 2021.
[4] Xi and Khan, DEXTRA: A fast algorithm for optimization over directed graphs, IEEE TAC 2017.

**Limitations:**

Some minor limitations on the analysis should be stated; e.g.,
* The second term in the min condition in theorem 1 can easily be vacuous for any reasonable number of iterations, and thus the theorem requires all clients to participate in each round… this should be made clear in a remark following the theorem. Numerical results in section 5.1 also have all clients participate in each round.


**Strengths And Weaknesses:**

Originality: Despite the growing literature of work on the role of federated learning in representation learning, this work is very interesting and quite timely. Rather than proposing a new method, the authors provide an analysis explaining why FedAvg can produce more generalizable representations than Distributed SGD.

Quality: Despite minor errors (probably typos in the math), this work is sound.

Clarity: The exposition is clear, although the proof sketches are a little obscure; and cannot be as readily comprehended without going through the proofs themselves.

I am happy to recommend this work for acceptance.

---

> ### Author Response · Authors · 2022-08-02
> **Response to Reviewer Qkio**
>
> Thank you very much for taking the time to read our paper and provide constructive feedback.
>
> **Q1: Gossip literature.** Thank you for highlighting these contributions.  We have added these references to the revised paper in the related works in the main body and in Appendix A. [1,2,3] study algorithms that generalize FedAvg, but converge in $T = \Omega(M^{2 M \tau})$ [1,3] or $T = \Omega(M^{2\tau^2})$ [2]  iterations in the federated case, rendering these results vacuous in settings with a large number of clients and local updates. The setting in [4] does not include FedAvg since it considers a static, strongly-connected graph. Of course, these points are in a sense moot because we are concerned with generalization via representation learning, not convergence to a first-order stationary point (FOSP) of the global objective. These works use a diminishing step size to show convergence to the global objective at sublinear rates, whereas our work introduces a novel perspective by showing that FedAvg with *fixed* step size and multiple local updates leads to *linear* convergence to the ground-truth representation.
>
> **Q2: $\mathbf{\bar{w}}_{\ast,i}$ typo.** Thank you for noticing - we have corrected it in the revised paper.
>
> **Q3: Diminishing step size will lead to convergence.** This is an accurate point. It has also been observed in the federated learning literature that diminishing step size is necessary for FedAvg to solve the global objective in data heterogeneous settings, leading to sublinear convergence rates [e.g. 5,9,16 in paper]. However, as we discuss in our response to Q1, we are interested in learning a generalizable model by finding the ground-truth representation, not converging to a FOSP of the global training objective. These prior works evince an important point: the literature has focused on the convergence of algorithms to stationary points of the global objective, whereas our work presents a new perspective on how to analyze distributed algorithms that is independent of their convergence to a global FOSP. Specifically, we show that FedAvg with *fixed* step size converges *linearly* to the ground-truth representation in a nonconvex, data heterogeneous setting *despite not converging* to a FOSP of the global objective. We have updated the related work in the main paper to address this point, and also added a longer remark to this effect in the revision in Appendix C.1, due to space constraints.
>
> **Q4: Convergence to stationary point vs subspace learning.** Again, this is a very good point.  The reviewer is correct in that if we fixed $\mathbf{B}$ after learning the ground-truth representation, the subsequent iterates of $\mathbf{w}$ would converge to a FOSP of the global loss. However, this is not FedAvg -- FedAvg does not fix learnable parameters. Even if $\mathbf{B}\_t = \mathbf{B}\_\ast$, the fact that $\mathbf{w}\_t \neq \mathbf{w}\_{\ast,i}$ for at least one $i$ means that the corresponding local gradients for both the representation and head are nonzero, so both variables change locally, and there is no guarantee that $\mathbf{B}\_{t+1}\mathbf{w}\_{t+1}= \mathbf{B}\_\ast\mathbf{\bar{w}}\_{\ast}$, which is required for $(\mathbf{B}\_{t+1},\mathbf{w}\_{t+1})$ to be a FOSP of the global objective (7). Again, this observation aligns with prior work that has noticed that FedAvg with fixed step size may not converge to a FOSP of the global objective in data heterogeneous settings [5,9,16]. Fig. 3 shows the loss of the model learned by FedAvg after fine-tuning the entire model on a *new* client, which is unrelated to the global objective (7).
>
> **Q5: Role of $\tau$.** The role of $\tau$ in preventing overfitting is captured in the upper bound on the step size of the form $\alpha=O({\tau}^{-1/2})$. Thus, $\tau = O(\alpha^{-2})$. Furthermore, since Thm. 1 shows the number of iterations required for convergence decreases with $\tau$, the ideal choice of $\tau$ is as large as possible while still satisfying $\tau = O(\alpha^{-2})$. We've highlighted this point in line 215 in the revised paper.
>
> **L1: Number of clients $m$ active/round.** Regarding the lower bound on $m$ in Thm. 1: Recall that the lower bound on the number of clients sampled per round is of the form $m \geq \min(M,\Omega(\log(kT)))$.The number of iterations $T$ is a function of the desired representation learning accuracy $\epsilon$, in particular $T=O(\log(\tfrac{1}{\epsilon}))$. Thus, to achieve $\epsilon$-accuracy we require $m\geq\min(M,\Omega(\log(k\log(\tfrac{1}{\epsilon}))))$, where the second term in the min is small for any reasonable $\epsilon$.
>
> Regarding the experiments for the linear setting: we chose full participation to allow for an idealized comparison of FedAvg vs D-GD. However this choice is not needed for FedAvg to converge to the ground-truth representation.  We have added a plot showing that FedAvg converges to the ground-truth representation with partial participation to Appendix C1, along with a remark.

---

### Official Review · Reviewer_EBRg · 2022-07-18

**Rating:** 7
**Confidence:** 3
**Soundness:** 3 good
**Presentation:** 4 excellent
**Contribution:** 3 good

**Summary:**

Motivated by the empirical success of Federated Averaging to generalize to new tasks (after fine-tuning), this paper seeks to connect this empirical success with a theoretical argument for how well FedAvg is capable of learning good representations, which is not an explicit objective of the optimization. Theoretical guarantees are presented for a multi-task linear regression setting and further empirical results demonstrate the effectiveness of learning representations with image classification tasks.

**Questions:**

While much of the empirical success of FedAvg was discussed in the paper, I would appreciate a discussion of related work that specifically addresses the representation learning aspect [1,2] or extensions that aim to improve the personalization aspect [3] and how they fit with the assumptions and insights provided by this paper's theoretical analysis.

[1]: Zhuang, W., Gan, X., Wen, Y., Zhang, S., & Yi, S. (2021). Collaborative unsupervised visual representation learning from decentralized data. In Proceedings of the IEEE/CVF International Conference on Computer Vision (pp. 4912-4921).

[2]: Li, Q., He, B., & Song, D. (2021). Model-contrastive federated learning. In Proceedings of the IEEE/CVF Conference on Computer Vision and Pattern Recognition (pp. 10713-10722).

[3]: Hahn, S. J., Jeong, M., & Lee, J. (2021). Connecting Low-Loss Subspace for Personalized Federated Learning. arXiv preprint arXiv:2109.07628.

**Limitations:**

The paper adequately addresses its limitations.

**Strengths And Weaknesses:**

### Strengths
- The paper is extremely well written and well organized. The mathematical notation is clearly described and easy to follow.
- The theoretical results are well presented, with clear assumptions and, at each steps, the assumptions and statements are described in an intuitive way.
- While the result under study (effectiveness of FedAvg for Representation Learning) has been shown (empirically) before, the formalization is an original contribution
- The experiments are well described and effective at supporting the main theoretical claims

### Weaknesses
- The paper ends abruptly. It would be nice to have a conclusion section to tie up the paper. Perhaps some of the problem setting and notation could be abbreviated or moved to the appendix.
- The main result is proved in simplified setting (linear regression tasks), which is far from the usual application of FedAvg for deep learning

---

> ### Author Response · Authors · 2022-08-02
> **Response to Reviewer EBRg**
>
> Thank you very much for taking the time to read our paper and provide constructive feedback.
>
> **W1: The paper ends abruptly. It would be nice to have a conclusion section to tie up the paper...** Thanks for raising this point. We have added a conclusion to the revised paper. Please check the revised paper.
>
> **W2: The main result is proved in simplified setting (linear regression tasks), which is far from the usual application of FedAvg for deep learning.** Indeed the multi-task linear regression setting we study is a simplified version of the settings in which FedAvg is used in practice. However, even this setting presents difficult analytical challenges (controlling local updates, showing aggregation leads towards the true column space, etc.) requiring a careful inductive argument to overcome. It is in part due to these challenges that *no prior works have shown that FedAvg is capable of representation learning in any setting*. Our work is thus a foundation for analyzing FedAvg from a representation learning perspective.
> Moreover, our experiments on non-linear networks suggest that our insights generalize to more realistic scenarios. As discussed in the new conclusion of the paper, this paper is the first theoretical study towards understanding the connection between the generalization properties of the model trained with FedAvg and Representation Learning. Extending our theoretical results to the nonlinear setting is a future research direction that we plan to explore. We have mentioned this point in the conclusion of the revised paper. Thanks for your comment.
>
> **Q1: While much of the empirical success of FedAvg was discussed in the paper, I would appreciate a discussion of related work that specifically addresses the representation learning aspect [1,2] or extensions that aim to improve the personalization aspect [3] and how they fit with the assumptions and insights provided by this paper's theoretical analysis.** Thanks for bringing these papers to our attention. We have added these references to the revised paper the related works in the main body and in Appendix A. We would first like to note that the representation-learning abilities of FedAvg are not well-studied in the literature. Our motivation to study the representation-learning abilities of FedAvg comes from empirical observations that FedAvg + Fine-Tuning performs well, which suggests that FedAvg learns an effective representation. To our knowledge, [2] is the only work that directly observes that FedAvg can learn effective representations, as it provides empirical evidence that FedAvg learns more expressive representations than those learned by local-only training (each client learns separately without any communication) on CIFAR-10. However, the empirical approach in [2] is substantially different from ours in that it does not study the benefit of local updates to learning good representations via a comparison with D-SGD/FedSGD. Moreover, no theoretical analysis is provided.
>
> Reference [1] also studies federated representation learning, but in unsupervised settings and for a novel algorithm distinct from FedAvg due to exponential moving averaging and a divergence-aware predictor update, and the paper does not make any theoretical contributions. [3] empirically studies an algorithm for personalized federated learning in which each client learns a mixture of local and global models. Their contributions are different from ours in that they do not study FedAvg (as FedAvg learns only a global model), do not provide theoretical results, and do not study representation learning. Please see the related works for other personalized federated learning methods more closely related to representation learning [40,48-50]. While there exist many more personalized federated learning methods, these are orthogonal to our work since they do not aim to understand FedAvg or representation learning.

---

> > ### Comment · Reviewer_EBRg · 2022-08-08
> > **Updated score after rebuttal**
> >
> > I thank the authors for their response to mine and other reviewers' comments and for the changes and added discussion on the paper. They address my comments adequately and add to the quality of the paper. I would recommend to accept this paper and I've increased my score to reflect that.

---

### Meta-Review · Area_Chair_tHJe · 2022-08-26

**Recommendation:** Accept
**Confidence:** Certain

**Metareview:**

This work provides an analysis explaining why FedAvg can produce more generalizable representations than distributed SGD. Theoretical guarantees are presented for a multi-task linear regression setting and further empirical results demonstrate the effectiveness of learning representations with image classification tasks. The theoretical analysis presented can be an important building block for the study of more complex settings in federated optimization. All reviewers recommend acceptance.

Please take the (few) suggestions by the reviewer into account, and also incorporate the explanations and clarifications provided during the rebuttal in the camera ready version.


**Award:**

No

---

### Decision · Program_Chairs · 2022-09-14

Accept